# Desirable Effort Fairness and Optimality Trade-offs in Strategic Learning

Valia Efthymiou [* 1]   Ekaterina Fedorova [* 1]   Chara Podimata [* 1]

## Abstract

Strategic classification examines how decision rules interact with agents who strategically adapt their features. Most existing models focus on maximizing predictive performance, assuming agents best respond to the learned classifier. However, real decision-making systems are rarely optimized solely for accuracy: ethical, economic, and institutional considerations often make some feature changes more desirable than others. At the same time, principals may wish to incentivize these changes fairly across heterogeneous agents. While prior work has studied causal structure between features, notions of desirability, and information disparities in isolation, this work initiates a unified treatment of these components within a single framework. We frame the problem as a constrained optimization problem that captures the trade-offs between optimality, desirability, and fairness. We provide theoretical guarantees on the principal's optimality loss constrained to a particular desirability fairness tolerance for multiple broad classes of fairness measures. Finally, through experiments on real datasets, we show the explicit tradeoff between maximizing accuracy and fairness in desirability effort.

## 1. Introduction

Most automated decision-making systems–whose task is to assign a human (aka "agent") a numeric score or a classification based on which a *decision* about the individual is made–must contend with the fact that, given sufficient information and ability, agents may strategically alter their features to improve their assigned outcomes. This strategic alteration jeopardizes the effectiveness of the originally proposed decision-making policies; e.g., if a bank's loan-approval

algorithm positively values having multiple credit cards, agents may take out extra cards to improve approval rate while not truly improving their creditworthiness. Similar human adaptations to automated decision-making rules have been observed in healthcare (Chang et al., 2024; Efthymiou et al., 2025) and recommendation systems (Haupt et al., 2023; Fedorova et al., 2026) among others.

In the classic strategic learning literature (e.g., (Hardt et al., 2016; Dong et al., 2018; Chen et al., 2020)), this problem is modeled as a Stackelberg game: the principal publicly commits to an accuracy-maximizing algorithm, anticipating that agents will best respond by submitting altered features that maximize their utility about predicted outcomes. Formally, the principal commits to a scoring rule $\mathbf{w} \in \mathbb{R}^d$, the agents observe $\mathbf{w}$ along with their *true* feature-score pair $(\mathbf{x}, y)$ (where $\mathbf{x} \in \mathbb{R}^d$ and $y \in [0, 1]$), and each agent reports an *altered* feature $\mathbf{x}'$ (where $\mathbf{x}'$ is a "best-response", i.e., it maximizes the agent's underlying utility function for obtaining a better outcome $\mathbf{w}^\top \mathbf{x}'$). The principal's goal is to identify the optimal $\mathbf{w}$ such that the loss between predictions on the altered features $\mathbf{w}^\top \mathbf{x}'$ and the true scores $y$ is minimized.

This classical model–albeit great at highlighting *some* of the complexities that arise from the agents' best-responding behavior–fails to capture several of the intricacies of the real-world settings it wishes to model. Take for example a content recommendation platform (e.g., YouTube) that algorithmically determines how much to promote a video based on various video features such as topic, title, explicitness etc. Content creators want their videos highly promoted and can change their videos (to an extent) so that the algorithm recommends them to the greatest number of people. This is a nuanced interaction between creators and the platform. On a creator's side, there are several issues that complicate her video changes beyond the perfect best response (i.e., the perfect video edit for maximum promotability): **(1)** She may only know about the algorithm what she/her friends can test/learn; **(2)** Her video edits influence each other (e.g., changing topic also affects how explicit the video is); and **(3)** Some changes really *do* change how popular a video *should* be, i.e., they are not purely gaming of the recommender system! As for the platform, independent of effects on true promotability, creating incentives to alter certain video features might be (un)desirable. For example, clickbait titles may genuinely make clicking the video more appealing,

*Equal contribution [1]Massachusetts Institute of Technology. Correspondence to: Valia Efthymiou <valia554@mit.edu>, Ekaterina Fedorova <fedorova@mit.edu>, Chara Podimata <podimata@mit.edu>.

*Proceedings of the 43rd International Conference on Machine Learning*, Seoul, South Korea. PMLR 306, 2026. Copyright 2026 by the author(s).

thus making the video more promotable, but being known as a platform inundated with clickbait titles (if creators are incentivized to use them) may hurt the platform's reputation to advertisers and users. In the same vein, to be neutral in the eyes of advertisers, a platform may further be concerned with *which creators* are more/less incentivized to clickbait.

Altogether, the aforementioned example illustrates several complexities for the modeling of the *strategic agents* and the *principal* in algorithmic decision making systems. For the agents: **(A1)** they may *not* know the principal's algorithm fully when choosing their best responses; **(A2)** changes in certain features can causally trigger changes in other features; **(A3)** not all feature alterations are "bad"; instead, some represent genuine improvement. And for the principal: **(P1)** independent of their predictive power, some feature changes may be more/less *desirable* for external stakeholders; and so **(P2)** the principal may need to create equitable incentives to improve desirable features even for heterogeneous agents. Putting **(A1) – (A3)** and **(P1) – (P2)** together and focusing on the principal's perspective, we ask:

> **Q1**: *How can the principal provide fair incentives for improvement among heterogeneous groups?*

Incorporating fairness creates a binding constraint on the principal's optimization problem, forcing a trade-off between their original objective (e.g., accuracy) and equitable treatment. This observation frames our next question:

> **Q2**: *How much of his primary objective (e.g., accuracy) does the principal trade off to provide equitable incentives with respect to certain desirable features?*

### 1.1. Our Contributions

We develop a framework to study **Q1** and **Q2**, proposing measures of equitable incentives and characterizing fair and optimal policies. Our main contributions are:

**Our model (Section 2).** We develop a game-theoretic framework that unifies three previously isolated modeling considerations: (i) agents strategically respond based on peer-learned estimates of the classifier rather than perfect information; (ii) feature changes exhibit causal spillovers captured by a contribution matrix; and (iii) an external stakeholder, representing social, institutional, or regulatory interests, designates which features are *desirable* to improve and defines a *discrepancy function* measuring inequity in desirable effort across groups. To remain compliant with the stakeholder, the principal must choose a *fairness tolerance* (i.e., the maximum desirable effort discrepancy they will permit) and constrain their classifier accordingly. This tolerance represents the principal's commitment to equitable treatment: tighter tolerance means stricter fairness but potentially lower accuracy or welfare. While the stakeholder defines fairness (the discrepancy function), the principal

decides how much unfairness to tolerate, facing a trade-off between predictive performance and fairness guarantees. While prior work has addressed (i)–(iii) separately, we are the first to combine all three and analyze the resulting trade-offs. The comprehensive optimization problem enables the principal to find a fair policy accounting for heterogeneity, answering **Q1**. For **Q2**, we theoretically analyze objective value trade-offs in equilibrium as a function of the stakeholder's choice of discrepancy function, for which we consider two broad families: convex and nonconvex measures. This distinction is fundamental and reflects the range of equity criteria stakeholders might reasonably impose.

**Convex discrepancy functions (Section 3).** When the discrepancy function is convex (which includes natural measures like sum-of-absolute and sum-of-squared differences in desirable effort vectors across groups), we upper bound the principal's optimality loss as a function of the fairness tolerance and system parameters to answer **Q2**. This enables the principal to quantify the worst-case trade-offs *before* deployment: given estimates of group adaptation costs and information disparities, a principal can determine how much accuracy or welfare is sacrificed for any fairness tolerance level. Notably, our bounds capture broad classes of discrepancy functions, allowing the principal to assess sensitivity to a range of stakeholder fairness criteria.

**Nonconvex discrepancy functions (Section 4).** Stakeholders may impose *asymmetric* fairness constraints, e.g., requiring that disadvantaged groups not be under-incentivized relative to privileged groups, but permitting the reverse. Such constraints naturally yield nonconvex discrepancy functions, making the principal's optimization problem computationally intractable for *any* fixed tolerance level, an issue for **Q1**. We address this and **Q2** in two steps. First, we construct an ellipsoidal restriction of the feasible set and apply our convex-case bounds for optimality loss guarantees on the restricted problem. Second, we bound the restriction's additional approximation error: the optimality loss from deploying the restricted fair policy rather than the (computationally intractable) true fair optimum. This approach provides both a tractable solution method and quality guarantees for nonconvex fairness constraints.

**Experiments (Section 5).** While our theoretical bounds answer **Q2** by characterizing *worst-case* optimality loss, they do not reveal how specific patterns of group heterogeneity affect equilibrium outcomes. Using the ADULT dataset, we compute the exact fair equilibria under absolute-difference fairness constraints for three demographic splits (age, country, education) with varying cost structures, allowing us to compute the *actual* trade-offs of **Q2**. Furthemore using the Taiwan dataset, we also evaluate our theoretical bounds against the realized losses induced by fairness constraints. We sweep across tolerance levels from highly restrictive

(near-perfect equity) to permissive (substantial inequity), tracing out the complete fairness-accuracy trade-off curve for each split. Our main finding: constraining incentive discrepancy is most costly when group disparities align with desirable features. Because alignment between existing disparities and stakeholder-designated desirable features affects fairness costs, this may effect what is practical in different domains.

## 1.2. Related work

Our work primarily builds on the literature of *strategic learning* and *performative prediction* (see Podimata (2025) for a review), which studies how principals should design optimal algorithms given agents become incentivized to "game" their features to improve outcomes (Hardt et al., 2016; Dong et al., 2018; Chen et al., 2020; Ahmadi et al., 2021; Trachtenberg & Rosenfeld, 2025; Rosenfeld & Rosenfeld, 2024; Shen et al., 2024; Perdomo et al., 2020). Classical strategic classification models treat all strategic behavior as gaming, but as Miller et al. (2020) argue, not all adaptation should be considered "gaming"; oftentimes, it can result in *genuine improvement*. Our work adopts this perspective, but focuses on the (previously overlooked) question of understanding how to provide equitable adaptation incentives for improvement across different subpopulations (**Q1**) while quantifying the trade-offs that the principal must incur as a byproduct (**Q2**).

**Causal models and incentivizing improvement.** Several works incorporate a causal structure into strategic classification. Horowitz & Rosenfeld (2023); Shavit et al. (2020) study principals recovering unknown causal parameters (despite the agents' strategic adaptation), Bechavod et al. (2021) show that strategic responses can actually aid recovery in online settings and Tsirtsis et al. (2024) present hardness results for finding an optimal decision policy given strategic effort investment. Other works try to directly design incentives: Kleinberg & Raghavan (2019) provide algorithms for incentivizing "good" effort profiles in single-shot settings, while Harris et al. (2021) extend this to repeated interactions, Eilat & Rosenfeld (2023) create recommender algorithms to incentivize diverse content creation, and (Penn & Patty) frame strategic adaptations as a $\{0, 1\}$ [non]compliance option and present algorithms to maximize the desired behavior. Alternative mechanism design approaches (Alon et al., 2020; Haghtalab et al., 2021) aim to promote beneficial effort through counterfactual explanations (Tsirtsis & Gomez-Rodriguez, 2020). Most closely related to our causal modeling approach, Efthymiou et al. (2025) study how agents choose desirable effort when features causally affect one another and agents have partial information about both the causal graph and the principal's rule. While we similarly model causal feature interactions and effort desirability, our focus differs: we study the principal's problem of inducing desirable incentives *fairly* across groups, characterizing the optimality loss from imposing fairness constraints.

**Incomplete information** Real agents rarely have perfect knowledge of classifiers. Bechavod et al. (2022) model agents who learn about the classifier through peer observations, finding that under certain group disparities, optimal policies can induce feature changes that worsen true outcomes. We adopt their peer-learning framework but extend it to study how desirable-effort fairness constraints affect principal optimality when agents learn imperfectly. Other works capture agent biases affecting best-response (Ebrahimi et al., 2025), unavailable classifiers (Ghalme et al., 2021), noisy policy signals (Avasarala et al., 2025), Bayesian approaches (Cohen et al., 2025), and randomized classifiers (Braverman & Garg, 2020; Ahmadi et al., 2023).

**Fairness in strategic settings.** Many fairness analyses of strategic learning precede ours. Penn & Patty ask whether principals produce "aligned incentives" between groups. We use a similar fairness metric, but directly constrain the principal to varying fairness levels. Alhanouti & Naghizadeh (2024) also study fairness constraints and find that equalizing true positive rates or acceptance rates can worsen incentive equity between advantaged and disadvantaged groups—the issue we address by directly constraining desirable *incentive* discrepancy rather than *outcomes*. Other work shows that traditionally fair classifiers may not be fair in strategic settings (Estornell et al., 2023) or analyzes other disparate effects of optimal strategic policies (Milli et al., 2019; Hu et al., 2019). Diana et al. (2025) provide algorithms for optimal rules under minimax fairness constraints. Our work differs by (1) focusing on desirable effort fairness (i.e., equalizing incentives for beneficial feature changes rather than outcomes or total costs) and (2) characterizing the principal's accuracy/welfare trade-offs as a function of fairness tolerance and group heterogeneity structure.

Finally, our work connects to the literature in *algorithmic recourse* which studies how a principal may provide explanations or recommended actions to agents who receive unfavorable scores (Kugelgen et al., 2022; Ehyaei et al., 2023; Karimi et al., 2021; Perello et al., 2025; Gupta et al., 2019); see Karimi et al. (2022) for a review. While these settings similarly consider agents who may alter their features, our model (and strategic classification as a whole) takes a more mechanism-design-based approach in that the principal indirectly creates "recourse" for agents by incentivizing changes exclusively through deploying the algorithm or policy rather than direct recommendations to agents.

## 2. Model

**Notation.** $\mathbb{I}_{...}$ is a function s.t. $\mathbb{I}_{...} = 1$ if the subscript is true and $\mathbb{I}_{...} = 0$ otherwise. Matrices are capital (i.e., $M \in \mathbb{R}^{k \times d}$), vectors are bolded lower-case (i.e., $\mathbf{w} \in \mathbb{R}^d$),

**Protocol 1** Decision making system interaction

1: Nature selects causal graph $\mathcal{G}$, ground truth rule $\mathbf{w}^\star$ and cost matrices $A_1$, and $A_2$.
2: Stakeholder selects desirability scores $\Pi_D$ and discrepancy function $\Delta(\mathbf{w})$.
3: Principal chooses desirability discrepancy tolerance $\beta$.
4: Principal uses an optimal (based on OBJ) $\mathbf{w}$ s.t., $\Delta(\mathbf{w}) \leq \beta$.
5: Agents draw initial features $\mathbf{x} \sim \mathcal{D}_g$.
6: Agents reveal features: $\mathbf{x}' \in \arg\max(\texttt{Score}(\mathbf{x}', g) - \texttt{Cost}(\mathbf{x}_e, g))$.

and one-dimensional variables are lower-case (i.e., $y \in \mathbb{R}$). For a vector, $\mathbf{w}$, $\mathbf{w}_+$: $(\mathbf{w}_+)_i = \mathbb{I}_{\mathbf{w}_i \geq 0} \mathbf{w}_i$. For a matrix, $M$, $M_{j,i}$ is the $j$th row $i$th column element. $\ker(M)$ is the *kernel*, i.e., $\{\mathbf{w} : M\mathbf{w} = 0\}$. $H(M)$ is the Hoffman constant (Appendix A.1.1). $\lambda_d(M)$ and $\sigma_d(M)$ are the $d$-th largest eigenvalues and singular values respectively. Sets are capital calligraphic (e.g., $\mathcal{W}(\cdot, \cdot)$). $\mathcal{B}(\rho)$ is a radius $\rho$ Euclidean ball, i.e., $\{\mathbf{w} \in \mathbb{R}^d : \mathbf{w}^\top \mathbf{w} \leq \rho^2\}$. We provide a summary notation table in Appendix A.1.

## 2.1. Model Summary

We begin by summarizing our model; more in-depth analysis of each of the moving pieces follows. We focus on a Stackelberg game between a principal and an agent population comprised by 2 subpopulations [1], with different distributions over the feature space and cost matrices $A_1, A_2$. An agent (she) belongs to group $g \in \{1, 2\}$ and has initial features $\mathbf{x} \in \mathcal{X} \subseteq \mathbb{R}^d$ drawn from distribution $\mathcal{D}_g$ over the feature subspace $\mathcal{S}_g \subseteq \mathcal{X}$. Let $\Pi_g \in \mathbb{R}^{d \times d}$ be the orthogonal projection matrix into a group subspace. Let $\mathbf{w}^\star \in \mathbb{R}^d$ denote the ground truth linear scoring rule; i.e., $\mathbb{E}[y|\mathbf{x}] = \mathbf{w}^{\star\top} \mathbf{x}$ is the expected (true) quality of an agent with features $\mathbf{x} \in \mathcal{X}$. Similarly to (Bechavod et al., 2022), $\mathbf{w}^\star$ may be optimal for *prediction* accuracy, but it may not be optimal given the other principal concerns we analyze. For generality, we model many aspects that may affect group disparities e.g. cost asymmetries. Any of these pieces can be excluded when irrelevant to the setting (e.g. feature changes are free) by setting the relevant matrices to the identity, $\mathbf{I}$.

**Altered features.** In response to information she has learned about $\mathbf{w}$ from her group, $g$, (see Section 2.2 for details) an agent alters her feature vector to $\mathbf{x}'$ in order to maximize her anticipated score, $\texttt{Score}(\mathbf{x}', g)$, minus the cost of alteration, $\texttt{Cost}(\mathbf{x}_e, g)$, where $\mathbf{x}_e$ is the "exogenous effort" for the agent. Following (Efthymiou et al., 2025), we assume that altering one feature may causally affect others. We refer to the initial effort as "exogenous", to distinguish it from the "spillover" effects across features. These spillovers are modeled using a weighted causal graph, where nodes

represent features. The contribution matrix $C$ captures the weighted flow of effort across features. [2] The relationship between $\mathbf{x}'$ and $\mathbf{x}_e$ is such that: $\mathbf{x}' = \mathbf{x} + C\mathbf{x}_e$.

**Definition 2.1 (Contribution matrix)** *Given a weighted directed acyclic graph (DAG) $\mathcal{G} = ([d], \mathcal{A}, \omega)$ where $[d]$ is a set of feature-nodes, $\mathcal{A}$ is a set of edges indicating causality between features, and $\omega$ is a weight function on edges, the contribution matrix $C$ is:*

$$C_{ii} = 1 \ \forall i \in [d], \ C_{ij} = \textstyle\sum_{p \in \mathcal{P}_{ij}} \omega(p) \ \forall i \neq j$$

*$\mathcal{P}_{ij}$ is the set of all direct paths from nodes $i$ to $j$ and $\omega(p)$ for $p \in \mathcal{P}_{ij}$ is the sum of the weights along path $p$, i.e., : $\omega(p) = \sum_{a \in p} \omega(a)$. $\ker(C) = \emptyset$ (Lemma A.1).*

**External stakeholder & principal's goals.** An external stakeholder assigns a "desirability score" to each feature, $\texttt{des}(i) > 0, i \in [d]$; let $\Pi_D := \text{Diag}(\{\texttt{des}(i)\}_{i \in [d]})$ denote the "desirability matrix". Roughly, a higher desirability score for a feature $i$ means that the stakeholder wants to *incentivize* the agent to exert effort and improve $i$. We assume that the external stakeholder also selects a function $\Delta(\mathbf{w}) : \mathbb{R}^d \to \mathbb{R}$, that measures the group discrepancy in desirable effort incentivized by $\mathbf{w}$. To remain trustworthy to the external stakeholder, the principal chooses a desirability discrepancy tolerance, $\beta > 0$, and must find the optimal policy (also referred to as "rule") $\mathbf{w}$ according to his own objective function $\texttt{OBJ}(\mathbf{w}; \mathbf{w}^\star)$; see Section 2.3 for details.

## 2.2. Agents

We assume that the agents do *not* have full information about the scoring rule $\mathbf{w}$. Instead, they engage in *peer learning* as modeled in (Bechavod et al., 2022); each agent sees features $\mathbf{x}_{g,i}$ and policy outcomes, $\hat{y}_{g,i} = \mathbf{x}_{g,i}^\top \mathbf{w}$, of $N_g$ random (nonstrategic) agents from her own group $g$ and does empirical risk minimization (ERM) to get $\mathbf{w}_{\texttt{EST}}$, the estimated policy. $\mathbf{w}_{\texttt{EST}}(g)$ is the solution to the following:

$$\min_{\tilde{\mathbf{w}} \in W} \quad \tilde{\mathbf{w}}^\top \tilde{\mathbf{w}} \tag{1}$$
$$\text{s.t.,} \quad W = \{\mathbf{w} : \mathbf{w} = \arg\min_{\mathbf{w}'} \sum_{i \in N_g} (\mathbf{x}_{g,i}^\top \mathbf{w}' - \hat{y}_{g,i})^2\}$$

Intuitively, each agent does a standard linear (ordinary least-squares, OLS) fit on past agents' data to estimate what policy the learner uses. Because OLS guarantees unbiasedness and asymptotic normality, this is a *rational* estimation of $\mathbf{w}$ assuming data spans the space. However, as OLS may not have a unique solution, tie-breaking is necessary. The norm minimization tie-breaking implies that agents are *risk-averse*. When agents are uncertain, they guess the $\mathbf{w}$ that

---

[1] For more than 2 groups, our work can extended by constraining the incentive disparity between each *pair* of groups.

[2] Note that $C$ is the transpose of the matrix used in (Efthymiou et al., 2025), who adopt a similar causal-flow perspective.

induces smaller feature changes, just in case. Also, note that that $N_g$ does not appear in the solution because agents peer learn from $\hat{y}_{g,i}$, an *non-noisy* outcome. We make this simplification to focus on the learner's fairness implications due to core group features, e.g., cost asymmetry, rather than limited peer learning sample size. Were there mean zero outcome noise, agents' estimates, $\mathbf{w}_{\text{EST}}(g)$ would converge to those of the above equation as $N_g \to \infty$.

In anticipation of the principal's policy $\mathbf{w}$, an agent modifies her features from $\mathbf{x}$ to $\mathbf{x}'$ by exerting exogenous effort, $\mathbf{x}_e \in \mathbb{R}^d$, that is added (after a linear transformation) to her original features. The final $\mathbf{x}'$ is a *best response*, meaning it is the maximizer of her utility function: $\mathcal{U}(\mathbf{x}, \mathbf{x}', g) := \texttt{Score}(\mathbf{x}', g) - \texttt{Cost}(\mathbf{x}_e, g)$ where $\texttt{Score}(\mathbf{x}', g) := \mathbf{w}_{\text{EST}}^\top \mathbf{x}'$ is the score she believes she will have after modification and $\texttt{Cost}(\mathbf{x}_e, g) := \frac{1}{2}\mathbf{x}_e^\top A_g \mathbf{x}_e$ is the cost she must incur for her exogenous effort, $\mathbf{x}_e$ determined by a known positive definite (PD) cost matrix $A_g$. Formally, we write $\mathbf{x}'(\mathbf{x}; \mathbf{w}, g)$ to denote the best-response of an agent from group $g$ with original features $\mathbf{x}$; we drop the dependence on $\mathbf{x}, g$ whenever clear from context. Putting everything together, we can find the closed form for $\mathbf{x}_e$; see Appendix A.1.2 for the proof.

**Proposition 2.1** *Given $\mathbf{w}$, the effort of an agent from group $g$ is $\mathbf{x}_e^{(g)}(\mathbf{w}) = A_g^{-1} C^\top \Pi_g \mathbf{w}$.*

### 2.3. The Principal's Problem

We compare the Stackelberg equilibrium (SE) of Protocol 1 for a selected $\beta$ to the SE were there no stakeholder. These SEs are the solutions to Equations (2) and (3), which we refer to as the *fairness-constrained* and *unconstrained* principal problem respectively. We refer to the solutions of Equations (2) and (3) as $\mathbf{w}_c^\star$ and $\mathbf{w}_u^\star$ respectively.

$$\max_{\mathbf{w} \in \mathcal{B}(1)} \texttt{OBJ}(\mathbf{w}; \mathbf{w}^\star), \quad \text{subject to} \quad \Delta(\mathbf{w}) \le \beta \quad (2)$$

$$\max_{\mathbf{w} \in \mathcal{B}(1)} \texttt{OBJ}(\mathbf{w}; \mathbf{w}^\star) \quad (3)$$

We evaluate *optimality loss*, the decrease in optimal value upon imposing fairness constraints:

**Definition 2.2 (Opt Loss)** $\texttt{OBJ}(\mathbf{w}; \mathbf{w}_u^\star) - \texttt{OBJ}(\mathbf{w}; \mathbf{w}_c^\star)$

Note that the domain of deployable rules is $\{\mathbf{w} | \mathbf{w} \in \mathcal{B}(1)\}$. This emulates that a real-world principal cannot deploy rules with infinite coefficients. Analogously, we can imagine $\mathbf{w}^\star \in \mathcal{B}(1)$ or that policies are normalized. Importantly, the principal's problems have the same objectives and only differ by *feasible region*. The feasible region of the fairness-constrained problem is $\{\mathbf{w} \in \mathbb{R}^d : \Delta(\mathbf{w}) \le \beta\}$ on the domain while the other is just the domain. Because we analyze equilibrium, optimizations are done as if $\mathbf{w}^\star$ is

known. If $\mathbf{w}^\star$ is unknown, then one can follow (Bechavod et al., 2022, Appendix A) and obtain the same results (up to a small error term). To simplify exposition, we use the known $\mathbf{w}^\star$ assumption. The principal chooses between two different objectives: Accuracy (ACC), or Social Welfare (SW). $\Delta(\mathbf{w})$ is the stakeholder-chosen function capturing the discrepancy (across groups) in terms of desirable effort incentivized by rule $\mathbf{w}$.

Formally, the accuracy, [3] $\texttt{ACC}(\mathbf{w}; \mathbf{w}^\star)$ is defined as:

$$-\sum_{g \in [2]} \mathbb{E}_{\mathbf{x} \sim \mathcal{D}_g}\left[\left(\mathbf{w}^{\star\top}\mathbf{x}'(\mathbf{x}; \mathbf{w}, g) - \mathbf{w}^\top \mathbf{x}'(\mathbf{x}; \mathbf{w}, g)\right)^2\right],$$

and the social welfare, $\texttt{SW}(\mathbf{w}; \mathbf{w}^\star)$ is defined as:

$$\sum_{g \in [2]} \mathbb{E}_{x \sim \mathcal{D}_g}[\mathbf{x}'(\mathbf{x}; g)^\top \mathbf{w}^\star]$$

Finally, we turn our attention to the stakeholder-chosen discrepancy function $\Delta(\mathbf{w})$. Recall that after choosing tolerance $\beta$, the principal constrains his problem such that no deployed rule induces desirable effort incentives whose discrepancy across agents, as measured by $\Delta(\mathbf{w})$, is greater than $\beta$. We call the set of $\mathbf{w}$ that satisfy this, $\beta$-fair.

**Definition 2.3 ($\mathcal{W}(\beta; \Delta)$, the set of $\beta$-fair rules)** $\mathcal{W}(\beta; \Delta) := \{\mathbf{w} \in \mathbb{R}^d : \Delta(\mathbf{w}) \le \beta\}$.

Our theoretical guarantees assume little about $\Delta(\cdot)$ beyond its satisfaction of broad properties, so we briefly provide intuition for natural structures of $\Delta(\cdot)$ that will appear throughout the paper. $\mathbf{x}_e^{(g)}(\mathbf{w})$ is the exogenous effort vector across features a group-$g$-agent exerts in best response to what she knows about a policy, $\mathbf{w}$, via peer learning. Thus, $\Pi_D \mathbf{x}_e^{(g)}(\mathbf{w})$ is a *desirability-weighted* effort vector for the agent. Therefore, a natural measure of the desirability discrepancy of a rule is $\Delta(\mathbf{w}) = \texttt{Dist}(\Pi_D \mathbf{x}_e^{(1)}(\mathbf{w}), \Pi_D \mathbf{x}_e^{(2)}(\mathbf{w}))$ where $\texttt{Dist}$ is some vector comparison function.

## 3. Convex Fairness Constraints

In this section, we consider *convex* discrepancy functions, which naturally arise when the stakeholder treats group disparities symmetrically. Two canonical examples are the $\ell_1$ and $\ell_2$ norms applied to the difference in desirability-weighted effort vectors $\Pi_D \mathbf{x}_e^{(g)}(\mathbf{w})$:

**Example 3.1 (Sum of absolute value differences)**

$$\Delta(\mathbf{w}) := \sum_{i \in [d]} \left|(\Pi_D \mathbf{x}_e^{(1)}(\mathbf{w}) - \Pi_D \mathbf{x}_e^{(2)}(\mathbf{w}))_i\right|$$

**Example 3.2 (Sum of squared differences)**

$$\Delta(\mathbf{w}) := \sum_{i \in [d]} (\Pi_D \mathbf{x}_e^{(1)}(\mathbf{w}) - \Pi_D \mathbf{x}_e^{(2)}(\mathbf{w}))_i^2$$

| $\beta$-Fair Space | Feasible Region | Accuracy Loss | SW Loss |
|---|---|---|---|
| no assumptions | convex | $4\max(\|\mathbf{w}^\star\|_2, 1)$ | $2\|\tilde{\mathbf{w}}\|_2$ |
| no assumptions | polyhedron (Property 3.1) | $\left[H(M)\,\|(M\mathbf{w}' - \beta\mathbf{1})_+\|_2\right]^2$ | $\sqrt{2}\|\tilde{\mathbf{w}}\|_2$ |
| ellipsoidal (Property 3.2) | no assumptions | $4\max(\|\mathbf{w}^\star\|_2, 1)$ | $\sqrt{2}\|\tilde{\mathbf{w}}\|_2$ |
| no assumptions | ellipsoidal (Property 3.3) | $2q(s' + s)$ | $\|\tilde{\mathbf{w}}\|_2 - \sqrt{\beta}\,\|\tilde{\mathbf{w}}\|_{Q^{-1}}$ |

*Table 1.* Upper bounds on optimality loss in $\beta$-fair SE under Properties 3.1–3.3. In terms of notation, let $\tilde{\mathbf{w}} := (CA_1^{-1}C^\top\Pi_1 + CA_2^{-1}C^\top\Pi_2)^\top\mathbf{w}^\star$, $\mathbf{w}'$ is the ground-truth classifier $\mathbf{w}^\star$ projected onto the $\ell_2$ unit ball i.e., $\mathbf{w}' = \Pi_{\mathcal{B}(1)}(\mathbf{w}^\star)$, $s = \sqrt{\frac{\beta}{\lambda_d(Q)}}$, $q = \mathbb{I}_{\mathbf{w}^\star\notin\mathcal{E}(\beta)}(s + \|\mathbf{w}^\star\|)$, $s' = \mathbb{I}_{\mathbf{w}^\star\notin\mathcal{E}(\beta)}(\min\{1, \|\mathbf{w}^\star\|\} + s)$ where $\mathcal{E}(\beta)$ is the feasible ellipsoid of Property 3.3.

Geometrically, under some mild conditions (Appendices A.2.3 and A.2.4), the $\ell_1$ and $\ell_2$ discrepancy functions have some "nice" properties: setting $\Delta(\mathbf{w}) \leq \beta$, they form a polyhedron and ellipsoid of feasible rules $\mathbf{w}$, respectively. Generalizing beyond $\ell_1$ and $\ell_2$ discrepancy functions, we focus on $\Delta(\cdot)$ such that the $\beta$-fair set of rules, $\mathcal{W}(\beta; \Delta)$, is *itself* an ellipsoid, or when intersected with the Euclidean ball forms an ellipsoidal or polyhedral *feasible region*. To illustrate, see Figures 1a, 1b, and 1c.

**Property 3.1 (Feasible polyhedron (Fig. 1a))** $\beta, \Delta(\cdot)$ *are s.t. the feasible region of the fairness constrained problem (Eq. (2)) forms a polyhedron:* $\mathcal{B}(1) \cap \mathcal{W}(\beta; \Delta) = \{\mathbf{w} : \mathbf{w} \in \mathbb{R}^d, M\mathbf{w} \leq \beta\mathbf{1}\}$ *for some $M \in \mathbb{R}^{k\times d}$ and $\mathbf{1} \in \mathbb{R}^k$, where $k \in \mathbb{N}$, $\beta > 0$.*

**Property 3.2 ($\beta$-fair ellipsoid (Fig. 1b))** $\beta, \Delta(\cdot)$ *are s.t. the $\beta$-fair space is an ellipsoid:* $\mathcal{W}(\beta; \Delta) = \{\mathbf{w} : \mathbf{w} \in \mathbb{R}^d, \mathbf{w}^\top Q\mathbf{w} \leq \beta\}$, $Q \succ 0$, $\beta > 0$.

**Property 3.3 (Feasible ellipsoid (Fig. 1c))** $\beta, \Delta(\cdot)$ *are s.t. the feasible region for the fairness constrained problem (Eq. (2)) is an ellipsoid:* $\mathcal{B}(1) \cap \mathcal{W}(\beta; \Delta) = \{\mathbf{w} : \mathbf{w} \in \mathbb{R}^d, \mathbf{w}^\top Q\mathbf{w} \leq \beta\}$, $Q \succ 0$, $\beta > 0$.

To connect these properties and our example fairness constraints more clearly, note that the feasible region for the fairness constrained problem of Example 3.1 forms a polyhedron (i.e., Prop. 3.1 with $M = \Pi_D(A_1^{-1}C^\top\Pi_1 - A_2^{-1}C^\top\Pi_2)$), see Appendix A.2.3. Likewise, in Example 3.2, the set of $\beta$-fair rules form an ellipsoid (i.e., Prop. 3.2 with $Q = M^\top M$), see Appendix A.2.4.

In Table 1, we present our upper bounds on accuracy and welfare loss in $\beta$-fair SE as a function of the setting parameters and the discrepancy function. We leave their individual formal statements to Appendix A.2. Notice that all bounds depend on some transformation of the true policy, $\mathbf{w}^\star$. Fundamentally, this is because $\mathbf{w}^\star$ is unrestricted, whereas fairness-constrained optimization is carried out over a specific feasible region, $\mathcal{B}(1) \cap \mathcal{W}(\beta; \Delta)$. Crucially, we

---

[3]Accuracy is defined as negative squared error to discuss both objectives as a maximization.

first establish bounds under the minimal assumption that the feasible region is convex, without imposing additional geometric structure. We then derive additional bounds for discrepancy functions satisfying Properties 3.1–3.3, corresponding to polyhedral and ellipsoidal feasible regions. In all cases, a principal who has knowledge/estimates of the system parameters ($C$, $A_g$, $\Pi_g$, and $\mathbf{w}^\star$) can obtain an explicit bound on worst-case equilibrium loss. Note that several bounds depend *explicitly* on the discrepancy tolerance $\beta$, allowing the principal to directly quantify worst-case trade-offs when selecting different fairness levels. See numerical examples A.1 and A.2 for concrete illustration.

### 3.1. Discussion of Table 1 Bounds

Let us first start from the bounds of row 1 of Table 1, where there are no assumptions about the $\beta$-fair space. In this case, the bounds that we present are essentially tight. To see this, we can construct a tight example for the accuracy and SW loss by considering a $\beta$-fair space $\mathcal{W}(\beta; \Delta)$ that merely touches the boundary of $\mathcal{B}(1)$ such that its intersection with $\mathcal{B}(1)$ (and thus $\mathbf{w}_c^\star$) lies antidiametrically of the unconstrained optimal policy, $\mathbf{w}_u^\star$, (See Appendix Figure 5 for illustration). In this regime, the derived upper bounds are essentially attained (formally, Proposition A.2). However, using the additional geometric properties on the $\beta$-fair space and the feasible region, we can often obtain tighter bounds.

**SW loss.** Both the $\beta$-fair space ellipsoidal structure (Property 3.2) and the polyhedron feasible region (Property 3.1) enables us to improve the bounds because these geometries ensure that the constrained optimum $\mathbf{w}_c^\star$ remains aligned with the maximizing the SW direction inside the ball (see Appendix Lemma A.5). When the feasible region is an ellipsoid, (Property 3.3), the optimization problem is even more well-behaved and we can calculate *exactly* the SW loss.

**Accuracy loss.** For accuracy, if the feasible region has polyhedron structure (Property 3.1) we can get additional bounds by invoking Hoffman (1952)'s classical results. However, with general $\beta$-fair ellipsoidal structures (Property 3.2), we cannot give an additional bound. That said, for ellipsoidal

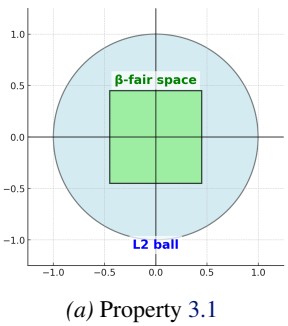

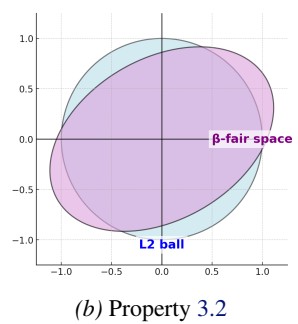

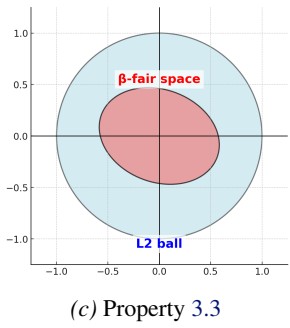

*(a)* Property 3.1          *(b)* Property 3.2          *(c)* Property 3.3

*Figure 1.* Examples of $\beta$-fair spaces in 2 dimensions that satisfy Properties 3.1 (feasible region polyhedral), 3.2 ($\beta$-fair space ellipsoidal), and/or 3.3 (feasible region ellipsoidal)

feasible regions (Property 3.3) we leverage this stronger assumption to provide one. Importantly though, this bound is not *always* better than the vanilla convexity-based bound; it may be larger depending on the parameters $\beta, Q, \mathbf{w}^\star$ (see Proposition A.6).

Finally, notice that for bounds depending on $\beta$, as $\beta$ *decreases* (i.e., the principal's tolerance of unfairness shrinks), the bound *increases*. This reflects a natural limitation: structural differences across groups can be mitigated only up to a point. As the feasible region contracts[4], the principal's optimization problem gets gradually more restrictive and eventually only the trivial solution ($\mathbf{w} = 0$) remains.

# 4. Nonconvex Fairness Constraints

What if the stakeholder wants a nonconvex desirability discrepancy function, $\Delta(\cdot)$? This may happen if the stakeholder compares desirability effort vectors, $\Pi_D \mathbf{x}_e^{(1)}(\mathbf{w})$ and $\Pi_D \mathbf{x}_e^{(2)}(\mathbf{w})$, asymmetrically. Consider the following:

**Example 4.1 (Asymmetric desirability fairness)**
$$\Delta(\mathbf{w}) := \|\Pi_D \mathbf{x}_e^{(g)}(\mathbf{w})\|_2^2 - \|\Pi_D \mathbf{x}_e^{(g')}(\mathbf{w})\|_2^2$$

In this case, $\Delta(\mathbf{w}) > 0$ when group $g$ is more desirably incentivized. So, upper bounding $\Delta$ by $\beta$ means that the principal's rules cannot better incentivize group $g$ than $g'$ by more than $\beta$, but the opposite is fine. This may be preferable if group $g$ is already externally privileged. Mathematically, this means that in general the $\beta$-fair space resulting from Example 4.1 is nonconvex; this holds under mild regularity conditions (formally in Appendix A.3.1). Informally, they are: (1) the projection matrix of the privileged group spans the feature space (2) the principal's tolerance is not too high.

Unfortunately, the principal's fairness-constrained problem (Eq. (2)) becomes less tractable in this case and computing the principal's equilibrium is difficult. For a broad family of generally nonconvex $\beta$-fair spaces, we replace the noncon-

vex constraint with a convex restriction, ensuring tractability while the desired $\beta$-fairness is still achieved. Then, we provide upper bounds on the optimality loss—relative to the nonconvex $\beta$-fair SE equilibrium—that the principal would incur by deploying the *restricted* $\beta$-fair policy.

**Definition 4.1 ($\mathcal{F}$, a class of nonconvex $\beta$-fair spaces)**
$\mathcal{W}(\beta; \Delta) \in \mathcal{F}$ *if the following is true for some* $Q \in \mathbb{R}^{d \times d}$ *s.t.* $Q \succ 0$: **(1)** $\Delta(\mathbf{w}) = \mathbf{w}^\top Q \mathbf{w} - f(\mathbf{w})$ *and* $\beta \leq \lambda_d(Q)$, *and* **(2)** $f : \mathbb{R}^d \to \mathbb{R}$ *is s.t.* $f(\mathbf{w}) \geq 0 \quad \forall \mathbf{w} \in \mathbb{R}^d$, $f(\mathbf{0}) = 0$, *and* $L$-*Lipschitz on* $\mathbf{w} \in \mathcal{W}(\beta; \Delta)$

As illustrated in Figure 2, the structure of $\mathcal{F}$ guarantees the existence of a simple, nonempty and $\beta$-fair ellipsoidal restriction of $\mathcal{W}(\beta; \Delta)$, which lies within the feasible region of the principal's fairness constrained problem. In Appendix A.3.2, we prove formally that ellipsoid $\mathcal{E}(\beta)$ where $\mathcal{E}(\beta) := \{\mathbf{w} : \mathbf{w} \in \mathbb{R}^d, \mathbf{w}^\top Q \mathbf{w} \leq \beta\}$ is such a restriction. On $\mathcal{E}(\beta)$ we can directly invoke Table 1's Property 3.3 optimality loss bounds. Thus, given the discrepancy function satisfies Definition 4.1, for any tolerance of reasonable magnitude, the principal has an estimate of worst-case optimality loss, $\texttt{OBJ}(\mathbf{w}_c) - \texttt{OBJ}(\mathbf{w}_c)$, in equilibrium.

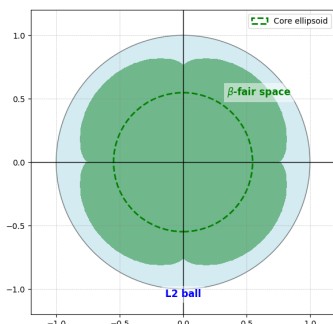

*Figure 2.* A nonconvex $\beta$-fair space in the form of Definition 4.1 $\Delta(\mathbf{w}) := \mathbf{w}^\top \mathbf{w} - (.3\sqrt{|\mathbf{w}_1|} + .3\sqrt{|\mathbf{w}_2|})$ and $\beta = .3$

**How tight is the convex restriction?** While the convex restriction enables tractable computation and optimality loss

---

[4]Notice $\mathbf{0} \in \mathcal{B}(1) \cap \mathcal{W}(\beta; \Delta)$ for Properties 3.1, 3.2, and 3.3

bounds, how much is sacrificed for this tractability? Formally, if $\mathbf{w}_{res}^{\star}$ is the optimal *convex-restricted* policy, how big is $|\text{OBJ}(\mathbf{w}_c) - \text{OBJ}(\mathbf{w}_{res})|$? Notably, the restriction does not sacrifice any required fairness, but some optimality is lost. Formally in Proposition A.13, for both the social welfare and accuracy objectives, we present upper bounds of the form $|\text{OBJ}(\mathbf{w}_c^{\star}) - \text{OBJ}(\mathbf{w}_{res}^{\star})| \leq \Phi_{\text{OBJ}}(L, D, \mathbf{w}^{\star}, \beta, Q)$ where $\mathbf{w}_{res}^{\star}$ is the optimal policy restricted according to the convex $\mathcal{E}(\beta)$. $\Phi_{\text{OBJ}}$, the optimality loss due to the restriction under a particular objective, depends on the parameters $L$ and $D$, where $L$ is the Lipschitz constant of the $f(\mathbf{w})$ from Definition 4.1 over the $\beta$-fair space, $\mathcal{W}(\beta; \Delta)$, and $D$ is a measure of that $\beta$-fair space's diameter. To interpret these bounds, recall Figure 2: $\mathcal{W}(\beta; \Delta)$ consists of a core ellipsoid together with an additional region induced by $f(\mathbf{w})$. Since $f(\mathbf{w}) = 0$ at least once over the $\beta$-fair space, the Lipschitz constant, $L$, controls how large this additional region can be. Thus, $L$ is a proxy for how much "extra" space $\mathcal{W}(\beta; \Delta)$ gets around its core ellipsoid. As $L$ decreases, our bounds on optimality loss due to restriction also decrease.

# 5. Experimental Evaluation

In Sections 3 and 4, we derived upper bounds on the trade-offs between accuracy, social welfare, and desirable-effort fairness. Since they are worst-case, these bounds do not directly reflect the *actual* objective value (accuracy /social-welfare) attained in practice. Real-world data allows us to complement the theory in two ways: first, by estimating the instance-dependent quantity $\mathbf{w}^{\star}$ and therefore the realized losses; and second, by varying the group and cost configurations in order to study how different patterns of group heterogeneity affect the equilibrium outcome. Using these observations we conduct two sets of experiments. On the ADULT dataset, we compute fair equilibria under different demographic splits, cost structures, and fairness tolerances, and report the resulting accuracy and social-welfare values. On the Taiwan dataset, we use the same setup to compare realized losses against the theoretical upper bounds from Table 1.[5]

## 5.1. Empirical evaluation of the realized losses on the ADULT dataset

**Setup.** We conduct experiments on the ADULT dataset and analyze how the same constraint impacts the principal's SE optimal value under different patterns of group disparity. Specifically, we study 3 population partitions based on age,

---

[5]For ADULT, the plots report absolute objective values attained by the fair equilibrium. For Taiwan, the accuracy plots are equivalent, up to sign, to realized accuracy losses because the unconstrained accuracy optimum equals zero. Welfare losses, however, are computed relative to the unconstrained welfare optimum, since the welfare benchmark is nonzero.

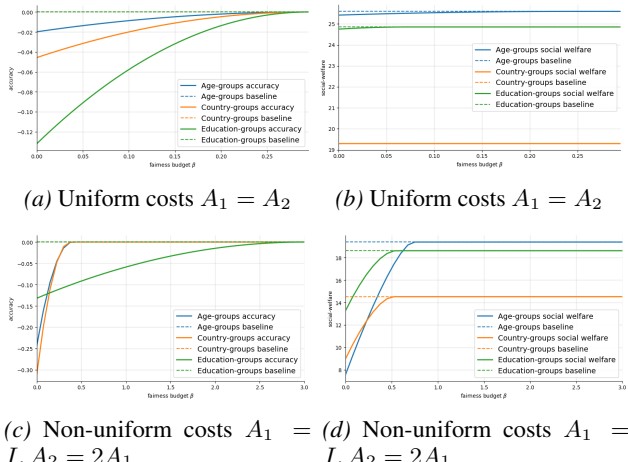

*(a)* Uniform costs $A_1 = A_2$    *(b)* Uniform costs $A_1 = A_2$

*(c)* Non-uniform costs $A_1 = I, A_2 = 2A_1$    *(d)* Non-uniform costs $A_1 = I, A_2 = 2A_1$

*Figure 3.* Optimal value in $\beta$-fair equilibria for $\ell_1$ fairness constraints. Desirable features: occupation, workclass, education. Solid curves: $\ell_1$-fairness-constrained outcomes; dashed lines: unconstrained optimum. Colors denote groups (blue: age, orange: country, green: education). Fig. (a), (c): ACC. Fig. (b), (d): SW.

country, and education level. Each partition induces a corresponding pair of projection matrices, $\Pi_1, \Pi_2$. To obtain our causal graph, we follow prior work on the Adult dataset (Kugelgen et al., 2022; Nabi & Shpitser, 2018; Chiappa, 2019) and instantiate an 8-node acyclic causal graph with nodes {sex, age, western, married, edu-num, workclass, occupation, hours}. We compute the $\beta$-fair equilibrium for various cost matrices, $A_1, A_2$ and desirable feature sets $\Pi_D$. Our main finding is that disparities aligned with the desirable feature space induce more persistent optimality loss, while misaligned disparities recover performance rapidly as $\beta$ increases. All details of our experimental setup can be found in Appendix A.4.1.

**Results.** Figure 3 summarizes our experimental results. Specifically, it plots accuracy (left) and social welfare (right) as the fairness budget $\beta$ varies, using the $\ell_1$-fairness constraint as the discrepancy function; additional experiments for the $\ell_2$ discrepancy function and random cost matrices can be found in Appendix A.4.2. For accuracy, all partitions share the same unconstrained optimum of 0 (perfect accuracy). For social welfare, the unconstrained optima differ across partitions, as the welfare coefficients (Lemma A.3) depend on each group's matrices.

In general, as the tolerance, $\beta$, increases, the fairness constraint relaxes and the optimal value (e.g., accuracy, social welfare) improves. In Figure 3a, the *Education* split, which separates agents into well and less educated groups, is consistently the most constrained, its curve starts at the lower point and improves most slowly. In this case, the desirable attributes (education, workclass, occupation) are very aligned with this group disparity, so the discrepancy function, $\Delta$, penalizes movements along the most predictive directions. Figure 3b shows that social welfare is rel-

atively insensitive to disparities in $\Pi_g$: for all partitions, the constrained optimum approaches the unconstrained value rapidly. Next, we introduce group disparity by cost ($A_1 \neq A_2$). In Figure 3c, we see that optimality loss for the *Education* split retains the same shape, though the maximum $\beta$ until optimality recovery has increased. Meanwhile, both *Country* and *Age* have significant optimality loss at the tightest fairness constraints, though recover 0 accuracy much faster than *Education*. This behavior is expected; the *Education* split is strongly correlated with desirable characteristics and thus fairness constraints continue to have long lasting effects even at high $\beta$. With increased cost disparities, optimality in $\beta$-fair equilibria for *Country* and *Age* suffers at the harshest constraints, but as they are not closely aligned with desirability, it does not impact looser constraints. Figure 3d exhibits a similar pattern. As SW is not so sensitive to disparities in $\Pi_g$ (Fig 3b), creating cost disparities on all groups invokes some optimality loss for all groups but with subtle differences.

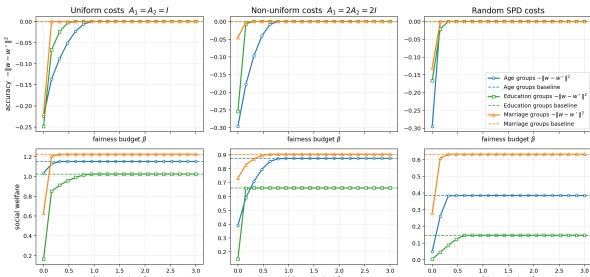

*Figure 4.* Taiwan dataset: realized objective gaps and theoretical bounds under $\ell_1$ fairness constraints. Top row: signed accuracy gap $L_{\mathrm{acc}}(\beta) = f(\mathbf{w}_c^\star) - f(\mathbf{w}_u^\star) \leq 0$, following the maximization convention; values closer to 0 indicate smaller loss. Bottom row: social-welfare loss $L_{\mathrm{SW}}(\beta) = h(\mathbf{w}_u^\star) - h(\mathbf{w}_c^\star) \geq 0$. Columns show uniform costs, non-uniform costs, and random SPD costs. Solid curves are realized losses/gaps; dashed curves are the corresponding bounds from Table 1. Colors denote the population partition.

### 5.2. Empirical Tightness of the Bounds on the `Taiwan` Dataset

For the Taiwan dataset, we additionally compare realized losses with the theoretical bounds in Table 1. **Setup.** We use the `Taiwan` credit-default dataset, which contains financial and demographic information about clients and is commonly used to predict whether a client is likely to default on repayment. We study three population partitions based on age, education, and marital status. Each partition induces a corresponding pair of projection matrices, $\Pi_1, \Pi_2$. Similarly to the previous experiment, we compute the $\beta$-fair equilibrium for various cost matrices, $A_1, A_2$ and desirable feature sets $\Pi_D$. A full description of the setup can be found in Appendix A.5.

**Results.** Figure 4 compares the realized losses with the theoretical upper bounds from Table 1. The main observation is that bounds which adapt to the geometry of the $\beta$-fair feasible region are substantially tighter. Specifically, the geometry-dependent accuracy-loss bounds decrease with $\beta$ and closely follow the empirical losses as the fairness constraint is relaxed. By contrast, bounds that do not incorporate such instance-specific information, such as the more general social-welfare bounds, remain essentially constant as $\beta$ varies. These bounds are therefore much more conservative: they are potentially tight only in very restrictive, near-worst-case regimes, while the realized losses on the empirical instance often decay rapidly. A more in depth discussion of results can be found in Appendix A.5.2.

## 6. Discussion

In this work, we provide a formal framework for incentivizing fair effort among heterogeneous groups, characterizing

the inherent trade-off between fairness constraints and the principal's objective (e.g., accuracy or welfare).

**Practical value for decision-makers.** Our bounds enable principals to make informed *ex-ante* decisions: given estimates of cost matrices and group structure, decision-makers can evaluate whether a desired fairness tolerance $\beta$ is achievable at acceptable performance loss *before* deployment. This is particularly valuable in regulated domains where fairness commitments may be externally mandated by stakeholders such as regulatory bodies, oversight committees, or public interest requirements. In practice, structural properties of group disparities may yield a trivial feasible set for a given choice of tolerance: upon detecting this, the principal can relax $\beta$ until a reasonable fairness-performance trade-off is achieved. In that case, Table 1 can be leveraged to select alternative discrepancy functions that better align with the specific structure of group disparities.

**Future directions.** There are several future work avenues. First, while we assume agents know the contribution matrix $C$, one may study settings where agents must *learn* causal relationships through experimentation or peer observation, or where groups hold different, possibly misspecified, causal beliefs. Second, for nonconvex settings, investigating tighter polyhedral outer approximations or problem-specific structure (e.g., sparsity in $C$, low-rank cost matrices) may reduce the gaps between restricted and true solutions. Third, characterizing how (potentially disparate) estimation errors in $A_g, \Pi_g$, and $C$ or agent's peer learning, $\mathbf{w}_{\mathrm{EST}}(g)$, affect our bounds would help practitioners understand data requirements for reliable fairness guarantees. Finally, richer fairness constraints (e.g., intersectional considerations across multiple attributes or absolute cost caps for disadvantaged groups) could be included to address more complex needs.

## Impact Statement

While our analysis is primarily theoretical, it is motivated by the deployment of ML-based classifiers in societal decision-making systems. Strategic classification highlights that data are not static: agents respond to the incentives induced by the classifier and may strategically modify their features. Since agents form beliefs about the decision rule, the designer's choices shape not only predictions but also the data distribution itself. Accounting for this interaction is therefore essential for reliable and responsible deployment.

Our contribution builds on prior work by studying not only how to design rules that incentivize desirable behavior, but also how informational disparities across groups affect equilibrium responses and downstream outcomes. In particular, we study how to choose policies such that they do not disproportionately disadvantage groups with limited access to information, motivating fairness constraints that explicitly account for such discrepancies.

Our work has several limitations. First, we assume the principal has unlimited capacity to assign positive classifications, whereas many applications involve explicit capacity constraints (e.g., admissions or hiring). Incorporating such constraints is an important direction for future work. Second, our modeling of "desirable" features is not the only reasonable choice: in degenerate cases, a principal could select target features that are equally inaccessible to all groups, making fairness constraints trivially satisfied. Our framework therefore implicitly assumes a good-faith principal whose objective is aligned with social welfare, rather than one that exploits modeling choices to satisfy fairness constraints without meaningful behavioral improvement.

## Acknowledgements

Research reported in this paper was partially supported by an Amazon Research Award Fall 2023. Any opinions, findings, and conclusions or recommendations expressed in this material are those of the author(s) and do not reflect the views of Amazon.

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

# A. Supplemental Material

## A.1. Supplemental material for Section 2

*Table 2.* Notation Table

| Symbol | Meaning |
|:---:|:---:|
| $d$ | dimension of features |
| $C$ | Contribution matrix |
| $A_g$ | group $g$ cost matrix |
| $\Pi_g$ | group $g$ projection matrix |
| $\Pi_D$ | diagonal desirability score matrix |
| $\text{des}(i)$ | feature $i$ desirability score |
| $g$ | group |
| $\mathcal{D}_g$ | group $g$ distribution of features |
| $\mathcal{S}_g$ | group $g$ feature subspace |
| $\mathbf{x}$ | initial feature |
| $\mathbf{x}'$ | altered feature |
| $\mathbf{x}_e$ | exogenous effort |
| $\mathbf{w}$ | principal's rule |
| $\mathbf{w}^\star$ | ground truth rule |
| $\mathbf{w}_u^\star$ | ground truth rule projected onto the Euclidean ball (solution to Problem 3) |
| $\mathbf{w}_c^\star$ | principal's $\beta$-fair desirable rule (solution to Problem 2) |
| $\text{Score}(\mathbf{x}', g)$ | group $g$ agent's estimated score |
| $\text{Cost}(\mathbf{x}_e; g)$ | agent cost for exerting effort $\mathbf{x}_e$ |
| $\Delta(\mathbf{w})$ | discrepancy in desirable effort incentivized by $\mathbf{w}$ |
| $\mathcal{G}$ | causal graph |
| $\mathcal{A}$ | set of edges |
| $\omega$ | edge weights |
| $\mathcal{P}_{ij}$ | set of all direct paths from node $i$ to node $j$ |
| $\omega(p)$ | sum of weights on path $p \in \mathcal{P}_{ij}$ |
| $\beta$ | principal's discrepancy tolerance |
| $\mathcal{U}(\mathbf{x}, \mathbf{x}', g)$ | group $g$ agent's utility as a function alteration |
| $\mathcal{B}(1)$ | euclidean ball with radius 1 |
| $\text{ACC}(\mathbf{w}, \mathbf{w}^\star)$ | accuracy of rule $\mathbf{w}$ |
| $\text{SW}(\mathbf{w}, \mathbf{w}^\star)$ | social welfare of rule $\mathbf{w}$ |
| $\mathcal{W}(\beta; \Delta)$ | set of $\beta$-fair rules |
| $\Pi_D \mathbf{x}_e^{(g)}(\mathbf{w})$ | desirability-weighted effort vector for group $g$ agent |
| $H(M)$ | Hoffman constant of matrix $M$ |
| $\mathcal{E}(\tilde{\beta})$ | ellipsoidal envelope of a nonconvex $\mathcal{W}(\beta; \Delta) \in \mathcal{F}$ |

A.1.1. SUPPLEMENTAL MATERIAL FOR SECTION 2

**Hoffman constant.** We use $H(M)$ to denote the Hoffman constant of matrix $M \in \mathbb{R}^{k \times d}$, i.e., a constant such that $\forall \mathbf{b} \in \mathcal{M} + \mathbb{R}_{\geq 0}^k$ and $\forall \mathbf{z} \in \mathbb{R}^d$ it is true that

$$\text{Dist}(z, P_A(\mathbf{b})) \leq H(M) \|(Mz - \mathbf{b})_+\|_2$$

where $\text{Dist}(z, P_A(\mathbf{b})) := \min\{\|z - x\|_2 : \mathbf{x} \in P_A(\mathbf{b})\}$. $\mathcal{M} := \{M\mathbf{w} : \mathbf{w} \in \mathbb{R}^d\}$ and $P_A(\mathbf{b}) := \{\mathbf{w} \in \mathbb{R}^d : M\mathbf{w} \leq \mathbf{b}\}$.

In particular, this is a Hoffman constant for $p = 2$ (i.e., $\text{Dist}$ is the $l2$ norm). This is an equivalent definition to the one used by (Peña et al., 2021).

A.1.2. SUPPLEMENTAL MATERIAL FOR SECTION 2.2

**Lemma A.1 (Contribution matrix of a DAG is invertible and kernel zero)** *Let $C$ be the contribution matrix of a DAG. Then $\ker(C) = \emptyset$. Equivalently, $C$ is invertible.*

**Proof of Lemma A.1.** Recall that for some exogenous effort $\mathbf{x}_e \in \mathbb{R}^d$, we have the post-causality effort $\mathbf{x} := C\mathbf{x}_e$. We shall prove $\ker(C) = \emptyset$ by contradiction. Suppose $\ker(C) \neq \emptyset$, then it must be the case that there exists $\mathbf{x}_e$ where $\mathbf{x}_e \neq \mathbf{0}$, s.t. $C\mathbf{x}_e = \mathbf{0}$ and thus this exogenous effort "cancels itself out". Let $\mathbf{x}_e$ be a nonzero vector in $\ker(C)$ and define $\mathcal{I} := \{i \in [d] : x_{e_i} \neq 0\}$. Because $C\mathbf{x}_e = \mathbf{0}$, $\forall i \in \mathcal{I}$, node $i$ in the causal graph must have at least one in-degree from some $j \in \mathcal{I}$ or else there is no way that $\mathbf{c_i}^\top \mathbf{x}_e = 0$. To see this, recall that by construction, a row $\mathbf{c_i}$, of $C$ is made up of paths *into* node $i$ and $C_{i,i} = 1$. Consider the subgraph, $\tilde{\mathcal{G}}$, represented by the collection of nodes in $\mathcal{I}$ and the edges between them. Each of these nodes have at least one in-degree from another in the subgraph. Thus, no node in the finite directed subgraph has $0$ in-degree. Clearly this means there must be cycle because if we consider traversing the graph from any vertex, we must eventually repeat a vertex as they are finite and all have an in-degree. However, this poses a contradiction to our assumption that $C$ comes from a DAG. Therefore, it must be the case that $\ker(C) = \emptyset$. □

**Proof of Proposition 2.1.** From Lemma 3.1 of (Bechavod et al., 2022), agents' estimate of $\mathbf{w}_{est}$ can be solved in closed form as a function of $\Pi_g, \mathbf{w}$: $\mathbf{w}_{est}(g) = \Pi_g \mathbf{w}$. Thus:

$$\mathcal{U}(\mathbf{x}, \mathbf{x}'; g) := \langle \Pi_g \mathbf{w}, \mathbf{x}' \rangle - \frac{1}{2} \|\sqrt{A_g}(\mathbf{x}_e)\|^2$$

$$= \langle \Pi_g \mathbf{w}, \mathbf{x} + C\mathbf{x}_e \rangle - \frac{1}{2} \|\sqrt{A_g}(\mathbf{x}_e)\|^2$$

This function should be concave (sum of 3 concave functions: a constant plus a linear term minus a norm) and hence:

$$\nabla \mathcal{U}(\mathbf{x}, \mathbf{x}'; g) = C^\top \Pi_g \mathbf{w} - A_g \mathbf{x}_e = 0 \iff$$

$$\mathbf{x}_e = A_g^{-1} C^\top \Pi_g \mathbf{w}$$

Therefore the best-response is:

$$\mathbf{x}'(\mathbf{x}; g) = \mathbf{x} + C A_g^{-1} C^\top \Pi_g \mathbf{w} \tag{4}$$

□

A.1.3. SUPPLEMENTAL MATERIAL FOR SECTION 2.3

**Lemma A.2 (Equivalent Accuracy Objective)** *An accuracy-maximizing principal can solve either Problem 2 or 3 using the following objective to find the optimal $\mathbf{w}_{\text{ACC}}$ policy in equilibrium*

$$\max_{\mathbf{w} \in \mathbb{R}^d} \quad - \|\mathbf{w}^\star - \mathbf{w}\|_2^2 \tag{5}$$

**Proof of Lemma A.2.** Using the solution from 2.1

$$-\text{ACC} = \sum_{g \in [2]} \mathbb{E}_{\mathbf{x} \sim \mathcal{D}_g} \left[ \left( \mathbf{w}^{\star\top} \hat{\mathbf{x}}(\mathbf{x}; g) - \mathbf{w}^\top \hat{\mathbf{x}}(\mathbf{x}; g) \right)^2 \right]$$

$$= \sum_{g \in [2]} \mathbb{E}_{\mathbf{x} \sim \mathcal{D}_g} \left[ (\mathbf{w}^{\star\top} \hat{\mathbf{x}}(\mathbf{x}; g))^2 + (\mathbf{w}^\top \hat{\mathbf{x}}(\mathbf{x}; g))^2 - 2(\mathbf{w}^{\star\top} \hat{\mathbf{x}}(\mathbf{x}; g))(\mathbf{w}^\top \hat{\mathbf{x}}(\mathbf{x}; g)) \right]$$

$$= \langle \mathbf{w}^\star, \mathbf{w}^\star \rangle + \langle \mathbf{w}, \mathbf{w} \rangle - 2 \langle \mathbf{w}^\star, \mathbf{w} \rangle$$

$$= \langle \mathbf{w}^\star - \mathbf{w}, \mathbf{w}^\star - \mathbf{w} \rangle$$

$$= \|\mathbf{w}^\star - \mathbf{w}\|_2^2$$

□

**Lemma A.3 (Equivalent SW Objective)** *A social-welfare-maximizing principal can solve either Problem 2 or 3 using the following objective to find the optimal $\mathbf{w}_{\text{SW}}$ policy in equilibrium*

$$\max_{\mathbf{w} \in \mathbb{R}^d} \quad \langle (C A_1^{-1} C^\top \Pi_1 + C A_2^{-1} C^\top \Pi_2)^\top \mathbf{w}^\star, \mathbf{w} \rangle \tag{6}$$

**Proof of Lemma A.3.** Recall that $\text{SW} := \sum_{i \in [2]} \mathbb{E}_{x \sim \mathcal{D}_g}[\langle \hat{\mathbf{x}}(\mathbf{x}; g), \mathbf{w}^\star \rangle]$. Using the $\hat{\mathbf{x}}$ solution from 2.1, we see that this is equivalent to the following

$$
\begin{aligned}
\text{SW} &= \sum_{g \in [2]} \mathbb{E}_{x \sim \mathcal{D}_g}[\langle \hat{\mathbf{x}}(\mathbf{x}; g), \mathbf{w}^\star \rangle] \\
&= \mathbb{E}_{\mathbf{x} \sim \mathcal{D}_1}[\langle \mathbf{x} + \Delta_1(\mathbf{w}), \mathbf{w}^\star \rangle] + \mathbb{E}_{\mathbf{x} \sim \mathcal{D}_2}[\langle \mathbf{x} + \Delta_2(\mathbf{w}), \mathbf{w}^\star \rangle] \\
&= \langle \Delta_1(\mathbf{w}), \mathbf{w}^\star \rangle + \langle \Delta_2(\mathbf{w}), \mathbf{w}^\star \rangle + \mathbb{E}_{\mathbf{x} \sim \mathcal{D}_1}[\langle \mathbf{x}, \mathbf{w}^\star \rangle] + \mathbb{E}_{\mathbf{x} \sim \mathcal{D}_2}[\langle \mathbf{x}, \mathbf{w}^\star \rangle] \\
&= \langle CA_1^{-1}C^\top \Pi_1 \mathbf{w}, \mathbf{w}^\star \rangle + \langle CA_2^{-1}C^\top \Pi_2 \mathbf{w}, \mathbf{w}^\star \rangle + \mathbb{E}_{\mathbf{x} \sim \mathcal{D}_1}[\langle \mathbf{x}, \mathbf{w}^\star \rangle] + \mathbb{E}_{\mathbf{x} \sim \mathcal{D}_2}[\langle \mathbf{x}, \mathbf{w}^\star \rangle] \\
&= \langle \mathbf{w}^\star, CA_1^{-1}C^\top \Pi_1 \mathbf{w} \rangle + \langle \mathbf{w}^\star, CA_2^{-1}C^\top \Pi_2 \mathbf{w} \rangle + \mathbb{E}_{\mathbf{x} \sim \mathcal{D}_1}[\langle \mathbf{x}, \mathbf{w}^\star \rangle] + \mathbb{E}_{\mathbf{x} \sim \mathcal{D}_2}[\langle \mathbf{x}, \mathbf{w}^\star \rangle] \\
&= \mathbf{w}^{\star\top} CA_1^{-1}C^\top \Pi_1 \mathbf{w} + \mathbf{w}^{\star\top} CA_2^{-1}C^\top \Pi_2 \mathbf{w} + \mathbb{E}_{\mathbf{x} \sim \mathcal{D}_1}[\langle \mathbf{x}, \mathbf{w}^\star \rangle] + \mathbb{E}_{\mathbf{x} \sim \mathcal{D}_2}[\langle \mathbf{x}, \mathbf{w}^\star \rangle] \\
&= \langle (CA_1^{-1}C^\top \Pi_1)^\top \mathbf{w}^\star, \mathbf{w} \rangle + \langle (CA_2^{-1}C^\top \Pi_2)^\top \mathbf{w}^\star, \mathbf{w} \rangle + \mathbb{E}_{\mathbf{x} \sim \mathcal{D}_1}[\langle \mathbf{x}, \mathbf{w}^\star \rangle] + \mathbb{E}_{\mathbf{x} \sim \mathcal{D}_2}[\langle \mathbf{x}, \mathbf{w}^\star \rangle] \\
&= \langle (CA_1^{-1}C^\top \Pi_1 + CA_2^{-1}C^\top \Pi_2)^\top \mathbf{w}^\star, \mathbf{w} \rangle + \mathbb{E}_{\mathbf{x} \sim \mathcal{D}_1}[\langle \mathbf{x}, \mathbf{w}^\star \rangle] + \mathbb{E}_{\mathbf{x} \sim \mathcal{D}_2}[\langle \mathbf{x}, \mathbf{w}^\star \rangle]
\end{aligned}
$$

The two expectation terms are constants with respect to $\mathbf{w}$, so they can be ignored when finding an optimal solution. Since the learner wants to maximize the linear objective, this is equivalent to minimizing the negative. $\qquad \square$

## A.2. Supplemental material for Section 3

### A.2.1. SUPPLEMENTAL MATERIAL OF TABLE 1

Recall The folllowing table:

*Table 3.* Restated table of upper bounds on optimality loss in $\beta$-fair SE under Properties 3.1–3.3.

| $\beta$-**Fair Space** | **Feasible Region** | **Accuracy Loss** | **SW Loss** |
|---|---|---|---|
| no assumptions | convex | $A$ | $B$ |
| no assumptions | polyhedron (Property 3.1) | $C$ | $D$ |
| ellipsoidal (Property 3.2) | no assumptions | $A$ | $D$ |
| no assumptions | ellipsoidal (Property 3.3) | $E$ | $F$ |

We will now derive the bounds that correspond to each combination in the table.

**CONVEXITY-BASED BOUNDS (A AND B).**

**Lemma A.4 (Sharp accuracy gap on the unit ball)** *Let* $f(\mathbf{w}) = \|\mathbf{w} - \mathbf{w}^\star\|_2^2$ *and let*

$$
\mathbf{w}_u^\star = \Pi_{\mathcal{B}(1)}(\mathbf{w}^\star) \quad \text{where} \quad \mathcal{B}(1) = \{\mathbf{w} : \|\mathbf{w}\|_2 \leq 1\}.
$$

*Then*

$$
\sup_{\mathbf{w} \in \mathcal{B}(1)} \left( f(\mathbf{w}) - f(\mathbf{w}_u^\star) \right) = \begin{cases} 4, & \text{if } \|\mathbf{w}^\star\| \leq 1, \\ 4\|\mathbf{w}^\star\|, & \text{if } \|\mathbf{w}^\star\| > 1. \end{cases}
$$

*In particular, for all* $\mathbf{w} \in \mathcal{B}(1)$,
$$
0 \leq f(\mathbf{w}) - f(\mathbf{w}_u^\star) \leq 4 \max\{1, \|\mathbf{w}^\star\|\},
$$
*and when* $\|\mathbf{w}^\star\| > 1$ *the upper bound* $4\|\mathbf{w}^\star\|$ *is attained at* $\mathbf{w} = -\mathbf{w}_u^\star$.

**Proof.** Since $\mathbf{w}_u^\star$ is the Euclidean projection of $\mathbf{w}^\star$ onto $\mathcal{B}(1)$, it minimizes $f(\mathbf{w})$ over $\mathcal{B}(1)$, and therefore $f(\mathbf{w}) - f(\mathbf{w}_u^\star) \geq 0$ for all $\mathbf{w} \in \mathcal{B}(1)$. We now compute the supremum explicitly.

**Case 1:** $\|\mathbf{w}^\star\| \leq 1$**.** In this case, $\mathbf{w}_u^\star = \mathbf{w}^\star$. Hence

$$
\sup_{\mathbf{w} \in \mathcal{B}(1)} \left( f(\mathbf{w}) - f(\mathbf{w}_u^\star) \right) = \sup_{\mathbf{w} \in \mathcal{B}(1)} \|\mathbf{w} - \mathbf{w}^\star\|_2^2.
$$

For any $\mathbf{w} \in \mathcal{B}(1)$,
$$\|\mathbf{w} - \mathbf{w}^\star\| \le \|\mathbf{w}\| + \|\mathbf{w}^\star\| \le 1 + \|\mathbf{w}^\star\| \le 2,$$

which implies $\|\mathbf{w} - \mathbf{w}^\star\|^2 \le 4$. Equality is attained at $\mathbf{w} = -\mathbf{w}^\star/\|\mathbf{w}^\star\|$ when $\mathbf{w}^\star \ne 0$ (and at any $\|\mathbf{w}\| = 1$ when $\mathbf{w}^\star = 0$). Therefore the supremum equals 4.

**Case 2: $\|\mathbf{w}^\star\| > 1$.** Write $\mathbf{w}^\star = r\mathbf{u}$ where $r = \|\mathbf{w}^\star\| > 1$ and $\|\mathbf{u}\| = 1$. Then $\mathbf{w}_u^\star = \mathbf{u}$. For any $\mathbf{w} \in \mathcal{B}(1)$,

$$\begin{aligned}
f(\mathbf{w}) - f(\mathbf{w}_u^\star) &= \|\mathbf{w} - r\mathbf{u}\|_2^2 - \|\mathbf{u} - r\mathbf{u}\|_2^2 \\
&= \left(\|\mathbf{w}\|^2 - 2r\langle \mathbf{w}, \mathbf{u}\rangle + r^2\right) - (r-1)^2 \\
&= \|\mathbf{w}\|^2 - 2r\langle \mathbf{w}, u\rangle + 2r - 1.
\end{aligned}$$

Since $\|\mathbf{u}\| = 1$, any vector $\mathbf{w} \in \mathbb{R}^d$ admits the orthogonal decomposition

$$\mathbf{w} = \alpha\,\mathbf{u} + \mathbf{z}, \qquad \alpha := \langle \mathbf{w}, \mathbf{u}\rangle, \qquad \mathbf{z} := \mathbf{w} - \alpha\mathbf{u},$$

where $\langle \mathbf{u}, \mathbf{z}\rangle = 0$. By orthogonality,
$$\|\mathbf{w}\|^2 = \alpha^2 + \|\mathbf{z}\|^2.$$

The constraint $\mathbf{w} \in \mathcal{B}(1)$ is therefore equivalent to $\alpha^2 + \|\mathbf{z}\|^2 \le 1$. Substituting into the expression above yields

$$f(\mathbf{w}) - f(\mathbf{w}_u^\star) = \alpha^2 + \|\mathbf{z}\|^2 - 2r\alpha + 2r - 1.$$

For fixed $\alpha$, this expression is increasing in $\|\mathbf{z}\|^2$, and hence is maximized when $\|\mathbf{z}\|^2 = 1 - \alpha^2$, i.e., when $\|\mathbf{w}\| = 1$. This reduces the problem to the one-dimensional maximization

$$\sup_{\alpha \in [-1,1]} \left(1 - 2r\alpha + 2r - 1\right) = \sup_{\alpha \in [-1,1]} 2r(1-\alpha).$$

The maximum is attained at $\alpha = -1$, giving

$$\sup_{\mathbf{w} \in \mathcal{B}(1)} \left(f(\mathbf{w}) - f(\mathbf{w}_u^\star)\right) = 2r(1-(-1)) = 4r = 4\|\mathbf{w}^\star\|.$$

The maximizer is $\mathbf{w} = -u = -\mathbf{w}_u^\star$, and substituting back gives

$$f(-\mathbf{w}_u^\star) - f(\mathbf{w}_u^\star) = (r+1)^2 - (r-1)^2 = 4r.$$

Combining the two cases proves the claimed formula and the uniform bound $f(\mathbf{w}) - f(\mathbf{w}_u^\star) \le 4\max\{1, \|\mathbf{w}^\star\|\}$. $\qquad\square$

**Proposition A.1 (Bound A: Convexity-based Accuracy loss)** *Assuming convexity of the fairness-constrained feasible region, for the Accuracy objective, the optimality loss between the unconstrained and fairness-constrained equilibrium is upper-bounded:*
$$|f(\mathbf{w}_c^\star) - f(\mathbf{w}_u^\star)| \le 4\max\{1, \|\mathbf{w}^\star\|\} \tag{7}$$

**Proof.** This follows directly from Lemma A.4 and Lemma A.2. notice that if for all $\mathbf{w} \in \mathcal{B}(1)$,

$$0 \le f(\mathbf{w}) - f(\mathbf{w}_u^\star) \le 4\max\{1, \|\mathbf{w}^\star\|\},$$

then it holds for $\mathbf{w}_c^\star$, which must the euclidean ball because it is the optimal point to equation 2, whose feasible region is a subset of the euclidean ball. $\qquad\square$

**Proposition A.2 (Bound B: Convexity-based SW loss)** *Assuming convexity of the fairness-constrained feasible region, for the SW objective, the optimality loss between the unconstrained and fairness-constrained equilibrium is upper-bounded:*
$$|f(\mathbf{w}_c^\star) - f(\mathbf{w}_u^\star)| \le 2\|\tilde{\mathbf{w}}\| \tag{8}$$

*(where $\tilde{\mathbf{w}} = (CA_1^{-1}C^\top\Pi_1 + CA_2^{-1}C^\top\Pi_2)^\top\mathbf{w}^\star)$) and this social welfare bound is tight.*

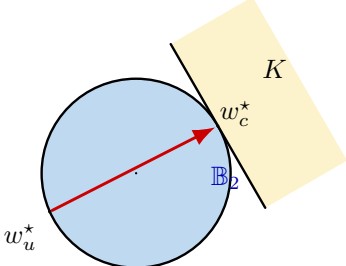

*Figure 5.* A tight example on optimality loss: the $\beta$-fair space is a convex set K touching the boundary of the ball

**Proof.** Our bounds are almost tight in the sense that there is a worst case example that reaches them - at least in the limit-. Starting with no assumptions about the $\beta$-fair space (first row of 1) we can construct a tight example by choosing the $\beta$-fair space $\mathcal{W}$ so that $\mathcal{W}$ touches the boundary of $\mathcal{B}(1)$ and the optimum $\mathbf{w}_c^\star$ lies antidiametrically of the optimum $\mathbf{w}_u^\star$.

Using Social Welfare's linear representation (Lemma A.3) we have that

$$\text{SW}(\mathbf{w}_u^\star) - \text{SW}(\mathbf{w}_c^\star) = \text{SW}(\mathbf{w}_u^\star - \mathbf{w}_c^\star) = \langle \tilde{\mathbf{w}}, \mathbf{w}_u^\star - \mathbf{w}_c^\star \rangle$$

where $\tilde{\mathbf{w}} = (CA_1^{-1}C^\top \Pi_1 + CA_2^{-1}C^\top \Pi_2)\mathbf{w}^\star$ Now for $\tilde{\mathbf{w}}$ lying in the same direction with $\mathbf{w}_u^\star - \mathbf{w}_c^\star$ we get:

$$\text{SW}(\mathbf{w}_u^\star) - \text{SW}(\mathbf{w}_c^\star) = \|\tilde{\mathbf{w}}\|\|\mathbf{w}_u^\star - \mathbf{w}_c^\star\|$$

Using the example above we get $\|\mathbf{w}_u^\star - \mathbf{w}_c^\star\| = 2$ (recall the ball is of radius 1) hence

$$\text{SW}(\mathbf{w}_u^\star) - \text{SW}(\mathbf{w}_c^\star) = 2\|\tilde{\mathbf{w}}\|$$

We finally argue that indeed exists a chance for $\tilde{\mathbf{w}}$ to lie in the same direction with $\mathbf{w}_u^\star - \mathbf{w}_c^\star$.

Therefore, it suffices to enforce $\tilde{\mathbf{w}} \parallel \mathbf{w}^\star$. One simple sufficient condition is that

$$CA_1^{-1}C^\top \Pi_1 + CA_2^{-1}C^\top \Pi_2 = \lambda I \quad \text{for some } \lambda > 0,$$

in which case $\tilde{\mathbf{w}} = \lambda \mathbf{w}^\star$ and thus $\tilde{\mathbf{w}}$ is colinear with $\mathbf{w}_u^\star - \mathbf{w}_c^\star$ (and oriented in the same direction).

As a concrete instance, take $C = I$, $A_1 = A_2 = I$, and choose $\Pi_1, \Pi_2$ such that $\Pi_1 + \Pi_2 = \lambda I$ (e.g., *complementary orthogonal projections* scaled to sum to $\lambda I$). Then the above condition holds and the Cauchy–Schwarz step is tight. $\qquad\square$

**POLYHEDRAL FEASIBLE REGION BOUNDS (C AND D).**

**Proposition A.3 (Bound C: Internal polyhedron (Property 3.1) accuracy loss)** *For fairness spaces satisfying Property 3.1, when the learner's objective, $f$, is accuracy, optimality loss is upper bounded.*

$$\textit{if } \mathbf{w}^\star \in \mathcal{B}(1) : |f(\mathbf{w}_u^\star) - f(\mathbf{w}_c^\star)| \leq [H(M)\|(M\mathbf{w}^\star - \beta\mathbf{1})_+\|_2]^2$$

$$\textit{if } \mathbf{w}^\star \notin \mathcal{B}(1) : |f(\mathbf{w}_u^\star) - f(\mathbf{w}_c^\star)| \leq \left[H(M)\|(M\frac{\mathbf{w}^\star}{\|\mathbf{w}^\star\|_2} - \beta\mathbf{1})_+\|_2\right]^2$$

*Where $M \in \mathbb{R}^{k \times d}$ and $\mathbf{1} \in \mathbb{R}^k$ define the polyhedral representation of the fairness space. That is, the feasible region, $\mathcal{B}(1) \cap \mathcal{W}(\beta) = \{\mathbf{w} \in \mathbb{R}^d : M\mathbf{w} \leq \mathbf{b}\}$*

**Proof.** First, recall that accuracy optimization is simply euclidean projection onto the respective feasible region (Lemma A.2). Therefore, $\mathbf{w}_u^\star$ and $\mathbf{w}_c^\star$ are projections of $\mathbf{w}^\star$ onto $\mathcal{B}(1)$ and $\mathcal{B}(1) \cap \mathcal{W}(\beta)$ respectively. In order to prove Proposition A.3, we will leverage that $\mathbf{w}_c^\star$ is closer to $\mathbf{w}^\star$ than the projection of $\mathbf{w}_u^\star$ would be onto $\mathcal{W}(\beta) \cap \mathcal{B}(1)$. Let $\mathbf{z} := P_{\mathcal{W}(\beta) \cap \mathcal{B}(1)}(\mathbf{w}_u^\star)$ be this projection. Clearly, we have:

$$\|\mathbf{w}_c^\star - \mathbf{w}^\star\|_2^2 \leq \|\mathbf{z} - \mathbf{w}^\star\|_2^2 \qquad (\mathbf{w}_c^\star \text{ is optimal})$$
$$\leq \|\mathbf{w}_u^\star - \mathbf{w}^\star\|_2^2 + \|\mathbf{z} - \mathbf{w}_u^\star\|_2^2 \qquad (\text{triangle ineq})$$

Using Lemma A.2 this implies that

$$|f(\mathbf{w}_u^\star) - f(\mathbf{w}_c^\star)| = \|\mathbf{w}_c^\star - \mathbf{w}^\star\|_2^2 - \|\mathbf{w}_u^\star - \mathbf{w}^\star\|_2^2 \leq \|\mathbf{z} - \mathbf{w}_u^\star\|_2^2$$

Note that clearly $\|\mathbf{w}_c^\star - \mathbf{w}^\star\|_2 \geq \|\mathbf{w}_u^\star - \mathbf{w}^\star\|_2$. So now, we must simply upper bound $\|\mathbf{z} - \mathbf{w}_u^\star\|_2$. Using a Hoffman bound ((Hoffman, 1952)), we have that $\exists \mathbf{w}_0 \in \mathcal{W}(\beta) \cap \mathcal{B}(1), H(M) > 0$ such that,

$$[H(M)\|(M\mathbf{w}^\star - \beta\mathbf{1})_+\|_2]^2 \geq \|\mathbf{w}_u^\star - \mathbf{w}_0\|_2^2 \geq \|\mathbf{w}_u^\star - \mathbf{z}\|_2^2$$

Of course, we want this bound in terms of $\mathbf{w}^\star$ not $\mathbf{w}_u^\star$, but this is simple because since $\mathbf{w}_u^\star$ is the projection onto $\mathcal{B}(1)$, we have a closed form in terms of $\mathbf{w}^\star$. In particular, if $\mathbf{w}^\star \in \mathcal{B}(1)$, then $\mathbf{w}^\star = \mathbf{w}_u^\star$. Otherwise, we normalize it by the l-2 norm: $\mathbf{w}^\star/\|\mathbf{w}^\star\|_2 = \mathbf{w}_u^\star$ □

**Lemma A.5 (SW Loss on feasible sets that include the origin)** *For any feasible space $K$ of Problem 2 such that $K \subseteq \mathcal{B}(1)$ and $0 \in K$, we have that*

$$f(\mathbf{w}_u^\star) - f(\mathbf{w}_c^\star) \leq \sqrt{2}\|\tilde{\mathbf{w}}\|_2$$

*where $\tilde{\mathbf{w}} = (CA_1^{-1}C^\top\Pi_1 + CA_2^{-1}C^\top\Pi_2)\mathbf{w}^\star$ and $f$ is the social welfare.*

**Proof.** First, recall the social welfare linear representation as $f(\mathbf{w}) = \langle\tilde{\mathbf{w}}, \mathbf{w}\rangle$ from Lemma A.3. Since $0 \in K$, and $\mathbf{w}_c^\star$ is the maximizer over $K$, we have $\langle\tilde{\mathbf{w}}, \mathbf{w}_c^\star\rangle \geq \langle\tilde{\mathbf{w}}, 0\rangle = 0$. In addition, we know that the optimum over the unit Euclidean ball $\mathcal{B}(1)$ is exactly $\mathbf{w}_u^\star = \frac{\tilde{\mathbf{w}}}{\|\tilde{\mathbf{w}}\|_2}$. Hence:

$$\|\mathbf{w}_c^\star - \mathbf{w}_u^\star\|_2 = \sqrt{\|\mathbf{w}_c^\star\|_2^2 + \|\mathbf{w}_u^\star\|_2^2 - 2\langle\mathbf{w}_c^\star, \mathbf{w}_u^\star\rangle}$$

$$= \sqrt{\|\mathbf{w}_c^\star\|_2^2 + \|\mathbf{w}_u^\star\|_2^2 - 2\frac{1}{\|\tilde{\mathbf{w}}\|_2}\tilde{\mathbf{w}}^\top\mathbf{w}_c^\star}$$

$$\leq \sqrt{1 + 1 - 2*0} = \sqrt{2}$$

Since social welfare is a linear function:

$$\begin{aligned} f(\mathbf{w}_u^\star) - f(\mathbf{w}_c^\star) &= f(\mathbf{w}_u^\star - \mathbf{w}_c^\star) \\ &= \langle\tilde{\mathbf{w}}, \mathbf{w}_u^\star - \mathbf{w}_c^\star\rangle \\ &\leq \|\tilde{\mathbf{w}}\|_2\|\mathbf{w}_u^\star - \mathbf{w}_c^\star\|_2, \quad \text{by Cauchy-Schwarz inequality} \\ &\leq \sqrt{2}\|\tilde{\mathbf{w}}\|_2 \end{aligned}$$

□

**Corollary A.1 (Bound D:Internal polyhedron (Property 3.1) social welfare loss)** *For fairness spaces satisfying Property 3.1, when the learner's objective, $f$, is social welfare, optimality loss is upper bounded by*

$$|f(\mathbf{w}_u^\star) - f(\mathbf{w}_c^\star)| \leq \sqrt{2}\|\tilde{\mathbf{w}}\|_2$$

*where $\tilde{\mathbf{w}} := (CA_1^{-1}C^\top\Pi_1 + CA_2^{-1}C^\top\Pi_2)^\top\mathbf{w}^\star$.*

**Proof.** Notice that Property 3.1 defines a the feasible region as $\mathcal{B}(1) \cap \mathcal{W}(\beta; \Delta) = \{\mathbf{w} : \mathbf{w} \in \mathbb{R}^d, M\mathbf{w} \leq \beta\mathbf{1}\}$ for some $M \in \mathbb{R}^{k \times d}$ and $\mathbf{1} \in \mathbb{R}^k$, where $k \in \mathbb{N}, \beta > 0$. Clearly, $\mathbf{w} = \mathbf{0} \in$ said space. Therefore we can directly apply Lemma A.5. □

**ELLIPSOIDAL FAIR SPACE BOUNDS (A AND D).**

**Corollary A.2 (Bound A: Ellipsoidal (Property 3.2) accuracy loss)** *For fairness spaces satisfying Property 3.2, when the learner's objective, $f$, is accuracy, optimality loss is upper bounded by*

$$|f(\mathbf{w}_u^\star) - f(\mathbf{w}_c^\star)| \leq 4\max(\|\mathbf{w}^\star\|_2, 1)$$

**Proof.** Notice that the feasible region is the intersection of an ellipsoid (a convex space) and a euclidean ball (a convex space). Therefore, the by laws of convexity, the feasible region must also be convex. Thus we can invoke Proposition A.1. $\square$

**Corollary A.3 (Bound D: Ellipsoidal (Property 3.2) social welfare loss)** *For fairness spaces satisfying Property 3.2, when the learner's objective, $f$, is social welfare, optimality loss is upper bounded by*

$$|f(\mathbf{w}_u^\star) - f(\mathbf{w}_c^\star)| \leq \sqrt{2}\|\tilde{\mathbf{w}}\|_2$$

*where $\tilde{\mathbf{w}} := (CA_1^{-1}C^\top \Pi_1 + CA_2^{-1}C^\top \Pi_2)^\top \mathbf{w}^\star$.*

**Proof.** Notice that Property 3.2 defines a $\beta$-fair ellipsoid as: $\mathcal{W}(\beta; \Delta) = \{\mathbf{w} : \mathbf{w} \in \mathbb{R}^d, \mathbf{w}^\top Q\mathbf{w} \leq \beta\}, Q \succ 0, \beta > 0$. Clearly, $\mathbf{w} = \mathbf{0} \in \mathcal{W}(\beta; \Delta)$. Also $\mathbf{0} \in \mathcal{B}(1)$, so the feasible region, $\mathcal{W}(\beta; \Delta) \cap \mathcal{B}(1)$ includes $\mathbf{w} = \mathbf{0}$. Therefore we can directly apply Lemma A.5. $\square$

## ELLIPSOIDAL FEASIBLE REGION BOUNDS (E AND F).

**Lemma A.6** *Consider the following convex optimization problem, where $Q \in \mathbb{R}^{d \times d}$ and $Q \succ 0$. and $\tilde{\mathbf{w}} \neq \mathbf{0}$*

$$\begin{aligned} minimize_{w \in \mathbb{R}^d} \quad & \langle \tilde{\mathbf{w}}, \mathbf{w} \rangle \\ subject\ to \quad & \mathbf{w}^\top Q\mathbf{w} - \beta \leq 0 \end{aligned} \tag{9}$$

*The optimal solution is*

$$\hat{\mathbf{w}}_{ellipsoid}^\star = \frac{-\sqrt{\beta}Q^{-1}\tilde{\mathbf{w}}}{\|\sqrt{Q^{-1}}\tilde{\mathbf{w}}\|_2}$$

**Proof.** By Slater's condition, we have that the KKT conditions are necessary and sufficient for optimality. Therefore, any $\mathbf{w}$ satisfying them must be optimal. We shall proceed by solving the KKT conditions.

$$\begin{aligned} minimize_{w \in \mathbb{R}^d} \quad & \langle \tilde{\mathbf{w}}, \mathbf{w} \rangle \\ subject\ to \quad & \mathbf{w}^\top Q\mathbf{w} - \beta \leq 0 \end{aligned} \tag{10}$$

The KKT conditions state:

$$-\tilde{\mathbf{w}} = 2\lambda Q\mathbf{w}$$
$$\lambda \geq 0$$
$$\lambda(\mathbf{w}^\top Q\mathbf{w} - \beta) = 0$$
$$\mathbf{w}^\top Q\mathbf{w} \leq \beta$$

$\lambda \neq 0$ because if it were, then $\tilde{\mathbf{w}} = \mathbf{0}$, which would be a contradiction. Therefore it must be the case that $\lambda > 0$.

Using the first KKT condition we have that $\mathbf{w} = \frac{-Q^{-1}\tilde{\mathbf{w}}}{2\lambda}$. $Q$ is PD, therefore it is also invertible. Because $\lambda > 0$, it must be the case that $\mathbf{w}^\top Q\mathbf{w} = \beta$ (3rd KKT condition). Thus we have:

$$\mathbf{w}^\top \mathbf{w} = \left[\frac{-Q^{-1}\tilde{\mathbf{w}}}{2\lambda}\right]^\top Q \frac{-Q^{-1}\tilde{\mathbf{w}}}{2\lambda} = \frac{1}{4\lambda^2}\tilde{\mathbf{w}}^\top Q^{-1}\tilde{\mathbf{w}} = \beta$$

Notice that if $Q$ is PD, it is symmetric. Solving for $\lambda$, we see that $\lambda^\star = \frac{1}{2\sqrt{\beta}}\|\sqrt{Q^{-1}}\tilde{\mathbf{w}}\|_2$. Substituting this into $\mathbf{w} = \frac{-Q^{-1}\tilde{\mathbf{w}}}{2\lambda}$ we have

$$\hat{\mathbf{w}}_{ellipsoid}^\star = \frac{-\sqrt{\beta}Q^{-1}\tilde{\mathbf{w}}}{\|\sqrt{Q^{-1}}\tilde{\mathbf{w}}\|_2}$$

$\square$

**Proposition A.4 (Bound F: Internal ellipsoid (Property 3.3) SW loss)** *Assume desirability fairness space, $\mathcal{W}(\beta)$ satisfies property 3.3. Then social welfare loss is exactly:*

$$SW(\mathbf{w}_u^\star) - SW(\mathbf{w}_c^\star) = \|(CA_1^{-1}C^\top \Pi_1 + CA_2^{-1}C^\top \Pi_2)^\top \mathbf{w}^\star\|_2 - \sqrt{\beta}\|(CA_1^{-1}C^\top \Pi_1 + CA_2^{-1}C^\top \Pi_2)^\top \mathbf{w}^\star\|_{Q^{-1}}$$

**Proof.** Recall that Social Welfare is a functionally linear objective (Lemma A.3). Specifically: $\langle \mathbf{c}, \mathbf{w} \rangle$ where $\mathbf{c} := (CA_1^{-1}C^\top \Pi_1 + CA_2^{-1}C^\top \Pi_2)^\top \mathbf{w}^\star$

Using Lemma A.6 we get the closed from for the solution for the unconstrained and fairness constrained problem, which we then plug back in for the optimal value. In each case, $\tilde{\mathbf{w}} := -(CA_1^{-1}C^\top \Pi_1 + CA_2^{-1}C^\top \Pi_2)^\top \mathbf{w}^\star$

Equation 3: $Q = I, \beta = 1$. This yields:

$$\mathbf{w}_u^\star := \frac{(CA_1^{-1}C^\top \Pi_1 + CA_2^{-1}C^\top \Pi_2)^\top \mathbf{w}^\star}{\|(CA_1^{-1}C^\top \Pi_1 + CA_2^{-1}C^\top \Pi_2)^\top \mathbf{w}^\star\|_2} \tag{11}$$

$$SW_u = \langle (CA_1^{-1}C^\top \Pi_1 + CA_2^{-1}C^\top \Pi_2)^\top \mathbf{w}^\star, \frac{(CA_1^{-1}C^\top \Pi_1 + CA_2^{-1}C^\top \Pi_2)^\top \mathbf{w}^\star}{\|(CA_1^{-1}C^\top \Pi_1 + CA_2^{-1}C^\top \Pi_2)^\top \mathbf{w}^\star\|_2} \rangle$$

$$= \|(CA_1^{-1}C^\top \Pi_1 + CA_2^{-1}C^\top \Pi_2)^\top \mathbf{w}^\star\|_2$$

Equation 2: $Q = Q, \beta = \beta$. This yields:

$$\hat{\mathbf{w}}_c^\star := \frac{\sqrt{\beta} Q^{-1}(CA_1^{-1}C^\top \Pi_1 + CA_2^{-1}C^\top \Pi_2)^\top \mathbf{w}^\star}{\|\sqrt{Q^{-1}}(CA_1^{-1}C^\top \Pi_1 + CA_2^{-1}C^\top \Pi_2)^\top \mathbf{w}^\star\|_2} \tag{12}$$

$$SW_c = \langle (CA_1^{-1}C^\top \Pi_1 + CA_2^{-1}C^\top \Pi_2)^\top \mathbf{w}^\star, \frac{\sqrt{\beta} Q^{-1}(CA_1^{-1}C^\top \Pi_1 + CA_2^{-1}C^\top \Pi_2)^\top \mathbf{w}^\star}{\|\sqrt{Q^{-1}}(CA_1^{-1}C^\top \Pi_1 + CA_2^{-1}C^\top \Pi_2)^\top \mathbf{w}^\star\|_2} \rangle$$

$$= \sqrt{\beta} \|(CA_1^{-1}C^\top \Pi_1 + CA_2^{-1}C^\top \Pi_2)^\top \mathbf{w}^\star\|_{Q^{-1}}$$

Subtracting the two yields the result of the proposition. □

**Lemma A.7 (Useful fact about ellipsoids)** *Consider an ellipsoid,* $\mathcal{E}(\beta) := \{\mathbf{w} \in \mathbb{R}^d : \mathbf{w}^\top Q \mathbf{w} \leq \beta\}$.

*The largest radius:* $\sqrt{\frac{\beta}{\lambda_d(Q)}} = \max_{\mathbf{w} \in \mathcal{E}(\beta)} \|\mathbf{w}\|_2$

*Thus the ball envelope of the ellipsoid is* $\mathcal{B}(\sqrt{\frac{\beta}{\lambda_d(Q)}}) := \{\mathbf{w} \in \mathbb{R}^d : \mathbf{w}^\top \mathbf{w} \leq \frac{\beta}{\lambda_d(Q)}\}$

**Proof.** We can prove the first claim by applying KKT and noting that Slater's conditions applies. Consider the problem:

$$\begin{aligned} \text{maximize} \quad & \|\mathbf{w}\|_2 \\ \text{subject to} \quad & \mathbf{w}^\top Q \mathbf{w} - \beta \leq 0 \end{aligned}$$

KKT conditions:

1. $\frac{\mathbf{w}}{\|\mathbf{w}\|} = 2\mu Q \mathbf{w}$

2. $\mu \geq 0$

3. $\mu(\mathbf{w}^\top Q \mathbf{w} - \beta) = 0$

4. $\mathbf{w}^\top Q \mathbf{w} \leq \beta$

First, notice that $\mu > 0$, because otherwise, $\mathbf{w}$ must be 0. So it must be the case that for complementary slackness, $\mathbf{w}^\top Q \mathbf{w} = \beta$. Then, from the first condition we have: $Q\mathbf{w} = \frac{\mathbf{w}}{2\mu\|\mathbf{w}\|} = \lambda \mathbf{w}$ where $\lambda = \frac{1}{2\mu\|\mathbf{w}\|}$. Thus we have $\beta = \mathbf{w}^\top \lambda \mathbf{w} = \lambda \|\mathbf{w}\|^2$ which means that $\|\mathbf{w}\|_2 = \sqrt{\frac{\beta}{\lambda}}$. But now notice that $\lambda$ is, by definition, an eigenvalue of of $Q$! Then getting $\mathbf{w}^\star = \sqrt{\frac{\beta}{\lambda_d(Q)}}$ is clear because $\mathbf{w}^\star$ is a maximum and $\lambda_d(Q)$ is the smallest eigenvalue of $Q$.

The statement about the ball envelope follows immediately. Notice that if $\mathbf{w} \in \mathcal{E}(\beta)$, we have now proven that it must be the case that $\|\mathbf{w}\| \leq \sqrt{\frac{\beta}{\lambda_d(Q)}}$ thus $\mathbf{w} \in$ the stated ball. □

**Proposition A.5 (Bound E: Internal ellipsoid(Property 3.3) accuracy loss)** *Assume property 3.3 is satisfied. Let $\mathbf{w}'$ be a maximizer on the problem:*

$$\max -\|\mathbf{w} - \mathbf{w}^\star\|_2^2$$
$$\mathbf{w}^\top Q \mathbf{w} \leq \beta$$

*Then we can bound the accuracy loss between $\mathbf{w}'$ and unconstrained (normalized on L2) policy $\mathbf{w}_u^\star$ (solution to Equation 3)*

$$|\text{ACC}(\mathbf{w}') - \text{ACC}(\mathbf{w}_u^\star)| \leq (2q)(t + s)$$

$q = \mathbb{I}_{\mathbf{w}^\star \notin \mathcal{E}(\beta)}\left(\sqrt{\frac{\beta}{\lambda_d(Q)}} + \|\mathbf{w}^\star\|\right)$, $s = \mathbb{I}_{\mathbf{w}^\star \notin \mathcal{E}(\beta)}(\min\{1, \|\mathbf{w}^\star\|\} + \sqrt{\frac{\beta}{\lambda_d(Q)}})$, *and* $t = 2\sqrt{\frac{\beta}{\lambda_d(Q)}}$

**Proof.** Define $\mathcal{E}(\beta) := \{\mathbf{w} \in \mathbb{R}^d : \mathbf{w}^\top Q \mathbf{w} \leq \beta\}$ to be the ellipsoid internal to the euclidean ball.

Let $P_{\mathcal{E}(\beta)}(\mathbf{w})$ be the projection of $\mathbf{w}$ onto $\mathcal{E}(\beta)$ and $s_{\max}$ be $\max_{\mathbf{w} \in \mathcal{E}(\beta)} \|\mathbf{w}\|_2$, the maximum length a vector, $\mathbf{w}$ could be and still be in the ellipsoid.

Notice by triangle inequality and properties of projection:

$$\|P_{\mathcal{E}(\beta)}(\mathbf{w}_u^\star) - \mathbf{w}_u^\star\| \leq \mathbb{I}_{\mathbf{w}^\star \notin \mathcal{E}(\beta)}(\|\mathbf{w}_u^\star\|_2 + \|P_{\mathcal{E}(\beta)}(\mathbf{w}_u^\star)\|_2) \leq \mathbb{I}_{\mathbf{w}^\star \notin \mathcal{E}(\beta)}(\min\{1, \|\mathbf{w}^\star\|\} + s_{\max})$$

Thus we have,

$$\begin{aligned}
\|\mathbf{w}' - \mathbf{w}_u^\star\|_2 &= \|P_{\mathcal{E}(\beta)}(\mathbf{w}^\star) - \mathbf{w}_u^\star\|_2 && \text{(definition)} \\
&\leq \|P_{\mathcal{E}(\beta)}(\mathbf{w}^\star) - P_{\mathcal{E}(\beta)}(\mathbf{w}_u^\star)\|_2 + \|P_{\mathcal{E}(\beta)}(\mathbf{w}_u^\star) - \mathbf{w}_u^\star\|_2 && \text{(triangle ineq)} \\
&\leq \|P_{\mathcal{E}(\beta)}(\mathbf{w}^\star) - P_{\mathcal{E}(\beta)}(\mathbf{w}_u^\star)\|_2 + \mathbb{I}_{\mathbf{w}^\star \notin \mathcal{E}(\beta)}(\min\{1, \|\mathbf{w}^\star\|\} + s_{\max}) && \text{(earlier claim)} \\
&\leq 2s_{\max} + \mathbb{I}_{\mathbf{w}^\star \notin \mathcal{E}(\beta)}(\min\{1, \|\mathbf{w}^\star\|\} + s_{\max}) && \text{(diameter of } \mathcal{E}(\beta)\text{)}
\end{aligned}$$

Additionally, we will also want:

$$\begin{aligned}
\|\mathbf{w}' - \mathbf{w}^\star\|_2 &\leq \mathbb{I}_{\mathbf{w}^\star \notin \mathcal{E}(\beta)}(\|P_{\mathcal{E}(\beta)}(\mathbf{w}^\star)\| + \|\mathbf{w}^\star\|) && \text{(triangle ineq, } \mathcal{E}(\beta) \subseteq \mathcal{B}(1)\text{)} \\
&\leq \mathbb{I}_{\mathbf{w}^\star \notin \mathcal{E}(\beta)}(s_{\max} + \|\mathbf{w}^\star\|) && \text{(definition of } s_{\max}\text{)}
\end{aligned}$$

Now we can write using Lemma A.2:

$$\begin{aligned}
|\text{ACC}(\mathbf{w}') - \text{ACC}(\mathbf{w}_u^\star)| &= |-\|\mathbf{w}' - \mathbf{w}^\star\|_2^2 + \|\mathbf{w}_u^\star - \mathbf{w}^\star\|_2^2| && \text{(Lemma A.2)} \\
&= |(\|\mathbf{w}' - \mathbf{w}^\star\|_2 + \|\mathbf{w}_u^\star - \mathbf{w}^\star\|_2)(\|\mathbf{w}' - \mathbf{w}^\star\|_2 - \|\mathbf{w}_u^\star - \mathbf{w}^\star\|_2)| && (a^2 - b^2 = (a+b)(a-b)) \\
&= (\|\mathbf{w}' - \mathbf{w}^\star\|_2 + \|\mathbf{w}_u^\star - \mathbf{w}^\star\|_2)|(\|\mathbf{w}' - \mathbf{w}^\star\|_2 - \|\mathbf{w}_u^\star - \mathbf{w}^\star\|_2)| && \text{(factor is nonneg)} \\
&\leq (\|\mathbf{w}' - \mathbf{w}^\star\|_2 + \|\mathbf{w}_u^\star - \mathbf{w}^\star\|_2)(\|\mathbf{w}' - \mathbf{w}_u^\star\|_2) && \text{(reverse triangle ineq)} \\
&\leq (2\|\mathbf{w}' - \mathbf{w}^\star\|_2)(\|\mathbf{w}' - \mathbf{w}_u^\star\|_2) && (\mathcal{E}(\beta) \subseteq \mathcal{B}(1)) \\
&\leq (2q)(t + s) && \text{(all earlier claims)}
\end{aligned}$$

where $q = \mathbb{I}_{\mathbf{w}^\star \notin \mathcal{E}(\beta)}(s_{\max} + \|\mathbf{w}^\star\|)$, $s = \mathbb{I}_{\mathbf{w}^\star \notin \mathcal{E}(\beta)}(\min\{1, \|\mathbf{w}^\star\|\} + s_{\max})$, and $t = 2s_{\max}$

Finally Lemma A.7 gives us the solution to $s_{\max} = \sqrt{\frac{\beta}{\lambda_d(Q)}}$ $\qquad \square$

**Proposition A.6** *Bound E from Proposition A.5, is not strictly better than Bound A:* $4\max(\|\mathbf{w}^\star\|_2, 1)$ *of Proposition A.2*

**Proof.** Consider for example $\|w^\star\| \leq 1$ but $w^\star \notin \mathcal{E}(\beta)$, Q diagonal and $\beta = (1-\delta)^2 \lambda_d(Q)$. Then we have $s = 1 - \delta$ and $q = s' = \|\mathbf{w}^\star\| + 1 - \delta$, hence our bound reduces to $2(\|\mathbf{w}^\star\| + 1 - \delta)(\|\mathbf{w}^\star\| + 1 - \delta + 1 - \delta) \approx 2(\|\mathbf{w}^\star\| + 1)(\|\mathbf{w}^\star\| + 2)$ which is larger than $4\max(\|w^\star\|_2, 1)$ for all $\|w^\star\| > 0$.
$\square$

A.2.2. SUPPLEMENTAL NUMERICAL EXAMPLES

To supplement Table 1 bounds, one may consider the following numerical examples.

**Example A.1 (Bounded optimality loss given non-disparate costs)** *Let agents have 2 features ($d = 2$). Cost is non-disparate and changing either feature requires unit cost, so $A_g = \mathbf{I}_2$. Features do not have any causal flow between one another, so $C = \mathbf{I}_2$. The first feature is desirable, while the 2nd feature is slightly less so, $\Pi_D : diag(1, 3/4)$. Agents come from a feature distribution that results in $\Pi_1 := diag(1, 0)$ and $\Pi_2 := diag(0, 1)$. Finally $\mathbf{w}^\star = (1/2, 1/2)$. Using the fairness constraint of Example 3.1, how much accuracy is lost in Stackelberg equilibrium?*

*By Corollary A.5, if $\beta \leq 3/4$, this satisfies Property 3.1. Thus using the polyhedral construction of Lemma A.8 and the bound in Table 1, accuracy loss is bounded:*

$$|f(\mathbf{w}_u^\star) - f(\mathbf{w}_c^\star)| \leq \left[H(\widehat{M})\right]^2 [(1/8 - \beta)_+]^2 + (7/8 - \beta)^2]$$

*if we further have $1/8 \leq \beta \leq 3/4$:*

$$|f(\mathbf{w}_u^\star) - f(\mathbf{w}_c^\star)| \leq \left[\frac{3H(\widehat{M})}{4}\right]^2$$

*Where:*

$$\widehat{M} = \begin{bmatrix} -1 & -3/4 \\ 1 & -3/4 \\ 1 & 3/4 \\ -1 & 3/4 \end{bmatrix}$$

**Example A.2 (Bounded accuracy loss given non-disparate feature distributions)** *Let agents have 2 features ($d = 2$). Feature space is non-disparate and nonskewed, so $\Pi_g = \mathbf{I}_2$. Features do not have any causal flow between one another, so $C = \mathbf{I}_2$. The first feature is desirable, while the 2nd feature is slightly less so, $\Pi_D : diag(1, 3/4)$. Agents have disparate costs such that all change is easier for group 1 agents: $\Pi_1 := diag(1/2, 1/2)$ and $\Pi_2 := diag(1, 1)$. Finally $\mathbf{w}^\star = (1/2, 1/2)$. Using the fairness constraint of Example 3.1, how much accuracy is lost in Stackelberg equilibrium?*

*By Corollary A.6 if $\beta \leq 3/4$, this satisfies Property 3.1. Thus using the polyhedral construction of Lemma A.8 and the bound in Table 1, accuracy loss is bounded:*

$$|f(\mathbf{w}_u^\star) - f(\mathbf{w}_c^\star)| \leq \left[H(\widehat{M})\right]^2 [(1/8 - \beta)_+]^2 + (7/8 - \beta)^2]$$

*if we further have $1/8 \leq \beta \leq 3/4$:*

$$|f(\mathbf{w}_u^\star) - f(\mathbf{w}_c^\star)| \leq \left[\frac{3H(\widehat{M})}{4}\right]^2$$

*Where:*

$$\widehat{M} = \begin{bmatrix} -1 & -3/4 \\ 1 & -3/4 \\ 1 & 3/4 \\ -1 & 3/4 \end{bmatrix}$$

A.2.3. SUPPLEMENTAL MATERIAL FOR EXAMPLE 3.1

First we note some usually assumptions used in this Appendix section and Appendix A.2.4

**Assumption A.1 (Unknown feature space is exclusive)** $\ker(\Pi_1) \cap \ker(\Pi_2) = \emptyset$

**Assumption A.2 (Estimated w is different)** $\Pi_1 \mathbf{w} \neq \Pi_2 \mathbf{w} \quad \forall \mathbf{w} \neq \mathbf{0}$

**Lemma A.8 ($\mathcal{W}(\beta)$ from Example 3.1 is a polyhedron)** $\mathcal{W}(\beta)$ *of Example 3.1 can be rewritten as:*

$$\mathcal{W}(\beta) = \{\mathbf{w} : \mathbf{w} \in \mathbb{R}^d, \mathbf{a}^\top M \mathbf{w} \leq \beta \quad \forall \mathbf{a} \in \mathcal{A}\}$$

*Where $M := \Pi_D A_1^{-1} C^\top \Pi_1 - \Pi_D A_2^{-1} C^\top \Pi_2$ and $\mathcal{A} := \{(a_1, \ldots, a_d) \in \mathbb{R}^d : a_i \in \{-1, 1\} \forall i \in [d]\}$*

*This is clearly a polyhedron of $2^d$ constraints each defined by $\mathbf{a}^\top M$*

**Proof.**

$$\sum_{i \in [d]} |(\Pi_D \mathbf{x}_e^{(1)}(\mathbf{w}) - \Pi_D \mathbf{x}_e^{(2)}(\mathbf{w}))_i| = \sum_{i \in [d]} |(\Pi_D A_1^{-1} C^\top \Pi_1 \mathbf{w} - \Pi_D A_2^{-1} C^\top \Pi_2 \mathbf{w})_i|$$

$$= \sum_{i \in [d]} |((\Pi_D A_1^{-1} C^\top \Pi_1 - \Pi_D A_2^{-1} C^\top \Pi_2) \mathbf{w})_i|$$

$$= \|M\mathbf{w}\|_1$$

Where $M := \Pi_D A_1^{-1} C^\top \Pi_1 - \Pi_D A_2^{-1} C^\top \Pi_2$

Recall that $\{\mathbf{y} : \mathbf{y} \in \mathbb{R}^d, \|\mathbf{y}\|_1 \leq \beta\}$ describes a $d$-dimensional cube in $\mathbf{y}$ space, such shapes are clearly polyhedra. We will show that $\|M\mathbf{w}\|_1 \leq \beta$ creates the polyhedron described by the lemma.

We can describe $\{\mathbf{y} : \mathbf{y} \in \mathbb{R}^d, \|\mathbf{y}\|_1 \leq \beta\}$ equivalently as: $\{\mathbf{y} : \mathbf{y} \in \mathbb{R}^d, \mathbf{a}^\top \mathbf{y} \leq \beta \quad \forall \mathbf{a} \in \mathcal{A}\}$ where $\mathcal{A} := \{(a_1, \ldots, a_d) \in \mathbb{R}^d : a_i \in \{-1, 1\} \forall i \in [d]\}$ Note that $|\mathcal{A}| = 2^d$. Let $\mathbf{y} = M\mathbf{w}$ and this gives polyhedron of the lemma. $\square$

Importantly, Property 3.1 requires that $\mathcal{B}(1) \cap \mathcal{W}(\beta)$ is a polyhedron! Therefore, for the optimality loss bound associated with this Property, we should ensure that $\mathcal{W}(\beta) \subseteq \mathcal{B}(1)$

**Proposition A.7 (Necessary and sufficient conditions for Example 3.1 to satisfy property 3.1)** *The fairness function described by Example 3.1 satisfies Property 3.1 if and only if $M$, where $M := \Pi_D A_1^{-1} C^\top \Pi_1 - \Pi_D A_2^{-1} C^\top \Pi_2$ is such that (1) $\ker(M) = \emptyset$ and (2) $\beta \leq \inf_{\|\mathbf{w}\|_2=1} \|M\mathbf{w}\|_1$*

**Proof.** First, notice that by Lemma A.8, the fairness space, $\mathcal{W}(\beta)$ is a polyhedron for any $M$ and $\beta > 0$. Because $\mathcal{B}(1)$ is an ellipsoid, this means that for $\mathcal{W}(\beta) \cup \mathcal{B}(1)$ to be a polyhedron, $\mathcal{W}(\beta) \subseteq \mathcal{B}(1)$. So we must show that conditions (1) and (2) are necessary and sufficient to ensure that $\mathcal{W}(\beta) \subseteq \mathcal{B}(1)$. We will first prove that conditions (1) and (2) are necessary.

Notice that if $\ker(M) \neq 0$, then there exists some $\mathbf{w}_0 \in \ker(M)$ s.t. $\|\mathbf{w}\|_2 \geq 1$, but $\|M\mathbf{w}_0\|_1 = 0 < \beta$. That would mean $\mathcal{W}(\beta) \not\subseteq \mathcal{B}(1)$ Thus $\ker(M) = \emptyset$ must be necessary.

Let $\mu(M) := \inf_{\|\mathbf{w}\|_2=1} \|M\mathbf{w}\|_1$ Now notice that $\|M\mathbf{w}\|_1 = \|\mathbf{w}\|_2 \|M \frac{\mathbf{w}}{\|\mathbf{w}\|_2}\|_1 \geq \mu(M)\|\mathbf{w}\|_2 \quad \forall \mathbf{w}$ This implies $\frac{\|M\mathbf{w}\|_1}{\mu(M)} \geq \|\mathbf{w}\|_2$. Of course $\forall \mathbf{w} \in \mathcal{W}(\beta)$:

$$\frac{\beta}{\mu(M)} \geq \frac{\|M\mathbf{w}\|_1}{\mu(M)} \geq \|\mathbf{w}\|_2$$

From this, we see that in order for $\mathcal{W}(\beta) \subseteq \mathcal{B}(1)$, it is necessary that $\beta \leq \inf_{\|\mathbf{w}\|_2=1} \|M\mathbf{w}\|_1$.

Now we will show that conditions (1) and (2) are sufficient. We will do this by contradiction. Suppose that $\mathbf{w}_0 \in \mathcal{W}(\beta)$, but $\mathbf{w}_0 \notin \mathcal{B}(1)$ while both (1) and (2) hold. This means that $\|\mathbf{w}_0\|_2 > 1$ and $\|M\mathbf{w}_0\|_1 \leq \beta$. From condition (2), $\|\mathbf{w}_0\|_2 \|M \frac{\mathbf{w}_0}{\|\mathbf{w}_0\|_2}\|_1 = \|M\mathbf{w}_0\|_1 \leq \beta$. From condition (1) we know that $\mu(M) > 0$ and then from the definition of $\mu(M)$: $\|M \frac{\mathbf{w}_0}{\|\mathbf{w}_0\|_2}\|_1 \geq \mu(M)$. Thus it must be the case that $\|\mathbf{w}_0\|_2 \leq 1$. But this poses a contradiction! Thus we see that when conditions (1) and (2) hold, there cannot exist such a $\mathbf{w}$ where $\mathbf{w}_0 \in \mathcal{W}(\beta)$, but $\mathbf{w}_0 \notin \mathcal{B}(1)$, which means $\mathcal{W}(\beta) \subseteq \mathcal{B}(1)$. $\square$

**Corollary A.4 (Sufficient conditions for Example 3.1 to satisfy property 3.1)** *The fairness function described by Example 3.1 satisfies Property 3.1 if $M$, where $M := \Pi_D A_1^{-1} C^\top \Pi_1 - \Pi_D A_2^{-1} C^\top \Pi_2$ is such that $\ker(M) = \emptyset$ and $\beta \leq \sigma_d(M)$*

**Proof.** We should show that $\beta \leq \sigma_d(M) \implies \beta \leq \inf_{\|\mathbf{w}\|_2=1} \|M\mathbf{w}\|_1$ This is simple. Note that rank of $M$ must be $d$ because it is a $d \times d$ matrix with an empty kernel. So the $d$th singular value is the smallest nonzero singular value. So we have $\inf_{\|\mathbf{w}\|_2=1} \|M\mathbf{w}\|_2 = \sigma_d(M)$ from the Min-Max theorem for singular values. And because $\forall \mathbf{y} \in \mathbb{R}^d, \|\mathbf{y}\|_2 \leq \|\mathbf{y}\|_2$, we have $\inf_{\|\mathbf{w}\|_2=1} \|M\mathbf{w}\|_2 = \sigma_d(M) \leq \inf_{\|\mathbf{w}\|_2=1} \|M\mathbf{w}\|_1$. Thus clearly $\beta \leq \sigma_d(M) \implies \beta \leq \inf_{\|\mathbf{w}\|_2=1} \|M\mathbf{w}\|_1$ $\square$

These conditions, even in only the sufficient form are hard to interpret in the context of the setting specifically given that $M$ is a function of several setting parameters. To make things a clearer, we can simplify to more specific settings in which groups have either cost or information discrepancy:

**Corollary A.5 (Sufficient conditions for Example 3.1 to satisfy property 3.1 w/ no cost asymmetry)** *Suppose that agents in our setting have the same cost to feature change (i.e. $A_1 = A_2 = A_g$). Then, the fairness function described by Example 3.1 satisfies Property 3.1 if Assumption A.1 is satisfied, $\Pi_1\mathbf{w} \neq \Pi_2\mathbf{w} \quad \forall \mathbf{w}, \mathbf{w} \neq \mathbf{0}$, and $\beta \leq \sigma_d(\Pi_d A_g^{-1} C^\top [\Pi_1 - \Pi_2])$.*

**Proof.**

$$M = \Pi_D A_1^{-1} C^\top \Pi_1 - \Pi_D A_2^{-1} C^\top \Pi_2$$
$$= \Pi_D A_g^{-1} C^\top [\Pi_1 - \Pi_2]$$

Clearly, the singular value condition is the same as the sufficient condition from A.4. Thus, all we must prove that Assumption A.1 and $\Pi_1\mathbf{w} \neq \Pi_2\mathbf{w} \quad \forall \mathbf{w}, \mathbf{w} \neq \mathbf{0} \implies \ker(M) = \emptyset$. Notice that that $\ker(\Pi_D A_g^{-1} C^\top) = \emptyset$ because $\Pi_D$ and $A_g$ are positive definite by setting assumptions and $\ker(C) = \emptyset$ by Lemma A.1. Thus all that matters is $\ker(\Pi_1 - \Pi_2)$. Clearly, as long as

1. $\forall \mathbf{w} \in \ker(\Pi_1)$, $\mathbf{w} \notin \ker(P_2)$ and $\forall \mathbf{w}' \in \ker(\Pi_2)$, $\mathbf{w}' \notin \ker(P_1)$

2. $\Pi_1\mathbf{w} \neq \Pi_2\mathbf{w} \quad \forall \mathbf{w}, \mathbf{w} \neq \mathbf{0}$

then $\ker(\Pi_D A_g^{-1} C^\top [\Pi_1 - \Pi_2])$ will be empty. $\square$

**Corollary A.6 (Sufficient conditions for Example 3.1 to satisfy property 3.1 w/ no info asymmetry)** *Suppose that agents in our setting have the same information (i.e. $\Pi_1 = \Pi_2 = \Pi_g$). Then, the fairness function described by Example 3.1 satisfies Property 3.1 if $\ker(\Pi_g) = \emptyset$, $A_2 \succ A_1$ and $\beta \leq \sigma_d(\Pi_D[A_1^{-1} - A_2^{-1}]C^\top \Pi_g)$.*

**Proof.**

$$M = \Pi_D A_1^{-1} C^\top \Pi_1 - \Pi_D A_2^{-1} C^\top \Pi_2$$
$$= \Pi_D[A_1^{-1} - A_2^{-1}]C^\top \Pi_g$$

Clearly, the singular value condition is the same as the sufficient condition from A.4. Thus, all we must prove that $\ker(\Pi_g) = \emptyset$, $A_2 \succ A_1$ is sufficient to show that $\ker(M) = \emptyset$. $\Pi_D$ and $A_g$ are positive definite by setting assumptions and $\ker(C) = \emptyset$ by Lemma A.1. So we consider $\ker(A_1^{-1} - A_2^{-1})$ and $\ker(\Pi_g)$. Clearly if $\ker(A_1^{-1} - A_2^{-1}) = \emptyset$ and $\ker(\Pi_g) = \emptyset$ then $\ker(\Pi_D[A_1^{-1} - A_2^{-1}]C^\top \Pi_g) = \emptyset$. Note that:

$$A_2 \succ A_1 \implies A_1^{-1} \succ A_2^{-1} \implies A_1^{-1} - A_2^{-1} \succ 0 \implies \ker(A_1^{-1} - A_2^{-1}) = \emptyset$$

$\square$

A.2.4. SUPPLEMENTAL MATERIAL FOR EXAMPLE 3.2

**Proposition A.8 (Example 3.2 is (sometimes) an ellipsoid)** *Example 3.2 represents and ellipsoid if and only if $M = \Pi_D(A_1^{-1} C^T P_1 - A_2^{-1} C^T P_2)$ is invertible.*

**Proof.** Expanding $\mathbf{x}_e^{(1)}, \mathbf{x}_e^{(2)}$ according to the closed-form solutions we identified in Proposition 2.1, we can think of Example 3.2 as $\|A\mathbf{w} - B\mathbf{w}\|_2^2 = \|(A - B)\mathbf{w}\|_2^2 \leq \beta$ where $A = \Pi_D(A_1^{-1} C^T P_1)$ and $B = \Pi_D(A_2^{-1} C^T P_2)$. Now can rewrite $\|(A - B)\mathbf{w}\|_2^2 = (A - B)^\top(A - B) \leq \beta$. This set represents an ellipsis when $M = (A - B)^\top(A - B) \succ 0$. Now we know $M \succeq 0$ always and $M \succ 0$ when $(A - B)$ is invertible.
Now we can study $(A - B) = \Pi_D(A_1^{-1} C^\top P_1 - A_2^{-1} C^\top P_2)$.
Hence $A - B$ is invertible iff $M = \Pi_D(A_1^{-1} C^\top P_1 - A_2^{-1} C^\top P_2)$ is invertible

$\square$

**Corollary A.7 (Sufficient conditions for Example 3.2 to satisfy property 3.2 w/ no cost asymmetry)** *Suppose that agents in our setting have the same cost to feature change (i.e. $A_1 = A_2 = A_g$). Then, the fairness function described by Example 3.2 satisfies Property 3.2 if Assumptions A.1 and A.2 are satisfied.*

**Proof.**

$$M = \Pi_D A_1^{-1} C^\top \Pi_1 - \Pi_D A_2^{-1} C^\top \Pi_2$$
$$= \Pi_D A_g^{-1} C^\top [\Pi_1 - \Pi_2]$$

All we must prove that Assumptions A.1 and A.2 $\implies \ker(M) = \emptyset$. Notice that that $\ker(\Pi_D A_g^{-1} C^\top) = \emptyset$ because $\Pi_D$ and $A_g$ are positive definite by setting assumptions and $\ker(C) = \emptyset$ by Lemma A.1. Thus all that matters is $\ker(\Pi_1 - \Pi_2)$. Clearly, as long as

1. $\forall \mathbf{w} \in \ker(\Pi_1)$, $\mathbf{w} \notin \ker(P_2)$ and $\forall \mathbf{w}' \in \ker(\Pi_2)$, $\mathbf{w}' \notin \ker(P_1)$

2. $\Pi_1 \mathbf{w} \neq \Pi_2 \mathbf{w} \quad \forall \mathbf{w}, \mathbf{w} \neq \mathbf{0}$

then $\ker(\Pi_D A_g^{-1} C^\top [\Pi_1 - \Pi_2])$ will be empty. $\qquad\square$

**Corollary A.8 (Sufficient conditions for Example 3.2 to satisfy property 3.2 w/ no info asymmetry)** *Suppose that agents in our setting have the same information (i.e. $\Pi_1 = \Pi_2 = \Pi_g$). Then, the fairness function described by Example 3.2 satisfies Property 3.2 if $\ker(\Pi_g) = \emptyset$, $A_2 \succ A_1$.*

**Proof.**
$$M = \Pi_D A_1^{-1} C^\top \Pi_1 - \Pi_D A_2^{-1} C^\top \Pi_2$$
$$= \Pi_D [A_1^{-1} - A_2^{-1}] C^\top \Pi_g$$

Thus, all we must prove that $\ker(\Pi_g) = \emptyset$, $A_2 \succ A_1$ is sufficient to show that $\ker(M) = \emptyset$. $\Pi_D$ and $A_g$ are positive definite by setting assumptions and $\ker(C) = \emptyset$ by Lemma A.1. So we consider $\ker(A_1^{-1} - A_2^{-1})$ and $\ker(\Pi_g)$. Clearly if $\ker(A_1^{-1} - A_2^{-1}) = \emptyset$ and $\ker(\Pi_g) = \emptyset$ then $\ker(\Pi_D [A_1^{-1} - A_2^{-1}] C^\top \Pi_g) = \emptyset$. Note that:

$$A_2 \succ A_1 \implies A_1^{-1} \succ A_2^{-1} \implies A_1^{-1} - A_2^{-1} \succ 0 \implies \ker(A_1^{-1} - A_2^{-1}) = \emptyset$$

$\qquad\square$

## A.3. Supplemental material for Section 4

### A.3.1. Supplemental material for Section Example 4.1

**Proposition A.9 ($\mathcal{W}(\beta) \in \mathcal{F}$ where $\mathcal{W}(\beta)$ defined by Example 4.1)** *$\mathcal{W}(\beta) \in \mathcal{F}$ where $\mathcal{W}(\beta)$ is as defined in Example 4.1 if and only if $\ker(\Pi_g) = \emptyset$ and $\beta \leq \lambda_d(M^\top M)$ where $M := \Pi_D A_g^{-1} C^\top \Pi_g$*

**Proof.**
$$\Delta(\mathbf{w}) = \|\Pi_D \mathbf{x}_e^{(g)}(\mathbf{w})\|_2^2 - \|\Pi_D \mathbf{x}_e^{(g')}(\mathbf{w})\|_2^2$$
$$= \langle \mathbf{w}, \mathbf{w} \rangle_{M_g^\top M_g} - \langle \mathbf{w}, \mathbf{w} \rangle_{M_{g'}^\top M_{g'}}$$

Where $M_g := \Pi_D A_g^{-1} C^\top \Pi_g$. Let $f(\mathbf{w}) := \langle \mathbf{w}, \mathbf{w} \rangle_{M_{g'}^\top M_{g'}}$ clearly $f(\mathbf{w}) \geq 0 \quad \forall \mathbf{w}$ and $f(\mathbf{0}) = 0$. Further, $f$ is Lipschitz on a finite space:

$$\|f(\mathbf{w}) - f(\mathbf{w}')\|_2 = \|\|M_{g'}^\top M_{g'} \mathbf{w}\|^2 - \|M_{g'}^\top M_{g'} \mathbf{w}'\|^2\|_2 = |(\mathbf{w} - \mathbf{w})^\top M_{g'}^\top M_{g'} (\mathbf{w} + \mathbf{w}')|$$
$$\leq \lambda_{\max}(M_{g'}^\top M_{g'}) \cdot \|\mathbf{w} - \mathbf{w}'\|_2 \|\mathbf{w} + \mathbf{w}'\|_2 \leq 2R\lambda_{\max}(M_{g'}^\top M_{g'})\|\mathbf{w} - \mathbf{w}'\|_2$$

where $R$ is the diameter of the Euclidean-Ball that contains the fairness space of Example 4.1.

Thus, because the above hold for any instantiation of Example 4.1, it is clearly necessary and sufficient by definition of $\mathcal{F}$ that $\beta \leq \lambda_d(M^\top M)$ and $M_g^\top M_g$ is PD.

To see the connection to $\ker(\Pi_g)$, Note that $M_g^\top M_g \succ 0 \iff \ker(M_g) = \emptyset$. Further note that in our setting, $\ker(\Pi_D A_g^{-1} C^\top) = \emptyset$ by definition and Lemma A.1, so all that matters is $\Pi_g$. Thus

$$M_g^\top M_g \succ 0 \iff \ker(M_g) = \emptyset \iff \ker(\Pi_g) = \emptyset$$

$\qquad\square$

A.3.2. SUPPLEMENTAL MATERIAL FOR SECTION 4'S CONVEX RESTRICTION CONSTRUCTION

Notice the principal's fair-constrained problem will generally become a nonconvex optimization problem when the fair space is defined as in $\mathcal{F}$. Thus, it becomes hard to reason about how much optimality is lost when the principal maximizes over the nonconvex feasible region rather than the $\mathcal{B}(1)$ that is the feasible region of the unconstrained problem. To get optimality loss bounds we will do the following:

1. Construct an ellipsoidal *restriction*, $\mathcal{E}(\beta)$, such that $\mathcal{E}(\beta) \subseteq \mathcal{B}(1) \cap \mathcal{W}(\beta; \Delta)$. This will give us optimality loss bounds for the $\beta$-fair SE as compared to the unconstrained SE.

2. Construct an ellipsoidal *envelope*, $\mathcal{E}(\tilde{\beta})$, such that $\mathcal{W}(\beta; \Delta) \subseteq \mathcal{E}(\tilde{\beta})$. This will give us optimality loss bounds for the convex restricted $\beta$-fair problem as compared to the nonconvex $\beta$-fair SE.

First let's construct the ellipsoidal restriction. We'll use a nice lemma about when ellipsoids are inside the unit euclidean ball along the way:

**Lemma A.9 (Necessary and Sufficient Conditions for ellipsoid in unit ball)** *Let* $\mathcal{E}(\beta) := \{\mathbf{w} \in \mathbb{R}^d : \mathbf{w}^\top Q \mathbf{w} \leq \beta\}$ *be a (centered at 0) ellipsoid.*

$\mathcal{E}(\beta) \subseteq \mathcal{B}(1)$ *iff* $\beta \leq \lambda_d(Q)$

**Proof.** Recall the Rayleigh quotient states that:

$$\lambda_d(Q)\|\mathbf{w}\|^2 \leq \mathbf{w}^\top Q \mathbf{w} \leq \lambda_1(Q)\|\mathbf{w}\|^2 \quad \forall \mathbf{w} \in \mathbb{R}^d$$

First we consider the direction $\mathcal{E}(\beta) \subseteq \mathcal{B}(1) \implies \beta \leq \lambda_d(Q)$. Let $\mathcal{E}(\beta) \subseteq \mathcal{B}(1)$. Let $\mathbf{w} = tz$ where $z$ is the unit eigenvector corresponding to $\lambda_d(Q)$. For $t$ such that $\mathbf{w} \in \mathcal{E}(\beta)$: $\mathbf{w}^\top Q \mathbf{w} = t^2 \lambda_d(Q) \leq \beta$. Now suppose for contradiction that $\beta > \lambda_d(Q)$. Then for $t = \sqrt{\frac{\beta}{\lambda_d(Q)}} > 1$ we have that $\mathbf{w}^\top Q \mathbf{w} = \frac{\beta \lambda_d(Q)}{\lambda_d(Q)} = \beta$ which means that $\mathbf{w} \in \mathcal{E}(\beta)$. But! $\|\mathbf{w}\|_2 = t\|z\| = t > 1$, which means that $\mathbf{w} \notin \mathcal{B}(1)$. This is a contradiction.

Now we consider the direction $\mathcal{E}(\beta) \subseteq \mathcal{B}(1) \impliedby \beta \leq \lambda_d(Q)$. This is direct from the Rayleigh quotient. For any $\mathbf{w} \in \mathcal{E}(\beta)$ we would have:

$$\lambda_d(Q)\mathbf{w}^\top \mathbf{w} \leq \mathbf{w}^\top Q \mathbf{w} \leq \beta \leq \lambda_d(Q)$$

Which clearly yields that $\mathbf{w}^\top \mathbf{w} \leq 1$, which means its in the euclidean ball. $\square$

Now here is the ellipsoidal restriction.

**Proposition A.10 ($\mathcal{E}(\beta)$, ellipsoidal restriction)** *If* $\mathcal{W}(\beta) \in \mathcal{F}$, *then* $\mathcal{E}(\beta) \subseteq \mathcal{W}(\beta) \cap \mathcal{B}(1)$ *Where* $\mathcal{E}(\beta) := \{\mathbf{w} : \mathbf{w} \in \mathbb{R}^d, \mathbf{w}^\top Q \mathbf{w} \leq \beta\}$

**Proof.** Clearly if $\mathbf{w} \in \mathcal{E}(\beta)$ then we have: $\mathbf{w}^\top Q \mathbf{w} \leq \beta$. But by assumption $\mathcal{F}$ we have $\Delta(\mathbf{w}) \leq \mathbf{w}^\top Q \mathbf{w} \leq \beta$ thus $\mathcal{E}(\beta) \subseteq \mathcal{W}(\beta)$.

For the last part, definition of $\mathcal{F}$ says $\beta \leq \lambda_d(Q)$, so by Lemma A.9, we have $\mathcal{E}(\beta) \subseteq \mathcal{B}(1)$ $\square$

Now, lets define the additional optimization problems over the restriction of the $\beta$-fair space.

**Definition A.1 (Convex restriction of the fairness constrained problem)**

$$\max_{\mathbf{w} \in \mathbb{R}^d} \text{OBJ}(\mathbf{w}; \mathbf{w}^\star), \quad \textit{subject to} \quad \mathbf{w} \in \mathcal{E}(\beta) \tag{13}$$

*We will frequently say that* $\mathbf{w}_{res}^\star$ *is the solution point to this problem.*

**Remark A.1** *Note that it would be provably redundant to include and intersection with $\mathcal{B}(1)$ on the restricted problem (Equation 13) because $\mathcal{E}(\beta) \subseteq \mathcal{B}(1)$ as shown by Proposition A.10. Thus, we leave this additional constraint off.*

This will lead us to the key bounds of the optimality loss in Stackelberg equilibria! Prior to presenting the specific bounds for accuracy and social welfare note that they will clearly look like this:

**Lemma A.10 (Bounds on optimality loss)** *If $\mathcal{W}(\beta; \Delta) \in \mathcal{F}$, then we can bound optimality loss as follows:*

$$\texttt{OBJ}(\mathbf{w}_u^\star) - \texttt{OBJ}(\mathbf{w}_c^\star) \leq \texttt{OBJ}(\mathbf{w}_u^\star) - \texttt{OBJ}(\mathbf{w}_{res}^\star)$$

*Where $\texttt{OBJ}(\mathbf{w}_u^\star)$ is the optimal value of Equation 3, $\texttt{OBJ}(\mathbf{w}_c^\star)$ is the optimal value of Equation 2, $\texttt{OBJ}(\mathbf{w}_{env}^\star)$ is the optimal value of Equation 14, and $\texttt{OBJ}(\mathbf{w}_{res}^\star)$ is the optimal value of Equation 13*

**Proof.** By Proposition A.10:

$$\texttt{OBJ}(\mathbf{w}_{res}^\star) \leq \texttt{OBJ}(\mathbf{w}_c^\star)$$

$\square$

**Proposition A.11 (Accuracy and social welfare optimality bounds)** *Assume that the $\beta$-fair space, $\mathcal{W}(\beta, \Delta)$ is such that $\mathcal{W}(\beta, \Delta) \in \mathcal{F}$ then letting $\texttt{OBJ}(\mathbf{w}_u^\star)$ be the $\beta$-fairness constrained optimal value (Equation 3) and $\texttt{OBJ}(\mathbf{w}_c^\star)$ be the unconstrained optimal value (Equation 2),*

*If the principal is accuracy maximizing, i.e., $\texttt{OBJ} = \texttt{ACC}$ then he has the following bounds on accuracy loss in stackelberg equilibrium:*

$$\texttt{ACC}(\mathbf{w}_u^\star) - \texttt{ACC}(\mathbf{w}_c^\star) \leq (2q)(t + s)$$

*where $q = \mathbb{I}_{\mathbf{w}^\star \notin \mathcal{E}(\beta)}(\sqrt{\frac{\beta}{\lambda_d(Q)}} + \|\mathbf{w}^\star\|)$, $s = \mathbb{I}_{\mathbf{w}^\star \notin \mathcal{E}(\beta)}(\min\{1, \|\mathbf{w}^\star\|\} + \sqrt{\frac{\beta}{\lambda_d(Q)}})$, and $t = 2\sqrt{\frac{\beta}{\lambda_d(Q)}}$*

*If the principal is Social Welfare (SW) maximizing, i.e., $\texttt{OBJ} = \texttt{SW}$ then he has the following bounds on SW loss in stackelberg equilibrium:*

$$\texttt{SW}(\mathbf{w}_u^\star) - \texttt{SW}(\mathbf{w}_c^\star) \leq \|\tilde{\mathbf{w}}\|_2 - \sqrt{\beta}\|\tilde{\mathbf{w}}\|_{Q^{-1}}$$

*Where $\tilde{\mathbf{w}} := (CA_1^{-1}C^\top \Pi_1 + CA_2^{-1}C^\top \Pi_2)^\top \mathbf{w}^\star$*

**Proof.** We begin with social welfare. This is easy because notice that by Proposition A.10, we have that the ellipsoidal restriction fairness problem of Equation 13 satisfies Property 3.3. Therefore we can invoke the bounds of Table 1 (equivalently, Prop A.4) thus we have:

$$\texttt{SW}(\mathbf{w}_u^\star) - \texttt{SW}(\mathbf{w}_{res}^\star) \leq \|\tilde{\mathbf{w}}\|_2 - \sqrt{\beta}\|\tilde{\mathbf{w}}\|_{Q^{-1}}$$

where $\texttt{SW}(\mathbf{w}_{res}^\star)$ is the optimal value of Equation 13. Invoking Lemma A.10, we have the RHS of SW of Proposition A.11.

Now we consider accuracy. This is also easy because we can again invoke the bounds of Table 1 (equivalently, Prop A.4) thus we have:

$$\texttt{ACC}(\mathbf{w}_u^\star) - \texttt{ACC}(\mathbf{w}_{res}^\star) \leq (2q)(t + s)$$

where $q = \mathbb{I}_{\mathbf{w}^\star \notin \mathcal{E}(\beta)}(\sqrt{\frac{\beta}{\lambda_d(Q)}} + \|\mathbf{w}^\star\|)$, $s = \mathbb{I}_{\mathbf{w}^\star \notin \mathcal{E}(\beta)}(\min\{1, \|\mathbf{w}^\star\|\} + \sqrt{\frac{\beta}{\lambda_d(Q)}})$, and $t = 2\sqrt{\frac{\beta}{\lambda_d(Q)}}$ and where $\texttt{ACC}(\mathbf{w}_{res}^\star)$ is the optimal value of Equation 13. Invoking Lemma A.10, we have the RHS of SW of Proposition A.11. $\square$

### A.3.3. SUPPLEMENTAL MATERIAL FOR SECTION 4'S CONVEX RESTRICTION TIGHTNESS

But of course, Proposition A.11 does not feel very satisfying because we have little idea of how much of the accuracy and social welfare loss in the bound comes from the restriction and how much of it comes from the actual tightness of the nonconvex $\beta$-fair space! From a computational perspective, we'd like to have a better idea of how good our restriction was. While we discuss these bounds as optimality loss in *equilibria*, perhaps the principal has the necessary setting information or reasonable estimates and wants to directly solves for (approximate) equilibrium and using this convex restriction to ensure a $\beta$-fair policy that satisfies tolerance. He will want to know how much optimality he might be losing as a result of making the problem (Equation 2) a tractable convex restriction.

To get upper bounds on this loss, we can construct an ellipsoidal envelope.

Before we do that, we should make use of an important fact. We can upper bound $f(\mathbf{w})$ on $\mathcal{W}(\beta; \Delta)$ because it is $L$-Lipschitz on this space and because we know that at $\mathbf{w} = \mathbf{0}$ $f = 0$. We'll also use the "diameter" of the fair space frequently:

**Definition A.2 (Diameter of a $\beta$-fair space)** *For some $\mathcal{W}(\beta; \Delta)$,*

$$D := \sup_{\mathbf{w} \in \mathcal{W}(\beta; \Delta)} \|\mathbf{w}\|_2$$

*is the "diameter"*

**Lemma A.11 (Upper bound on $f$)** *For any $\mathcal{W}(\beta; \Delta) \in \mathcal{F}$, we have that the value of $f$ is upperbounded on the fair space:*

$$\sup_{\mathbf{w} \in \mathcal{W}(\beta; \Delta)} f(\mathbf{w}) \leq LD \quad \forall \mathbf{w} \in \mathcal{W}(\beta; \Delta)$$

**Proof.** By definition of $L$-lipschitz, we have that:

$$|f(\mathbf{w}) - f(\hat{\mathbf{w}})| \leq L\|\mathbf{w} - \hat{\mathbf{w}}\|_2 \quad \forall \mathbf{w}, \hat{\mathbf{w}} \in \mathcal{W}(\beta, \Delta)$$

Thus by the assumption in $\mathcal{F}$, we have that $f(\mathbf{w}) \leq L\|\mathbf{w}\|_2 \quad \forall \mathbf{w} \in \mathcal{W}(\beta; \Delta)$ which further implies that

$$\sup_{\mathbf{w} \in \mathcal{W}(\beta; \Delta)} f(\mathbf{w}) \leq L \sup_{\mathbf{w} \in \mathcal{W}(\beta; \Delta)} \|\mathbf{w}\|_2 \quad \forall \mathbf{w} \in \mathcal{W}(\beta; \Delta)$$

$$= LD \quad \forall \mathbf{w} \in \mathcal{W}(\beta; \Delta)$$

$\square$

Now we are ready to define our ellipsoidal envelope:

**Proposition A.12 ($\mathcal{E}(\beta + LD)$, ellipsoidal envelope)** *If $\mathcal{W}(\beta) \in \mathcal{F}$, then $\mathcal{W}(\beta) \subseteq \mathcal{E}(\beta + LD)$ Where $\mathcal{E}(\beta + LD) := \{\mathbf{w} : \mathbf{w} \in \mathbb{R}^d, \mathbf{w}^\top Q \mathbf{w} \leq \beta + LD\}$.*

**Proof.** Let $\mathbf{w} \in \mathcal{W}(\beta; \Delta)$. Clearly: $\langle \mathbf{w}, \mathbf{w} \rangle_Q - f(\mathbf{w}) \leq \beta$ This implies that $\langle \mathbf{w}, \mathbf{w} \rangle_Q \leq \beta + f(\mathbf{w})$ And from Lemma A.11, we have:

$$\langle \mathbf{w}, \mathbf{w} \rangle_Q \leq \beta + LD \quad \forall \mathbf{w} \in \mathcal{W}(\beta; \Delta)$$

Thus we see that for $\mathbf{w} \in \mathcal{W}(\beta; \Delta)$, we also have that $\mathbf{w} \in \mathcal{E}(\beta + LD)$, which gives us the statement of the Proposition. $\square$

**Definition A.3 (Convex envelope of the fairness constrained problem)**

$$\max_{\mathbf{w} \in \mathcal{B}(1)} \texttt{OBJ}(\mathbf{w}; \mathbf{w}^\star), \quad subject\ to \quad \mathbf{w} \in \mathcal{E}(\beta + LD) \tag{14}$$

*We will frequently say that $\mathbf{w}_{env}^\star$ is the solution point to this problem.*

And now, because the nonconvex $\beta$-fair space lives right in-between the convex restriction and the envelope, the optimality loss that happens envelope→restriction *upper bounds* the optimality loss that happens $\mathcal{W}(\beta; \Delta)$ →restriction

**Lemma A.12 (Upper bound on loss from ellipsoidal restriction)** *If $\mathcal{W}(\beta; \Delta) \in \mathcal{F}$, then we can upper bound as follows:*

$$\texttt{OBJ}(\mathbf{w}_c^\star) - \texttt{OBJ}(\mathbf{w}_{res}^\star) \leq \texttt{OBJ}(\mathbf{w}_{env}^\star) - \texttt{OBJ}(\mathbf{w}_{res}^\star)$$

*Where $\texttt{OBJ}(\mathbf{w}_c^\star)$ is the optimal value of Equation 2, i.e. the nonconvex $\beta$-fair constrained problem*

**Proof.** By proposition A.12,

$$\texttt{OBJ}(\mathbf{w}_c^\star) \leq \texttt{OBJ}(\mathbf{w}_{env}^\star)$$

$\square$

Before we get into the main proposition we will make a useful lemma:

**Lemma A.13 (Projection onto convex set with origin gets closer to origin)** *For any closed, compact convex set $\mathcal{S} \subseteq \mathbb{R}^d$ such that $\mathbf{0} \in \mathcal{S}$ and any $\mathbf{w}^\star \in \mathbb{R}^d$, we have the following:*

$$\|P_{\mathcal{S}}(\mathbf{w}^\star)\|_2 \leq \|\mathbf{w}^\star\|_2$$

**Proof.** Let $\mathbf{p} := P_S(\mathbf{w}^\star)$ denote the Euclidean projection of $\mathbf{w}^\star$ onto $S$; it exists and is unique by the projection theorem. By first-order optimality for $\min_{\mathbf{z} \in S} \frac{1}{2}\|\mathbf{z} - \mathbf{w}^\star\|_2^2$,

$$\langle \mathbf{w}^\star - \mathbf{p}, \; \mathbf{z} - \mathbf{p} \rangle \leq 0 \qquad \forall \mathbf{z} \in S.$$

Since $0 \in S$, taking $\mathbf{z} = 0$ yields $\langle \mathbf{w}^\star, \mathbf{p} \rangle \geq \|\mathbf{p}\|_2^2$. By Cauchy–Schwarz,

$$\|\mathbf{p}\|_2^2 \leq \langle \mathbf{w}^\star, \mathbf{p} \rangle \leq \|\mathbf{w}^\star\|_2 \|\mathbf{p}\|_2.$$

If $\mathbf{p} = 0$ we are done; otherwise divide by $\|\mathbf{p}\|_2$ to obtain $\|\mathbf{p}\|_2 \leq \|\mathbf{w}^\star\|_2$. Hence $\|P_S(\mathbf{w}^\star)\|_2 \leq \|\mathbf{w}^\star\|_2$. $\qquad\square$

**Proposition A.13 (Accuracy and social welfare loss from ellipsoidal restriction)** *If $\mathcal{W}(\beta; \Delta) \in \mathcal{F}$. Then we have the following upper bounds on optimality loss between the nonconvex fairness constrained problem (Equation 2) and the ellipsoidal restriction of Equation 13:*

$$|\mathrm{ACC}(\mathbf{w}_c^\star) - \mathrm{ACC}(\mathbf{w}_{res}^\star)| \leq (2c)(a + e) \quad |\mathrm{SW}(\mathbf{w}_c^\star) - \mathrm{SW}(\mathbf{w}_{res}^\star)| \leq \min\left\{\|\tilde{\mathbf{w}}\|_2, \sqrt{\beta + LD}\|\tilde{\mathbf{w}}\|_{Q^{-1}}\right\} - \sqrt{\beta}\|\tilde{\mathbf{w}}\|_{Q^{-1}}$$

*where*

$$a := \mathbb{I}_{\mathbf{w}^\star \notin \mathcal{E}(\beta)}\left(\sqrt{\frac{\beta}{\lambda_d(Q)}} + \min\left\{1, \|\mathbf{w}^\star\|, \sqrt{\frac{\beta + LD}{\lambda_d(Q)}}\right\}\right)$$

$$c := \mathbb{I}_{\mathbf{w}^\star \notin \mathcal{E}(\beta)}\left(\sqrt{\frac{\beta}{\lambda_d(Q)}} + \|\mathbf{w}^\star\|\right)$$

$$e := 2\sqrt{\frac{\beta}{\lambda_d(Q)}}$$

$$\tilde{\mathbf{w}} := (CA_1^{-1}C^\top\Pi_1 + CA_2^{-1}C^\top\Pi_2)^\top\mathbf{w}^\star$$

**Proof.** Note that throughout this proof, we will set $\tilde{\beta} = \beta + LD$ for conciseness. It is useful to keep in mind that $\mathcal{E}(\beta)$ is the *restricted* ellipsoid and $\mathcal{E}(\tilde{\beta})$ is the *envelope* (larger) ellipsoid!

**We will start with the accuracy claim.** This is a more intricate version of the Proposition A.5 proof.

First let $P_{\mathcal{E}(\beta)}(\mathbf{w})$ be the projection of $\mathbf{w}$ onto $\mathcal{E}(\beta)$. Further, let $s_{\max} := \max_{\mathbf{w} \in \mathcal{E}(\tilde{\beta})} \|\mathbf{w}\|_2$ Note that this is the largest $l2$ norm a $\mathbf{w}$ can have and still be potentially $\mathbf{w} \in \mathcal{E}(\tilde{\beta})$. Also, let $s_{\max}' := \max_{\mathbf{w} \in \mathcal{E}(\beta)} \|\mathbf{w}\|_2$ Note that this is the largest $l2$ norm a $\mathbf{w}$ can have and still be potentially $\mathbf{w} \in \mathcal{E}(\beta)$.

The following bound on the distance between the optimal envelope-constrained point and its projection onto the restriction ellipsoid will be useful:

**Claim A.1**
$$\|P_{\mathcal{E}(\beta)}(\mathbf{w}_{env}^\star) - \mathbf{w}_{env}^\star\|_2 \leq \mathbb{I}_{\mathbf{w}^\star \notin \mathcal{E}(\beta)}\left(s_{\max}' + \min\left\{1, s_{\max}, \|\mathbf{w}^\star\|\right\}\right) \tag{15}$$

**Proof of Claim A.1.** Note that this comes form the fact that we have:

$$\|P_{\mathcal{E}(\beta)}(\mathbf{w}_{env}^\star) - \mathbf{w}_{env}^\star\|_2 \leq \|P_{\mathcal{E}(\beta)}(\mathbf{w}_{env}^\star)\| + \|\mathbf{w}_{env}^\star\| \tag{triangle ineq}$$

And we can pedantically do case analysis:

CASE 1: $\mathbf{w}^\star \in \mathcal{E}(\beta)$

$\|P_{\mathcal{E}(\beta)}(\mathbf{w}_{env}^\star) - \mathbf{w}_{env}^\star\|_2 = 0$

CASE 2: $\mathbf{w}^\star \in \mathcal{E}(\tilde{\beta}), \in \mathcal{B}(1), \notin \mathcal{E}(\beta)$

If $\mathbf{w}^\star$ is already in the ball and the envelope ellipsoid, then Opt problem of Equation 14 doesn't do anything and the optimal point $\mathbf{w}^\star_{env} = \mathbf{w}^\star$.

$$\|P_{\mathcal{E}(\beta)}(\mathbf{w}^\star_{env})\| + \|\mathbf{w}^\star_{env}\| = \|P_{\mathcal{E}(\beta)}(\mathbf{w}^\star_{env})\| + \|\mathbf{w}^\star\|$$
$$= \|P_{\mathcal{E}(\beta)}(\mathbf{w}^\star_{env})\| + \min\{\|\mathbf{w}^\star\|, 1, s_{\max}\} \qquad \text{(by case assumption)}$$

CASE 3: $\mathbf{w}^\star \in \mathcal{B}(1), \notin \mathcal{E}(\tilde{\beta})$

If $\mathbf{w}^\star$ is already in the ball, but not yet in the envelope ellipsoid, then $\mathbf{w}^\star \neq \mathbf{w}^\star_{env}$, but of course $\mathbf{w}^\star_{env}$ must be $\in \mathcal{E}(\tilde{\beta})$ and $\in \mathcal{B}(1)$, thus it must be be that $\|\mathbf{w}^\star_{env}\| \leq 1, s_{\max}$ Additionally, by Lemma A.13, we have $\|\mathbf{w}^\star_{env}\| \leq \|\mathbf{w}^\star\|$

$$\|P_{\mathcal{E}(\beta)}(\mathbf{w}^\star_{env})\| + \|\mathbf{w}^\star_{env}\| \leq \|P_{\mathcal{E}(\beta)}(\mathbf{w}^\star_{env})\| + \min\{s_{\max}, \|\mathbf{w}^\star\|, 1\}$$

CASE 4: $\mathbf{w}^\star \in \mathcal{E}(\tilde{\beta}), \notin \mathcal{B}(1), \notin \mathcal{E}(\beta)$

This means $\mathbf{w}^\star$ is in the envelope ellipsoid, but not in the ball (or the restriction ellipsoid). $\mathbf{w}^\star \neq \mathbf{w}^\star_{env}$ because recall that the feasible region of the envelope problem is the *intersection* of the ball and the envelope, but of course $\mathbf{w}^\star_{env}$ must be $\in \mathcal{E}(\tilde{\beta})$ and $\in \mathcal{B}(1)$, thus it must be be that $\|\mathbf{w}^\star_{env}\| \leq 1, s_{\max}$ Additionally, by Lemma A.13, we have $\|\mathbf{w}^\star_{env}\| \leq \|\mathbf{w}^\star\|$

$$\|P_{\mathcal{E}(\beta)}(\mathbf{w}^\star_{env})\| + \|\mathbf{w}^\star_{env}\| \leq \|P_{\mathcal{E}(\beta)}(\mathbf{w}^\star_{env})\| + \min\{s_{\max}, \|\mathbf{w}^\star\|, 1\} \qquad \text{(by case assumption)}$$

CASE 5: $\mathbf{w}^\star \notin \mathcal{E}(\tilde{\beta}), \notin \mathcal{B}(1), \notin \mathcal{E}(\beta)$

$\mathbf{w}^\star \neq \mathbf{w}^\star_{env}$, but of course $\mathbf{w}^\star_{env}$ must be $\in \mathcal{E}(\tilde{\beta})$ and $\in \mathcal{B}(1)$, thus it must be be that $\|\mathbf{w}^\star_{env}\| \leq 1, s_{\max}$ Additionally, by Lemma A.13 we have $\|\mathbf{w}^\star_{env}\| \leq \|\mathbf{w}^\star\|$

$$\|P_{\mathcal{E}(\beta)}(\mathbf{w}^\star_{env})\| + \|\mathbf{w}^\star_{env}\| \leq \|P_{\mathcal{E}(\beta)}(\mathbf{w}^\star_{env})\| + \min\{s_{\max}, \|\mathbf{w}^\star\|, 1\} \qquad \text{(by case assumption)}$$

Finally notice that $P_{\mathcal{E}(\beta)}(\mathbf{w}^\star_{env}) \in \mathcal{E}(\beta)$ thus $\|P_{\mathcal{E}(\beta)}(\mathbf{w}^\star_{env})\| \leq s'_{\max}$

This completes the proof for Claim A.1 $\qquad\qquad\square$

Now we will use the bounds of Claim A.1 to create another useful bound:

**Claim A.2**

$$\|\mathbf{w}^\star_{res} - \mathbf{w}^\star_{env}\|_2 \leq \mathbb{I}_{\mathbf{w}^\star \notin \mathcal{E}(\beta)}(s'_{\max} + \min\{1, s_{\max}, \|\mathbf{w}^\star\|\}) + \mathbb{I}_{\mathbf{w}^\star \notin \mathcal{E}(\tilde{\beta}) \cap \mathcal{B}(1)} \max\{(\|\mathbf{w}^\star\| - s_{\min})_+, (\|\mathbf{w}^\star\|_2 - 1)_+\}$$

**Proof of Claim A.2.**

$$\|\mathbf{w}^\star_{res} - \mathbf{w}^\star_{env}\|_2 = \|P_{\mathcal{E}(\beta)}(\mathbf{w}^\star) - \mathbf{w}^\star_{env}\|_2 \qquad\qquad \text{(definition)}$$
$$\leq \|P_{\mathcal{E}(\beta)}(\mathbf{w}^\star_{env}) - \mathbf{w}^\star_{env}\|_2 + \|P_{\mathcal{E}(\beta)}(\mathbf{w}^\star_{env}) - P_{\mathcal{E}(\beta)}(\mathbf{w}^\star)\|_2 \qquad\qquad \text{(triangle ineq)}$$
$$\leq \|P_{\mathcal{E}(\beta)}(\mathbf{w}^\star_{env}) - \mathbf{w}^\star_{env}\|_2 + 2s'_{\max} \qquad\qquad \text{(diameter of } \mathcal{E}(\beta))$$
$$\leq \mathbb{I}_{\mathbf{w}^\star \notin \mathcal{E}(\beta)}(s'_{\max} + \min\{1, s_{\max}, \|\mathbf{w}^\star\|\}) + 2s'_{\max} \qquad\qquad \text{(Claim A.1)}$$

$\qquad\qquad\square$

Finally, we will need one more bound.

**Claim A.3** $\|\mathbf{w}^\star_{res} - \mathbf{w}^\star\|_2 \leq \mathbb{I}_{\mathbf{w}^\star \notin \mathcal{E}(\beta)}(s'_{\max} + \|\mathbf{w}^\star\|)$

**Proof of Claim A.3.**

$$\|\mathbf{w}^\star_{res} - \mathbf{w}^\star\|_2 \leq \mathbb{I}_{\mathbf{w}^\star \notin \mathcal{E}(\beta)}(\|P_{\mathcal{E}(\beta)}(\mathbf{w}^\star)\| + \|\mathbf{w}^\star\|) \qquad\qquad \text{(triangle ineq, } \mathcal{E}(\beta) \in \mathcal{B}(1))$$
$$\leq \mathbb{I}_{\mathbf{w}^\star \notin \mathcal{E}(\beta)}(s'_{\max} + \|\mathbf{w}^\star\|)$$

□

And finally using Lemma A.2 and Claims A.1, A.2, and A.3:

$$
\begin{aligned}
|\mathrm{ACC}(\mathbf{w}^\star_{res}) - \mathrm{ACC}(\mathbf{w}^\star_{env})| &= |-\|\mathbf{w}^\star_{res} - \mathbf{w}^\star\|^2_2 + \|\mathbf{w}^\star_{env} - \mathbf{w}^\star\|^2_2| \\
&= |(\|\mathbf{w}^\star_{res} - \mathbf{w}^\star\|_2 + \|\mathbf{w}^\star_{env} - \mathbf{w}^\star\|_2)(\|\mathbf{w}^\star_{res} - \mathbf{w}^\star\|_2 - \|\mathbf{w}^\star_{env} - \mathbf{w}^\star\|_2)| \\
&\hspace{7cm} (a^2 - b^2 = (a+b)(a-b)) \\
&= (\|\mathbf{w}^\star_{res} - \mathbf{w}^\star\|_2 + \|\mathbf{w}^\star_{env} - \mathbf{w}^\star\|_2)|(\|\mathbf{w}^\star_{res} - \mathbf{w}^\star\|_2 - \|\mathbf{w}^\star_{env} - \mathbf{w}^\star\|_2)| \quad \text{(nonneg factor)} \\
&\leq (\|\mathbf{w}^\star_{res} - \mathbf{w}^\star\|_2 + \|\mathbf{w}^\star_{env} - \mathbf{w}^\star\|_2)(\|\mathbf{w}^\star_{res} - \mathbf{w}^\star_{env}\|_2) \quad \text{(reverse triangle ineq)} \\
&\leq (2\|\mathbf{w}^\star_{res} - \mathbf{w}^\star\|_2)(\|\mathbf{w}^\star_{res} - \mathbf{w}^\star_{env}\|_2) \quad\quad (\mathcal{E}(\beta) \subseteq \mathcal{E}(\tilde{\beta})) \\
&\leq (2c)(a + e) \quad\quad\quad \text{(Claims A.1, A.3, and A.2)}
\end{aligned}
$$

where $a := \mathbb{I}_{\mathbf{w}^\star \notin \mathcal{E}(\beta)} (s'_{\max} + \min\{1, s_{\max}, \|\mathbf{w}^\star\|\})$, $c := \mathbb{I}_{\mathbf{w}^\star \notin \mathcal{E}(\beta)}(s'_{\max} + \|\mathbf{w}^\star\|)$, and $e = 2s'_{\max}$

Finally, Lemma A.7 gives the solutions for $s_{\max}, s'_{\max}$. Taking note of Lemma A.12 yields the bounds that are in the Proposition for accuracy

**We move on to the social welfare claim.**

We want to find an upper bound on: $\mathrm{SW}(\mathbf{w}^\star_{env}) - \mathrm{SW}(\mathbf{w}^\star_{res})$ And then invoke Lemma A.12. This will turn out to be rather easy. We will simply present an upper bound on $\mathrm{SW}(\mathbf{w}^\star_{env})$ and then a closed form for $\mathrm{SW}(\mathbf{w}^\star_{res})$.

NOTE: By Lemma A.3 recall that social welfare maximization is functionally a linear objective. Because the constants of $\mathrm{SW}(\mathbf{w}^\star_{env})$ will cancel out with $-\mathrm{SW}(\mathbf{w}^\star_{res})$ we will just ignore them for this whole proof.

**Claim A.4** $\mathrm{SW}(\mathbf{w}^\star_{env}) \leq \min\left\{\|(CA_1^{-1}C^\top\Pi_1 + CA_2^{-1}C^\top\Pi_2)^\top\mathbf{w}^\star\|_2, \sqrt{\tilde{\beta}}\|(CA_1^{-1}C^\top\Pi_1 + CA_2^{-1}C^\top\Pi_2)^\top\mathbf{w}^\star\|_{Q^{-1}}\right\}$

**Proof of Claim A.4.** By Lemma A.3 recall that social welfare maximization is functionally a linear objective. Specifically: $\langle\mathbf{c}, \mathbf{w}\rangle$ where $\mathbf{c} := (CA_1^{-1}C^\top\Pi_1 + CA_2^{-1}C^\top\Pi_2)^\top\mathbf{w}^\star$

Clearly, because the envelope constrained problem (Equation 14) for Social welfare is linear maximization over the convex space: $\mathcal{E}(\tilde{\beta}) \cap \mathcal{B}(1)$, it will be the case that

$$
\mathrm{SW}(\mathbf{w}^\star_{env}) \leq \min\{\mathrm{SW}(\mathbf{w}^\star_u), \mathrm{SW}(\mathbf{w}^\star_{env'})\}
$$

Where $\mathbf{w}^\star_{env'}$ is the solution to the following problem:

$$
\max_{\mathbf{w}\in\mathbb{R}^d} \mathrm{SW}(\mathbf{w}; \mathbf{w}^\star), \quad \text{subject to} \quad \mathbf{w} \in \mathcal{E}(\tilde{\beta}) \tag{16}
$$

From Lemma A.6 we have solutions for these problems and substituting into $\langle\mathbf{c}, \mathbf{w}\rangle$ where $\mathbf{c} := (CA_1^{-1}C^\top\Pi_1 + CA_2^{-1}C^\top\Pi_2)^\top\mathbf{w}^\star$ We get:

$$
\mathrm{SW}(\mathbf{w}^\star_u) = \|(CA_1^{-1}C^\top\Pi_1 + CA_2^{-1}C^\top\Pi_2)^\top\mathbf{w}^\star\|_2
$$

$$
\mathrm{SW}(\mathbf{w}^\star_{env'}) = \sqrt{\tilde{\beta}}\|(CA_1^{-1}C^\top\Pi_1 + CA_2^{-1}C^\top\Pi_2)^\top\mathbf{w}^\star\|_{Q^{-1}}
$$

□

Now we can calculate the closed form for the restriction ellipsoid in the closed form.

**Claim A.5** $\mathrm{SW}(\mathbf{w}^\star_{res}) = \sqrt{\beta}\|(CA_1^{-1}C^\top\Pi_1 + CA_2^{-1}C^\top\Pi_2)^\top\mathbf{w}^\star\|_{Q^{-1}}$

**Proof of Claim A.5.** By Lemma A.3 recall that social welfare maximization is functionally a linear objective. Specifically: $\langle\mathbf{c}, \mathbf{w}\rangle$ where $\mathbf{c} := (CA_1^{-1}C^\top\Pi_1 + CA_2^{-1}C^\top\Pi_2)^\top\mathbf{w}^\star$ From Lemma A.6 we have solution and substituting into $\langle\mathbf{c}, \mathbf{w}\rangle$ where $\mathbf{c} := (CA_1^{-1}C^\top\Pi_1 + CA_2^{-1}C^\top\Pi_2)^\top\mathbf{w}^\star$ We get:

$$
\mathrm{SW}(\mathbf{w}^\star_{res}) = \sqrt{\beta}\|(CA_1^{-1}C^\top\Pi_1 + CA_2^{-1}C^\top\Pi_2)^\top\mathbf{w}^\star\|_{Q^{-1}}
$$

□

Using Lemma A.12, we get the social welfare claim in the Proposition.

$\square$

So what can we make of all these bounds? It's important to notice that the bounds of Propositions A.11 and A.13 look very similar! Are we sure that the restriction loss bounds of Proposition A.13 are telling us anything more than what Proposition A.11 already told us? After all, the true optimality loss between unconstrained SE and the ellipsoidal restriction clearly upper bounds the loss between $\beta$-fair SE and the ellipsoidal restriction, so we'd want Proposition A.13 to be stricter than Proposition A.11, otherwise it is clearly (too) loose. Turns out, in many cases, Proposition A.13 is *tighter* and thus telling us something "better" about how much the restriction hurts the principal than what we could glean from Proposition A.11. Formally:

**Proposition A.14 (Sufficient conditions for Prop A.13 to be more informative than Prop A.11)** *Given the $\beta$-fairness space is such that $\mathcal{W}(\beta; \Delta) \in \mathcal{F}$, if the principal convexifies his $\beta$-fair problem using the ellipsoidal restriction problem of Equation 13 then as long as the following hold:*

1. *Ground-truth policy is not already in the restricted ellipsoid (i.e., $\mathbf{w}^\star \notin \mathcal{E}(\beta)$)*

2. *Envelope ellipsoid is strictly inside the unit ball (i.e., $LD < \lambda_d(Q) - \beta$, see Lemma A.9)*

3. *Ground truth policy is strictly outside the ball envelope of the envelope ellipsoid (i.e., $\|\mathbf{w}^\star\| > \sqrt{\frac{\beta + LD}{\lambda_d(Q)}}$, see Lemma A.7)*

*The principal gets a (strictly) tighter bound on restriction optimality loss using Proposition A.13 rather than Proposition A.11.*

**Proof.** We will just compare the bounds promised between the two propositions under conditions.

CONDITION SET: $\mathbf{w}^\star \notin \mathcal{E}(\beta)$, $\beta + LD < \lambda_d(Q)$, and $\|\mathbf{w}^\star\| > \sqrt{\frac{\beta + LD}{\lambda_d(Q)}}$

Thus for accuracy bounds we can simplify the upper bounds given by the propositions.

RHS of Proposition A.11 says:

$$2\left(\sqrt{\tfrac{\beta}{\lambda_d(Q)}} + \|\mathbf{w}^\star\|\right)\left(2\sqrt{\tfrac{\beta}{\lambda_d(Q)}} + \sqrt{\tfrac{\beta}{\lambda_d(Q)}} + \min\{1, \|\mathbf{w}^\star\|\}\right)$$

While RHS of Proposition A.13 says:

$$2\left(\sqrt{\frac{\beta}{\lambda_d(Q)}} + \|\mathbf{w}^\star\|\right)\left(2\sqrt{\frac{\beta}{\lambda_d(Q)}} + \sqrt{\frac{\beta}{\lambda_d(Q)}} + \sqrt{\frac{\beta + LD}{\lambda_d(Q)}}\right)$$

Clearly, Proposition A.13 is tighter!

Now for social welfare:

Here, before we apply bounds, we should recall an important fact. By Lemma A.9, $\beta + LD < \lambda_d(Q) \implies \mathcal{E}(\beta + LD) \subseteq \mathcal{B}(1)$! This means the envelope ellipsoid is (strictly) inside the ball. Therefore, the social welfare of the envelope ellipsoid problem (equation 14) is strictly less than the unconstrained problem (equation 3). Formally:

$$\|\tilde{\mathbf{w}}\|_2 > \sqrt{\beta + LD}\|\tilde{\mathbf{w}}\|_{Q^{-1}}$$

Now we can see that:

RHS of Proposition A.11 says:

$$\|\tilde{\mathbf{w}}\|_2 - \sqrt{\beta}\|\tilde{\mathbf{w}}\|_{Q^{-1}}$$

While RHS of Proposition A.13 says:

$$\sqrt{\beta + LD}\|\tilde{\mathbf{w}}\|_{Q^{-1}} - \sqrt{\beta}\|\tilde{\mathbf{w}}\|_{Q^{-1}}$$

Clearly, Proposition A.13 is tighter!

$\square$

## A.4. Supplemental material for Section 5

### A.4.1. Supplemental material for experiment set-up

We set the parameters $C, \Pi_1, \Pi_2, A_1, A_2, \Pi_D$, and $\mathbf{w}^\star$ as follows.

**Causal graph.** We follow prior work on recourse/causal modeling for the `Adult` dataset (Kugelgen et al., 2022; Nabi & Shpitser, 2018; Chiappa, 2019) (see their cited sources for the Structural Causal Model (SCM)) and instantiate an 8-node acyclic causal graph with nodes {sex, age, western, married, edu-num, workclass, occupation, hours}. Edge weights are sampled as non–negative values on the existing edges (respecting a topological order so the adjacency is strictly upper–triangular), yielding a weighted adjacency matrix $A \in \mathbb{R}^{8 \times 8}$ and the corresponding contribution matrix $C$.

**Groups and projectors.** Following (Bechavod et al., 2022), we form three sets of different 2 group "splits". Groups sets are as follows:

- *Age* ($\leq 35$ vs. $> 35$),

- *Country* (western world vs. other)

- *Education* ($\geq$ high-school vs. $<$ high-school).

For each split $g \in \{1, 2\}$. We build a projection matrix $\Pi_g \in \mathbb{R}^{d \times d}$ by running SVD on the data points belonging to group $g$, taking the top $k$ right singular vectors ($k = 5$), and setting $\Pi_g = V_{g,k} V_{g,k}^\top$.

**Cost-matrices.** We analyse 3 distinct cases for the costs matrices.

1. We use uniform costs $A_1 = A2$ with $A_1 = I$

2. We use non-uniform costs with $A_2 = 2A_1$ and $A_1 = I$

3. We use non-uniform $A_2 = 2A_1$ random cost matrices ($A_1$ is sampled at random).

Their use is denoted in figure captions.

**Desirability.** We choose *education*, *occupation*, and *workclass* as **desirable**, since these attributes are realistic to improve and likely to have downstream, external effects outside of income. Thus, to external entities (e.g. government bodies) they should be desirable to incentivize. We specifically use for $\Pi_D$ a diagonal matrix with 1's in the coordinates that correspond to desirable features and zeros everywhere else.

**Ground truth rule.** We train a logistic-regression classifier on the ADULT dataset restricted to the eight variables that correspond to the nodes of our SCM (see above). Let $\tilde{\mathbf{w}} \in \mathbb{R}^8$ denote the learned coefficient vector. To make the comparisons, we project this vector onto $\tilde{\mathbf{w}}$ the unit $\ell_2$ ball, let $\mathbf{w}^\star$ denote the projection. This normalization ensures the baseline is bounded for the social welfare problem while at the same time yields scale-invariant results. Following our theory, our *accuracy loss* reports $\|\mathbf{w} - \mathbf{w}^\star\|_2^2$ (See Lemma A.2), and our *social welfare* uses the linear utility $u(\mathbf{w}) = \mathbf{c}^\top \mathbf{w}$ with $\mathbf{c}$ defined in Lemma A.3. Constrained learners are optimized under the $\ell_1$-$\beta$-desirability fairness constraint $\|M_g \mathbf{w}\|_1 \leq \beta$ (see example 3.1).

It is worth noticing that the unconstrained optimum for accuracy is the same for all the 3 splits (pair of groups). That is natural since the objective does depend on group characteristics. However, for the social-welfare we have 3 distinct unconstrained optimal values since the objective function (see Lemma A.3) does depend on the specific matrices of each group.

### A.4.2. Supplemental material for Section 5.1

In this part we present the experiments using the $\ell_2$-fairness desirability constraint (as defined in Example 3.2).

The results for $\ell_2$ remain consistent with the results for $\ell_1$ with the only noticeable difference being that the optimal accuracy is reached at a much faster rate (smaller value of $\beta$). This can be explained by the fact that $\|M\mathbf{w}\|_2 \leq \|M\mathbf{w}\|_1$ and therefore the $\ell_2$ fairness constraint is more relaxed that the $\ell_1$-fairness.

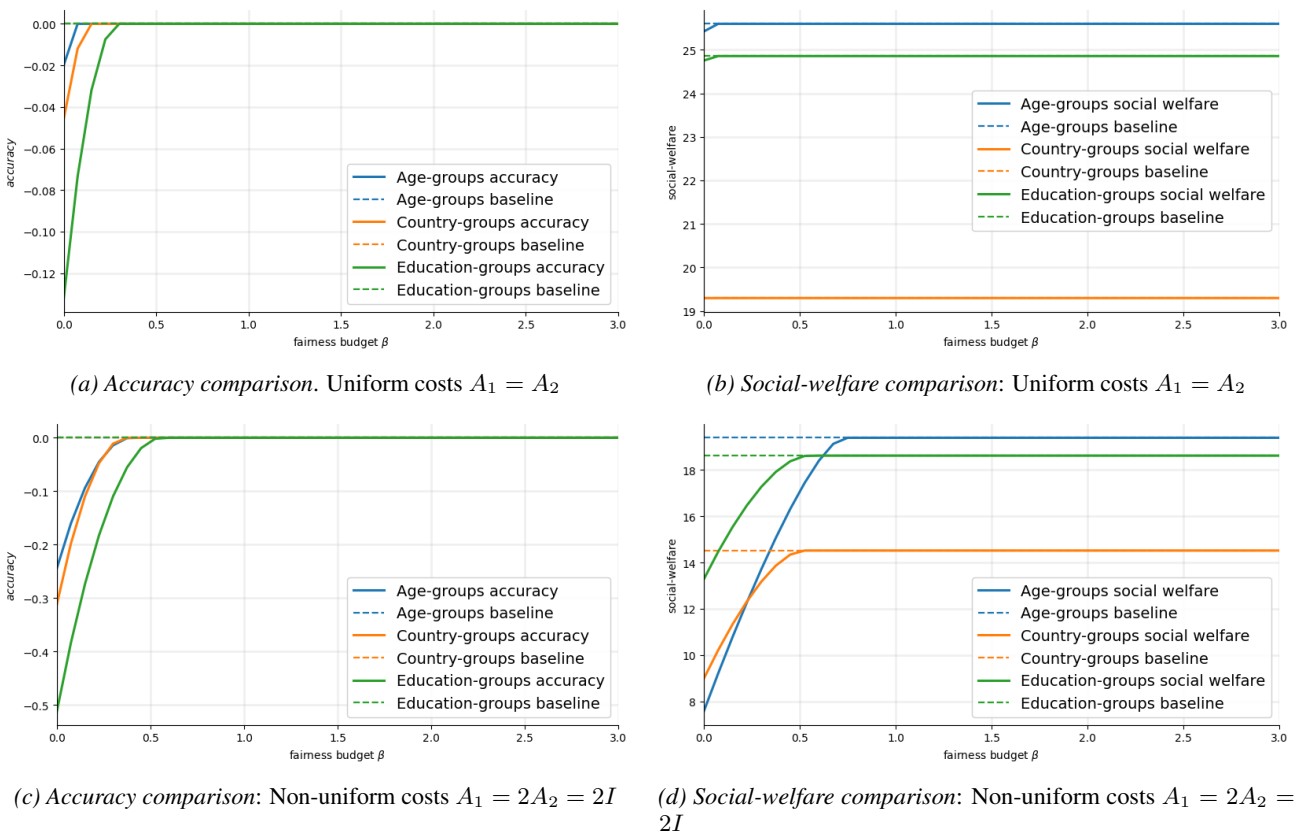

*(a) Accuracy comparison*. Uniform costs $A_1 = A_2$

*(b) Social-welfare comparison*: Uniform costs $A_1 = A_2$

*(c) Accuracy comparison*: Non-uniform costs $A_1 = 2A_2 = 2I$

*(d) Social-welfare comparison*: Non-uniform costs $A_1 = 2A_2 = 2I$

*Figure 6.* Optimal values for varying $\beta$ for different group splits under $\ell_2$-fairness constraint

Interestingly, when we allow the cost matrices to be more complex than the unit (e.g random) then information disparities become more irrelevant. However, that's not suprising considering the fact that the fairness-desirability function values equally the desirability matrix and the projection matrices of the groups.

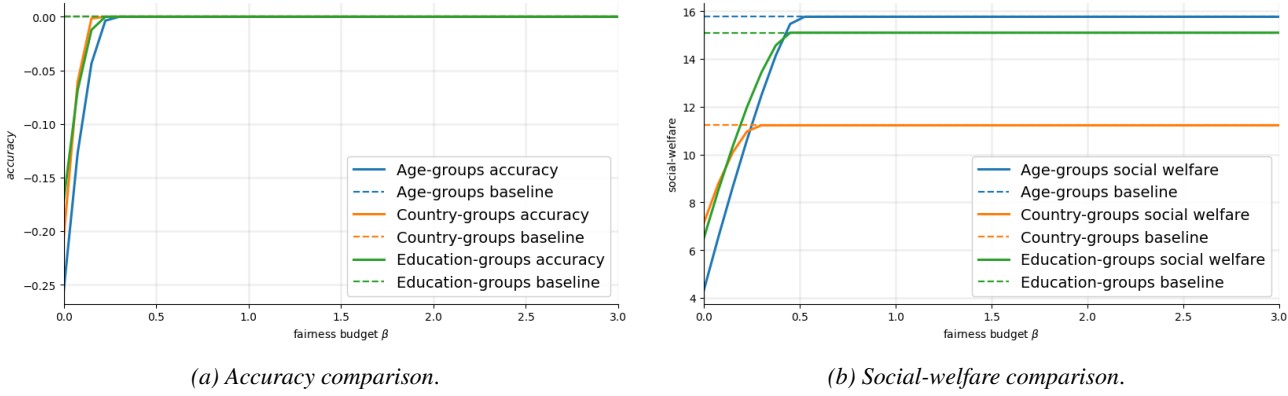

*(a) Accuracy comparison.*

*(b) Social-welfare comparison.*

*Figure 7.* Optimal values for varying $\beta$ for different group splits under non-uniform random cost matrices $A_1 = 2A_2$ and the $\ell_1$-fairness constraint

## A.5. Supplemental Material for the `Taiwan` Experiment

### A.5.1. SETUP

We now describe the experimental setup for the second dataset. The construction follows the `Adult` experiment, with one additional goal: we compute the theoretical upper bounds from Table 1 and compare them with the realized accuracy and social-welfare losses.

**Dataset.** We use the `Taiwan` credit-default dataset (see Yeh (2009)), which contains financial and demographic information about credit-card clients. The prediction task is to identify clients who are likely to default on repayment. We use this dataset as a second real-world instance for evaluating how conservative our theoretical bounds are in practice.

**Causal graph.** To model the interactions among the engineered variables, we use a fixed directed acyclic graph adapted from the causal structure proposed by Pitso & Michael (2026). The graph contains six feature nodes together with the outcome node `DEFAULT_PROB`. The variables `PAY_HISTORY`, `BILL_PAY`, `AGE`, `EDU`, and `SEX_MAR` are taken to influence `LIMIT_BAL`, reflecting the idea that repayment behavior and demographic characteristics may affect the credit limit assigned to a client. These variables, together with `LIMIT_BAL` itself, are then allowed to directly influence the default outcome. As a result, the effect of repayment and demographic variables on default may occur both directly and indirectly through the assigned credit limit.

**Groups and projectors.** Following the same construction as in the `Adult` experiment, we study three two-group population splits derived from the original raw dataset:

- *Age*: younger versus older clients,

- *Education*: higher-education versus lower-education clients,

- *Marriage*: married versus not-married clients.

Each split induces a corresponding pair of projection matrices $\Pi_1, \Pi_2$. For each group $g \in \{1, 2\}$, we construct $\Pi_g$ by restricting the data to the points belonging to group $g$, computing the top $k$ principal directions of that subsample, and setting

$$\Pi_g = V_{g,k} V_{g,k}^\top,$$

where $V_{g,k}$ contains the top $k$ right singular vectors. In our experiments we take $k = 5$. Thus, $\Pi_g$ represents the dominant empirical variation subspace of group $g$.

**Cost matrices.** We consider several cost models in order to study how cost heterogeneity affects both the realized losses and the tightness of the theoretical bounds:

1. *Uniform costs*: $A_1 = A_2 = I$.

2. *Non-uniform costs*: $A_1 = 2A_2 = 2I$, so one group faces twice the cost of the other group.

3. *Random SPD costs*: $A_1$ and $A_2$ are sampled as symmetric positive definite matrices.

Their use is indicated in the corresponding figure captions.

**Desirability.** We encode the desirable feature directions using a projection matrix $\Pi_D$. In particular we set : $D = \{$`PAY_HISTORY`, `BILL_PAY`, `EDU`$\}$,. These directions represent the coordinates that the principal or an external decision-maker wishes to incentivize. While this choice is modeling-dependent, it is a reasonable one for the Taiwan-Credit setting: these variables capture aspects of repayment behavior, billing/payment status, and education that are plausibly relevant to credit worthiness. As in the main experiment, $\Pi_D$ is taken to be a diagonal projection matrix, with ones on the coordinates corresponding to desirable features and zeros elsewhere.

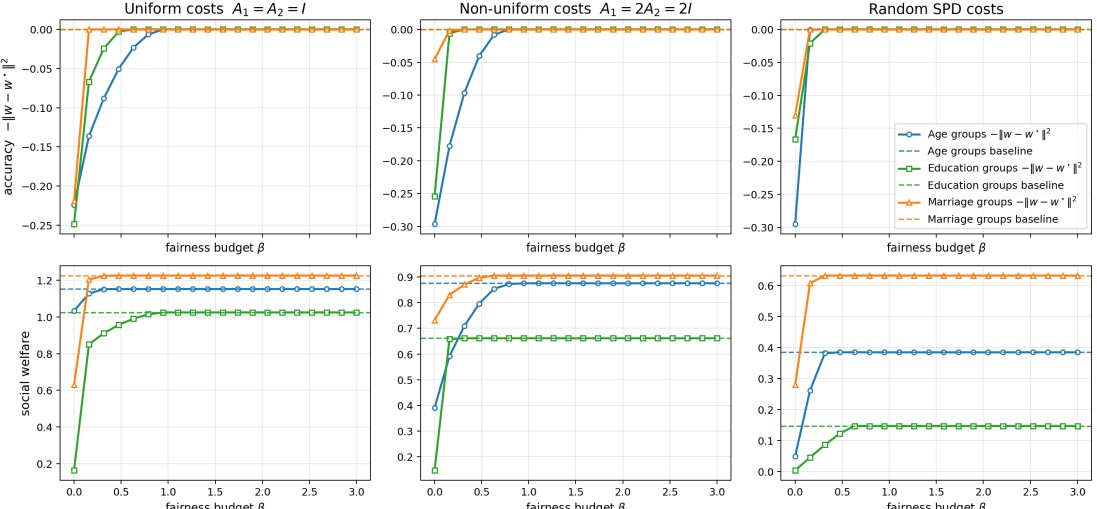

*Figure 8.* Realized losses versus theoretical upper bounds on the `Taiwan` dataset under $\ell_1$ fairness constraints. Columns correspond to cost models: uniform costs $A_1 = A_2 = I$, non-uniform costs $A_1 = 2A_2 = 2I$, and random SPD costs. The first row reports the realized accuracy loss, while the second row reports the realized social-welfare loss. Each column corresponds to a different cost configuration. Solid curves show empirical losses, while dashed curves show the corresponding upper bounds from Table 1. Colors denote the population partition: age, education, and marital status.

**Ground-truth rule and unconstrained benchmark.** As in the previous experiment, the accuracy objective is $f(\mathbf{w}) = -\|\mathbf{w} - \mathbf{w}^\star\|_2^2$, where $\mathbf{w}^\star$ is the unconstrained maximizer of the accuracy objective; in particular, $f(\mathbf{w}^\star) = 0$ (see Lemma A.2). Following the maximization convention used in the theory, we report the signed accuracy change $L_{\mathrm{acc}}(\beta) = f(\mathbf{w}_c^\star) - f(\mathbf{w}_u^\star) = -\|\mathbf{w}_c^\star - \mathbf{w}_u^\star\|_2^2 \leq 0$, where $\mathbf{w}_c^\star$ denotes the optimizer under the $\ell_1$-$\beta$-desirability fairness constraint and $\mathbf{w}_u^\star$ is the unconstrained accuracy optimizer. In this real-world dataset it is $w_u^\star = \mathbf{w}^\star$.

For social welfare, we use the linear objective $h(\mathbf{w}) = \mathbf{c}^\top \mathbf{w}$, with $\mathbf{c}$ defined in Lemma A.3. In the plots, we report the positive welfare loss relative to the unconstrained welfare benchmark, $L_{\mathrm{SW}}(\beta) = h(\mathbf{w}_u^\star) - h(\mathbf{w}_c^\star) \geq 0$, where $\mathbf{w}_u^\star \in \arg\max_{\|\mathbf{w}\|_2 \leq 1} h(\mathbf{w})$. Constrained learners are optimized under the $\ell_1$-$\beta$-desirability fairness constraint

$$\|M_g \mathbf{w}\|_1 \leq \beta$$

(see Example 3.1).

**Theoretical upper bounds.** For each experimental configuration, we also compute the corresponding theoretical upper bounds from Table 1. This allows us to compare the realized losses with the guarantees predicted by the theory.

### A.5.2. EXPERIMENTAL RESULTS

In Section 5, we evaluated the realized accuracy and welfare losses that arise when imposing $\beta$-desirability constraints across different group partitions, cost matrices, and disparity configurations. Since Section 3 provides theoretical upper bounds on these losses, summarized in Table 1, we now compare the empirical losses on the TAIWAN dataset with the corresponding theoretical guarantees.

The bounds in Table 1 are worst-case guarantees. Although some of them depend on instance-specific parameters, they are not intended to exactly predict the loss on every dataset. Thus, even when a bound is valid, it may be conservative if the empirical instance does not realize the worst-case geometry or cost configuration considered by the theory. This motivates the following question: for the TAIWAN dataset, are the theoretical bounds tight, highly conservative, or somewhere in between?

**Discussion of Results.** As it is shown in Figure 4, all three group definitions recover the unconstrained optimum relatively quickly as the fairness budget $\beta$ increases. Thus, the separation across population splits is weaker than in the `Adult` experiment. To better understand this behavior, we examine the geometry of the split-specific projection matrices.

Among the three splits, the *Age* partition is the most internally aligned: its two group-specific subspaces have the largest overlap,

$$\frac{\mathrm{tr}(\Pi_1 \Pi_2)}{k} \approx 0.93,$$

compared with approximately 0.80 for both the education and marriage splits. The age split also has the smallest subspace gap,

$$\|\Pi_1 - \Pi_2\|_F \approx 0.84, \qquad \|\Pi_1 - \Pi_2\|_2 \approx 0.59,$$

whereas for education and marriage we obtain

$$\|\Pi_1 - \Pi_2\|_F \approx 1.41, \qquad \|\Pi_1 - \Pi_2\|_2 = 1.$$

At the same time, the age split exhibits the strongest alignment with the desirability subspace $\Pi_D$:

$$\frac{\mathrm{tr}(\Pi_1 \Pi_D)}{3} \approx 0.998, \qquad \frac{\mathrm{tr}(\Pi_2 \Pi_D)}{3} \approx 0.869.$$

Taken together, these diagnostics suggest that the age split is strongly coupled to the prediction-relevant geometry while also exhibiting substantial overlap between its two group-specific subspaces. This helps explain why, even when the unconstrained solution $\mathbf{w}^\star$ is not initially feasible, the principal can recover performance quickly by choosing a feasible classifier that focuses on shared predictive directions across the two groups.

On the welfare side, the ranking is stable across all three cost models: *Marriage* achieves the highest welfare, followed by *Age*, and then *Education*. A plausible explanation is that the marriage split induces the weakest effective fairness restriction. Thus, enforcing fairness across marital status requires less deviation from the welfare-optimal unconstrained solution, whereas enforcing fairness across education is more restrictive and yields lower welfare. As expected, the absolute welfare levels still depend on the cost model, with more uniform costs yielding higher welfare overall.

Finally, when interpreting the tightness of the bounds in Table 1, it is important to distinguish between bounds that are uniform and bounds that depend on instance-specific parameters. Except for the accuracy bound in Property 3.2, the bounds depend on the group-specific projection matrices, cost matrices, and desirability directions. Consequently, each split induces its own theoretical upper bound and may have a different worst-case instance at which the bound is tight. For example, the same pair of cost matrices can induce different worst-case realized losses for the age split than for the marriage split, because the corresponding projection matrices define different feasible regions and interact differently with $\Pi_D$.

This distinction is reflected in the experiments. Bounds that adapt to the geometry of the $\beta$-fair feasible region track the realized accuracy losses more closely as $\beta$ varies. By contrast, bounds that do not encode this geometry, especially some of the social-welfare bounds, remain more conservative. These bounds are still valid worst-case certificates, but they are not expected to be numerically tight for every empirical partition.

