# OpenReview forum: "Desirable Effort Fairness and Optimality Trade-offs in Strategic Learning"
_ICML.cc/2026/Conference — ICML 2026 regular_

### Official Review · Reviewer_nd4H · 2026-03-13

**Soundness:** 3
**Presentation:** 3
**Significance:** 2
**Originality:** 3
**Overall Recommendation:** 4
**Confidence:** 3

**Summary:**

This paper looks at fairness in strategic classification, where individuals may change their features in response to incentives created by a deployed model. The authors point out that much of the existing literature treats strategic behavior primarily as gaming the system and focuses heavily on predictive accuracy. In contrast, in many real-world settings some feature changes (for example, improving education or work experience) may actually represent meaningful improvements that a system designer might want to encourage.

To explore this idea, the paper proposes a framework that studies trade-offs between predictive performance, social welfare, and what the authors call desirable effort fairness. The interaction is modeled as a Stackelberg game between a principal and agents from two groups (Section 2). Agents can modify their features by exerting effort, and those changes may propagate across features through a causal contribution matrix that captures spillover effects between features. Rather than observing the classifier directly, agents estimate it through a peer-learning process based on observations of others in their group.

The framework also introduces an external stakeholder who specifies which features are considered desirable to improve and defines a discrepancy function that measures inequity in the desirable effort induced by the classifier. The principal then chooses a decision rule subject to a fairness tolerance parameter that limits this discrepancy. The paper derives theoretical bounds on the performance loss incurred by imposing these fairness constraints under different assumptions about the geometry of the discrepancy function (discussed in Section 3 and summarized in Table 1). Finally, the authors run experiments on the Adult dataset to illustrate how varying the fairness tolerance affects predictive accuracy and social welfare across different group partitions.

Overall, my takeaway is that the paper aims to formalize how fairness constraints on incentivized effort interact with strategic behavior, and to characterize the resulting trade-offs between fairness and performance.

**Compliance With Llm Reviewing Policy:**

Affirmed.

**Final Justification:**

The rebuttal provides useful clarification of the paper’s motivation, particularly around modeling fairness in incentives rather than outcomes, and the additional experimental results help address concerns about generality. These responses improve my understanding of the contribution and resolve my main concerns. The response reaffirms my original score.

**Key Questions For Authors:**

1. The theoretical results in Section 3 derive upper bounds on optimality loss as a function of the fairness tolerance $\beta$ (summarized in Table 1). In practice, how tight are these bounds when evaluated on real datasets such as the Adult dataset used in the experiments?

2. The paper introduces a convex restriction approach for handling nonconvex discrepancy functions in Section 4. How sensitive are the resulting guarantees to the choice of restriction, and are there settings where the restricted feasible region significantly alters the optimal solution?

3. Could the experimental evaluation be extended beyond the Adult dataset to assess whether the observed fairness–performance trade-offs generalize across other datasets or domains?

4. How sensitive are the results to the stakeholder-defined desirability scores and discrepancy functions used in the framework? For example, do small changes in these desirability weights significantly change the optimal policy?

**Limitations:**

- The empirical evaluation focuses primarily on a single dataset and a limited set of experimental configurations.

- The framework assumes linear decision rules and particular cost structures for feature changes.

- The process by which desirability scores and discrepancy functions would be specified in real applications is not discussed in much detail.

**Strengths And Weaknesses:**

### Strengths

- I think the paper raises an interesting perspective within the strategic classification literature. Instead of focusing purely on prediction outcomes or error rates, it looks at fairness in the incentives created by the model, which seems like a meaningful angle in settings where individuals can respond to algorithmic decisions.

- The notion of desirable effort fairness is conceptually appealing. Distinguishing between harmful gaming and genuinely beneficial improvements feels like a useful framing for thinking about fairness when people adapt their behavior in response to algorithms.

- I also liked that the modeling framework tries to bring together several elements that are often studied separately. In particular, the model combines imperfect knowledge of the classifier through peer learning (Section 2.2), causal spillovers between features through the contribution matrix, and stakeholder-defined notions of desirable improvements.

- The theoretical analysis of fairness–optimality trade-offs is fairly detailed. The results summarized in Table 1, which relate fairness tolerance $\beta$ to upper bounds on accuracy or welfare loss, help illustrate how the geometry of the feasible region affects the constrained optimization problem.

- I also appreciated that the paper distinguishes between convex discrepancy functions (Section 3) and the more challenging nonconvex case (Section 4), where the authors introduce a convex restriction of the feasible set to obtain tractable guarantees.

- It is also nice that the paper attempts to connect the theory with experiments on the Adult dataset under several group partitions and cost structures.

---

### Weaknesses

- One issue I struggled with is understanding how much the paper adds beyond existing work in strategic classification and fairness. Many of the individual components in the framework (causal feature relationships, imperfect knowledge of the classifier, and fairness constraints) have appeared in prior work. While combining them is interesting, I was not fully convinced that this integration leads to fundamentally new insights beyond the resulting optimization formulation.

- The empirical evaluation felt somewhat limited to me. The experiments are all based on the Adult dataset with a small number of group partitions (age, country, and education). While these examples help illustrate the framework, it is hard to tell how general the observed patterns are across other datasets or domains.

- I also felt that the link between the theoretical analysis and the experiments could be stronger. For example, the paper derives bounds on optimality loss in the $\beta$-fair equilibrium (Table 1), but the experimental section mainly reports realized accuracy and welfare values rather than examining whether these theoretical bounds meaningfully predict empirical behavior.

---

> ### Author Rebuttal · Authors · 2026-03-31
>
> We thank the reviewer for their recognition of the novelty of our desirable effort fairness framing. We address weaknesses (W1-3), questions (Q1-4), and limitations (L1-3), if we do not answer any remaining concerns, we are happy to engage in the dicussion period.
>
> **W1** We understand the reviewer's concern as we discuss the novelty of component combination a lot. However, our biggest insights come from the novel study of *guaranteeing* equitable incentives on specific features. Inducing disparate incentives often leads to lower societal welfare and can perpetuate existing disparity. Trade-off characterizations implicitly answer: Is it realistic for principal to avoid this without being compensated? Consider a loan approval algorithm incentivizing only already high-earning people to keep credit utilization low, while incentivizing less well-off people to strategically time applications (e.g. right after credit score update), this is a poor distribution of incentives for societal improvement because the already well-off group is more desirably incentivized. Our work formalizes trade-offs of policy deployments guaranteed to avoid this.
>
> **W2, Q3, L1** We included an evaluation of only one dataset because in ADULT, agents are particularly disparate across groups, meaning that the enforcement of desirable incentive fairness would be especially important to characterize as any inequity in incentives could perpetuate disparities (W1). We ran additional experiments on the Taiwan-Credit dataset (https://anonymous.4open.science/r/Blind_2-BCFA/blind_experiments%20(1).pdf) with the same groupings along features. As in Sec 5 Fig 1, we see that, as $\beta$ increases, the fairness constraint is less binding and loss is less impacted meaning that the shape of trade-off is similar. However, groupings do not yield very disparate sets of agents (Frobenius norms of group matrices are similar), so though this is a setting in which trade-off exists, its magnitude and the effect of different groupings are less stark. For the revision we will reference them in an the Appendix
>
> **W3: theory/experiment link** We agree that the theory and experiments focus on different angles, but this is purposeful. In existing lit, "What happens (for the principal) when equitable (desirable) incentives are guaranteed?" is unclear. Our theoretical bounds present a worst-case POV, but a practitioner would want more information, particularly about what group parameters, i.e. $A_g, \Pi_g$ mean for loss. For example, as Reviewer ue6z points out: $A_g,\Pi_g$ may need to be estimated. Because Sec 4 highlights that Acc loss may be worse when imposing fairness given a parameter matrix is correlated with desirability, the practitioner's takeaway is to carefully approximate any matrix that may be correlated with desirability. As the theory results are upper bounds, this could not be gleaned from Table 1 alone (line 97-100 column 2). Overall, we intend Sec 5 to be a supplement to our paper's larger purpose.
>
> **Q1, W3: application of bounds to actual loss** This is a good question. For space, we point the reviewer to W1 for reviewer tP79.
>
> **Q2** The question of how sensitive are results to slightly different $\beta$-fair spaces is reasonable. We take the space defined by $\Delta(\mathbf{w})= \mathbf{w}^\top Q\mathbf{w} - f(\mathbf{w})\leq \beta$, meaning that the $\beta$-fairspace is a core ellipsoid with some extra space (Figure 2 and line 357-9 column 2), and cut off $f(\mathbf{w})$, so that it is just the core ellipsoid, $\mathcal{E}(\beta)$. Thus, $\Delta(\mathbf{w})=\mathbf{w}^\top Q\mathbf{w} - f(\mathbf{w})$ and $\Delta'(\mathbf{w})=\mathbf{w}^\top Q\mathbf{w} - f'(\mathbf{w})$, which would define different $\beta$ fairness spaces, have the same restriction. If $f'(\mathbf{w}) > f(\mathbf{w})$, it is possible that applying Table 1 bounds to $\mathcal{E}(\beta)$ will give looser Acc and SW loss bounds for $\Delta'(\mathbf{w})$ than $\Delta(\mathbf{w})$ because bounds are the same, but optimal fair policies may be different. However, when we bound $V = |OBJ(w_c) - OBJ(w_{res})|$, which bounds accuracy lost as a result of using $\mathcal{E}(\beta)$ (line 349 col 2) the result is sensitive to different $\Delta, \Delta'$ as long as $f$ and $f'$ sufficiently differ.
>
> **Q4** Fair optimal solutions are most impacted (at least for the Acc objective) when groups are split along (un)desirability (Sec 5, Fig 3c,a). Thus, changing desirability scores in a way that (un)correlates them with group parameters will likely have significant impact.
>
> **L2** This is a standard assumption in strategic classification lit.
>
> **L3** Realistically, what is desirable might vary highly across domains. A general protocol a practitioner could fall back on: for any feature improvement that is "good" in the eyes of the stakeholder (e.g. government body), he assigns desirability score 1. To all others, $\epsilon$. A natural discrepancy function is either l1 or l2 norm (ex 3.1 and 2).

---

> > ### Author Rebuttal · Reviewer_nd4H · 2026-04-02
> >
> > I appreciate the authors’ detailed responses and the additional clarifications. The discussion helped better motivate the focus on fairness in incentives rather than just outcomes, and the example provided makes this perspective clearer. I also appreciate the additional results on the Taiwan-Credit dataset and the clarification of the experimental setup.
> >
> > Overall, the rebuttal improves my understanding of the scope and contribution of the work and addresses my main concerns. While the empirical evaluation still feels somewhat limited in breadth, I continue to view the paper as a technically solid contribution on fairness in strategic classification with practical relevance.

---

### Official Review · Reviewer_ue6z · 2026-03-13

**Soundness:** 3
**Presentation:** 3
**Significance:** 2
**Originality:** 3
**Overall Recommendation:** 4
**Confidence:** 3

**Summary:**

This paper presents a framework for strategic classification where principals may wish to incentivize changes in the agents' features fairly across heterogeneous agent groups. This is framed as an optimization problem for the principal, with constraints pertaining to the fairness of the changes across groups. The authors consider both convex and non-convex constraints, and provide bounds on the losses in accuracy and social welfare upon introducing the fairness constraint. In their experiments, they vary the fairness parameter $\beta$ and demonstrate the resulting accuracy and social welfare values on a real-world dataset.

**Compliance With Llm Reviewing Policy:**

Affirmed.

**Key Questions For Authors:**

1. (Clarification) Is accuracy and welfare loss the difference between the objectives for the unconstrained and constrained problems? Although the meaning of loss here can be inferred, it doesn't seem to be clearly defined in the paper.
2. Is it possible to estimate some of these quantities ($C$, $A_g$, $\Pi_g$) from observed data? Which among these would be harder to estimate or need to be known beforehand? Which of these are the most critical in terms of estimation?
3. How realistic is it to assume that the agents are really solving the optimization problem in Equation (1)? $\mathbf{w}_{\text{est}}(g) = \Pi_g \mathbf{w}$ seems to be a consequence of choosing the weight vector with the smallest $\ell_2$ norm; however, might it not be the case that the learned weight vectors are inherently noisy and depend on the number of peers a particular agent has? How might that error propagate through to the accuracy and SW losses?
4. Can the polyhedral and ellipsoidal results from Section 3 also be extended to more than two subpopulations? If so, what would be some examples of discrepancy functions? Can Examples 3.1 and 3.2 be readily modified for multiple subpopulations?

**Limitations:**

Yes

**Strengths And Weaknesses:**

**Strengths**

- The paper is well-written and the problem is well-framed. While the model itself is simple and linear, it encodes intuitions about causal effects of feature alterations, peer learning, and fairness constraints in an effective manner.
- The objectives (accuracy and social welfare) and the fairness constraints (sum of absolute value/squared differences) are well-considered, and the analysis for loss bounds seems sound.
- The experiments clearly show the effect of alignment between desirable attributes and group disparity.

**Weaknesses**
- While the bounds in Section 3 are provided in more generic terms ($M$ and $Q$ matrices), it would be insightful to connect them explicitly to the two examples of discrepancy functions. This would provide further intuition on what these constraints really look like. Currently, these bounds are situated more independently from the rest of the paper.
- For Section 4, the convex restriction seems to result in bounds for $|OBJ(w_c) - OBJ(w_{res})|$; however, this isn't connected to $OBJ(w_u)$, the unconstrained objective, that clearly.

---

> ### Author Rebuttal · Authors · 2026-03-31
>
> We thank the reviewer for their note of our work's writing quality, theory soundness, and experimental clarity. Below we address weaknesses (W1-2) and questions (Q1-4). We are happy to discuss futher.
>
> **W1.** We agree that the intuition about Table 1's $M$ and $Q$ is unclear. Intuitively, $M$ and $Q$ are generalizations of l1 and l2 norm fairness constraints (Ex 3.1 and 3.2). In A.2.3-4 we reformulate 3.1 and 3.2 into polyhedrons, $M= \Pi_D(A_1^{-1}C^\top\Pi_1 - A_2^{-1}C^\top\Pi_2)$, and ellipsoids, $Q= M^\top M$, respectively. Generically, $M$ and $Q$ are like reweightings of l1 and l2 norms. In the revision we will explain this in the main body.
>
> **W2.** We apologize for the confusing presentation of Sec 4 bounds. To get bounds on Acc/SW loss due to fairness (i.e.,$U=|OBJ(w_c) - OBJ(w_{u})|$ ), apply Table 1 row 4 to the ellipsoidal restriction, $\mathcal{E}(\beta)$ (lines 360-4). Because $\mathcal{E}(\beta)$ is in the $\beta$ fair space, $\mathcal{W}(\beta)$, this yields: $|OBJ(w_{{res}}) - OBJ(w_{u})|> U$. Results on $V = |OBJ(w_c) - OBJ(w_{res})|$ are not to bound Acc/SW loss due to fairness constraints. $V$ is Acc or SW lost were the principal to use the convex restriction to deploy a fair, easy-to-compute policy instead of the *optimal* fair policy. To understand the trade-off of convexification, we bound on $V$, but we agree that it is a sudden new concept. In the revision, we will better describe the new interpretation.
>
> **Q1.** Accuracy loss is the absolute difference between the optimal values of Eqns 2 and 3 when $OBJ=$ accuracy (line 255) and SW loss is when $OBJ=$ Social Welfare (line 259). We will add these as formal definitions.
>
> **Q2.** $\Pi_g$ is a group projection matrix of the feature PDF for agents (line 191-2). PDFs can be reasonably estimated with enough $\mathbf{x}_{i}$ datapoints. Manipulation cost, $A_g$, can be estimated from a dataset or experiment where a learner has original features and updated features after agents are told of a classifier they'd like to pass. $C$ can be found by causal estimation tools given $\Pi_g, A_g$. Most critical to estimate well are matrices correlated with desirability. Fairness constraints are impactful on accuracy when groups are split along (un)desirability (Sec 5, Fig 3c,a). For example, if groups are highly/poorly educated, there are only high loss policies that induce equal incentives to further improve education across groups since one is already educated, but one needs more.
>
> **Q3.** We agree that reasoning for agents' behavior is unclear. We will add the following:
>
> **Outcome Noise**. If agents see noisy (mean zero) outcomes from $N_g$ others, i.e., $y_{g,i}$ = $\mathbf{w}^\top x_{g,i} + \epsilon$ where $\mathbb{E}[\epsilon] = 0$, as $N_g$ gets large, the constraint to eqn 1 has approximately all the same solutions, so $\mathbf{w}_{est}\rightarrow\Pi_g\mathbf{w}$.
>
> **Behavioral meaning of eqn 1.** The constraint is an ordinary least-squares (OLS). In linear regression (an agent's setting if $\mathbf{X}_g$, the "design" matrix representing all $N_g$ observed features, is full column rank), OLS guarantees unbiasedness and asymptotic normality. Thus, OLS is rationally estimating a $\mathbf{w}$ assuming data spans the space. Because OLS may not have a unique solution, we need a tie-break. Norm minimization tie-breaking's behavioral interpretation is that agents are risk-averse. Smaller weights ensure that feature changes will be small. When agents are uncertain, they guess $\mathbf{w}$ that has less impact on them, just in case. Thus, eqn 1 = rational and risk-averse agents (line 196 col 2)
>
> **$\mathbf{w}_{est}$ noise.** If $\mathbf{w}_{est}(g) = \Pi_g\mathbf{w} + \epsilon$ and a principal implements a "fair" policy (without accounting for noise) then loss and realized fairness are very impacted if error is correlated with group and/or desirability. Consider a group who underestimates coefficients. They will be under-incentivized: the principal believes a policy encourages s amount of improvement while agents believe only s' is enough. If another group correctly estimates, then a policy the principal thinks is fair will be unfair. Formal study of this is an interesting future work.
>
> **Q4.** We can use the same discrepancy function and add a constraint for every pairwise group discrepancy. For Ex 3.1, the $\beta$-fair space would be the intersection of l1 norms: $\sum\limits_{i \in [d]} |(\Pi_D \mathbf{x}_e^{(g)}(\mathbf{w})-\Pi_D \mathbf{x}_e^{(g')}(\mathbf{w}))_i|\leq \beta$ $\forall g, g'$. Now, a policy is fair if the difference between every group's induced desirable effort is small. The intersection between polyhedra is a polyhedron, so our bounds on polyhedra still apply ($M$ must be computed as the polyhedral intersection). An ellipsoidal intersection is not necessarily ellipsoidal, but still convex, so Table 1 row 1 would apply to Ex 3.2. We will add this extension to sec 6.

---

> > ### Author Rebuttal · Reviewer_ue6z · 2026-04-03
> >
> > I thank the authors for their response. I maintain my positive score of the paper.

---

### Official Review · Reviewer_bRyY · 2026-03-13

**Soundness:** 4
**Presentation:** 4
**Significance:** 3
**Originality:** 3
**Overall Recommendation:** 5
**Confidence:** 3

**Summary:**

This paper studies the strategic classification problem, and is does so on a model that unifies different extensions of the classic strategic ML problem: (1) there is a lack of full information by the agents gaming the algorithm (knowledge of the classifier is acquired through peer learning), (2) there are causal relations between feature changes, and (3) the goals of the system designer extend beyond predictive accuracy (including a focus on desirable changes as prescribed by an external stakeholder, and limiting disparities between protected groups in these induced changes). The analysis and numerical experiments explore accuracy loss (and welfare loss) at different levels of desired effort fairness and under different ways of measuring the fairness gaps.

**Compliance With Llm Reviewing Policy:**

Affirmed.

**Final Justification:**

The paper provides a comprehensive study with a new focus on how to assess disparities in strategic learning. I maintain a positive score.

**Key Questions For Authors:**

1. This paper combines extensions of the classical strategic machine learning problem. Can you comment on how the results (say, accuracy loss) compare to the results obtained if each of these had been introduced in isolation or not at all? For instance, how much of these losses are driven by the agents’ peer learning (or, what would these results look like under full information)? Similarly, what if there are no causal effects between features?

&nbsp;

2. There is a note in the impact statement that the work helps understand “how informational disparities across groups affect equilibrium responses” – could you elaborate more on this?

&nbsp;

3. Do agents not have a budget when best-responding in Proposition 2.1? Does that matter for the findings?

**Limitations:**

Yes discussed

**Strengths And Weaknesses:**

**Strengths**

1. The study of the proposed model is comprehensive.

2. The idea of wanting to incentivize effort fairly between groups is new, to my understanding. Some works look at incentivizing effort and some look at fairness (mostly about outcomes, occasionally about effort like Milli et. al’s assessment of social burden differences), but it is a different (and interesting angle) to make sure incentive interventions are fair/balanced across groups, and this paper offers that, by additionally presenting different (convex and nonconvex) functions of how, exactly, to assess this disparity.


**Weaknesses**

1. A stated contribution in Section 1.1 is that the paper develops a game-theoretic framework that unifies three previously isolated modeling considerations. It is at times unclear how much each element influences (and is important for) the obtained results.

---

> ### Author Rebuttal · Authors · 2026-03-31
>
> We thank the reviewer for their rating of accept and their recognition of our novel focus on what happens when the principal commits to a policy that guarantees equitable (desirable) incentives. As we detail in response to Reviewer nd4H Weakness 1, inequitable incentives could reinforce existing group disparities or have other negative downstream effects, so we believe the characterization of the principal's trade-offs is an important step in understanding when inequitable incentives could be realistically avoided. We respond to the weakness (W1) and questions (Q1-3) posed by the reviewer. We hope that our answers sufficiently address these points and we are happy to further discuss any remaining concerns.
>
> **W1.** This is a reasonable concern and the primary reason we include Sec 5. While our framework and bounds enable the computation of worst-case losses for many modeling considerations, meaning that the principal can see how they will affect him in aggregate, because they are *upper* bounds, a theoretical analysis of how each piece affects the bound would not be very fruitful given that the actual effect may be less in good cases (lines 97-101). In Sec 5 we see that it is not necessarily certain elements (e.g., $\Pi_g$) *consistently* affecting accuracy loss more/less, but that accuracy is particularly affected by any elements along which groups are disparate in a way that is correlated with (un)desirability (Sec 5, Fig 3c,a). For example, if groups are separated by a high/low education feature ($\Pi_g$ is correlated with desirable feature: education), there are only high loss policies that induce equal incentives to further improve education across groups since one is already educated, but one needs more.
>
> **Q1.** We intentionally build our results/model to go through even if no such effects exist. Any element that we want to omit, can be encoded as a unit matrix. In particular, if we don't want to give features individual desirabilites, $\Pi_D=I$ models this (each feature i has desirability score $des(i)=1$). Similarly, if we want to study how our results change under no information or cost discrepancies, we can set $\Pi_g = I$ and or $A_g=I$. Under full information, agents no longer need to estimate the deployed $\mathbf{w}$, they just best-respond to $\mathbf{w}$, note $\Pi_g=I, \mathbf{w}_\text{est}(g) = \Pi_g \mathbf{w} = \mathbf{w}$. Absence of causal effects between features can be similarly encoded as $C=I$ (effort on feature i affects only feature i). We will be sure to add a remark on this topic to Section 2, so that the generality is clear.
>
> **Q2.** Recall that $\Pi_g$ are the group projection matrices of feature PDFs (line 191). Thus, they represent the feature space a particular group has access to. An agent can only learn what $\mathbf{w}_{est}$ is from $N_g$ other agents in her group (eqn 1). Hence, an agent in group $g$ can only learn about the principal's policy $\mathbf{w}$ in the subspace that agents from her own group live, $\Pi_g$. Thus, when $\Pi_1 \neq \Pi_2$, this is _informational disparity_ because the info agents learn about $\mathbf{w}$ and thus their best responses are different. To see a concrete effect, fix the cost discrepancies to zero ($A=A_1=A_2$). In equilibrium, agents best-respond: $\mathbf{x}_e^{(g)} = A^{-1} C^\top \Pi_g \mathbf{w}$ (Proposition 2.1). As $\Pi_1$ becomes more similar to $\Pi_2$, group 1's best-response feature change becomes more similar to group 2's. We can see the same effect for the principal: looking at examples 3.1, 3.2 or 4.1, if $\Pi_1 = \Pi_2$ then $\Delta(\mathbf{w})= 0$ (under $A_1 = A_2$). In other words, the information discrepancies between the two groups more heavily constrain the equilibrium choice $\mathbf{w}$ of the principal as they grow and similarly are less important when they are minimal.
>
> **Q3.** In short, our agents have budgets, but rather than a fixed value, the implicit budget is the score an agent gets as a result of manipulation. This is standard in strat class lit, e.g. "Strategic Classification from Revealed Preferences", Dong et al. To build intuition: consider a 0,1 strategic *classification* (we instead use linear scores). Normalizing an agent's utility from positive classification to 1, an agent is willing to spend up to 1 in feature manipulations to get over the classification boundary $0\rightarrow 1$. Rather than $\{0,1\}$ we use the score an agent receives given her (potentially manipulated features) as a budget (lines 209-14 column 2). In particular, notice that any change in features that cost more than the score given by the new feature set would give negative utility. We can also derive an explicit upper bound on the strategic effort vector, $\mathbf{x}_e$. Using Proposition 2.1, $\mathbf{x}_e$ $= A_g^{-1}C^\top\Pi_g\mathbf{w}$. The operator norm inequality gives the bound: $||\mathbf{x}_e|| \leq ||A_g^{-1}C^\top\Pi_g|| ||\mathbf{w}||$ where the norm over the matrix parameters is the operator norm.

---

> > ### Author Rebuttal · Reviewer_bRyY · 2026-04-04
> >
> > Thank you for your explanations. I continue to view this paper making a solid contribution and maintain my positive support.

---

### Official Review · Reviewer_tP79 · 2026-03-18

**Soundness:** 2
**Presentation:** 3
**Significance:** 3
**Originality:** 3
**Overall Recommendation:** 4
**Confidence:** 3

**Summary:**

In this paper, the authors study the trade-off between fairness and accuracy/social welfare in strategic learning. Specifically, compared with standard strategic learning settings, they consider a more complex and realistic scenario in which (1) agents do not know the regressor and can observe only samples from their own group, (2) changes in one feature may affect other features, and (3) an independent stakeholder may impose constraints on the feasibility of the regressors. In this setting, the authors derive upper bounds for the target optimization objective under different types of constraints in convex settings. They further consider non-convex settings and provide experiments on real-world datasets to illustrate the resulting trade-off curves.

**Compliance With Llm Reviewing Policy:**

Affirmed.

**Final Justification:**

I am satisfied with the response and will change my score to 4.

**Key Questions For Authors:**

See the weakness part.

**Limitations:**

yes

**Strengths And Weaknesses:**

**Strengths:**

1. The paper considers a more complex and realistic strategic learning setting, which is valuable.
2. The authors provide a comprehensive theoretical analysis of upper bounds in convex settings.
3. The paper is generally well written.

**Weaknesses:**

1. The main technical contribution of the paper is the set of upper bounds established for convex settings. However, I think the paper would be stronger if the authors could provide more analysis of the tightness of these bounds. For example, could such tightness be examined empirically through experiments?
2. It is somewhat surprising to me that all of the theoretical results are independent of the number of samples, that is, $N\_g$, in each group. At the same time, the authors assume that each agent observes an unbiased sample from the same group, that is, $\hat{y}\_{g,i}$ in Line 193 does not include any noise term, which seems to be a strong simplification. Could the authors explain why the results are independent of the number of samples, and how the results would change if there were noise in the observation of $\hat{y}\_{g,i}$?
3. In Section 4, the authors provide some results for non-convex settings. However, the procedure described there does not seem to be directly applicable to genuinely non-convex settings in practice. Could the authors provide more details on how the non-convex optimization is actually carried out in practice?

---

> ### Author Rebuttal · Authors · 2026-03-31
>
> We thank the reviewer for their thoughtful review and recognition of the comprehensive nature of our model and high-quality writing. We address weaknesses (W1-3) below and are keen to answer any further questions during the discussion period.
>
> **W1: Theory.** Regarding the tightness analysis, we emphasize that this work does not attempt to optimize the tightness of the derived bounds, but several of them are attained (i.e., hold tightly) at particular inputs or even exact, as the Appendix illustrates. Table 1 row 1 bounds are attained (Lemma A.4 and Proposition A.2). The Table 1 row 4 bound for SW is the exact loss by construction (Proposition A.4). Upper bounds corresponding to Properties 3.1 and 3.2, are not tight in general, but they can be attained: since the $\beta$-fair space and the Euclidean ball are nested convex sets, there exists an instance in which $w^\star=w_u^\star$, and $w_c^\star,w^\star, w_u^\star$ are collinear. In that case, $f(w_c^\star)=0$, and the social-welfare loss is exactly $||\tilde w||$.
>
> **W1: Empirics.** We also acknowledge that we could include more explicit intuitive and visual of tightness in Sec 5. To this end, we visualize tightness of an equivalent simulation to Sec 5 on the Taiwan-Credit dataset (https://anonymous.4open.science/r/Blind_3-DB38/blind_experiments%20(2).pdf) for Table 1 Row 2 (which are the bounds that apply to l1 discrepancy functions). We see from the figure: when we can apply Table 1 bounds that include $\beta$ (e.g., row 2 Acc loss), our bounds are tight and adapt to changing $\beta$. On the other hand, bounds that do not include $\beta$ (e.g. row 2 SW Loss), are far less tight and remain constant, meaning they are only potentially useful for very low $\beta$ and nearly worst-case data. In some cases of theoretical bounds, as in row 2 SW loss, it is not possible (or very difficult) to create reasonable bounds that vary with $\beta$ as the geometric properties of the $\beta$-fair space or feasible region do not lend themselves to easily bounding that objective. We do not intentionally avoid discussion of certain bounds being less tight or not adaptive to $\beta$ as we intended this to be inferred from the appearance of $\beta$s in the Table and lines 298-302, but we will make this explicit by including this discussion and figures (as well as equivalents on the original ADULT dataset) in the revision.
>
> **W2.** This is a good observation. Intuitively, the answer for a lack of $N_g$ lies in the fact that while, intuitively, accumulating more or less samples would affect each group's understanding of the principal's rule, $w_{est}$, for the closed form solution to eqn 1, following "Information Discrepancy in Strategic Learning", Bechavod et al 2022, we get that $w_{\text{est}}(g)=\Pi_g w$ exactly. Looking closer at the Bechavod et al proof (their Lemma 3.1), $\Pi_g w$ is a solution because $\hat{y}=(\Pi_gx)^\top w$ (i.e. no outcome noise) and $\Pi_g w$ is the unique solution to eqn 1's l2 norm minimization when $N_g \geq \dim(S_g)$ and non-degenerate sampling in $S_g$. So the number of samples does actively affect our results in the sense that we implicitly assume that $N_g$ satisfies this condition; we will add this as a remark. As for the lack of outcome noise, we refer the reviewer to our response to reviewer ue6z Q3, put simply, as long as outcome noise were mean 0, $w_{est}$ would converge to $\Pi_gw$ as $N_g$ increases. Because we focus on the principal's problem, it is useful to be able to implicitly assume $N_g$ is large and isolate agent disparity effects rather than those due to small samples. We will also add this as a remark to the revision.
>
> **W3.** We interpret this weakness as referencing the fact that in Sec 4, we focus on a particular family of nonconvex discrepancy functions (Def 4.1), which may not be genuinely applicable. In response to this, we would slightly push back and point out that, as highlighted by Ex 4.1, realistic discrepancy functions that fall into this family do exist and thus our analysis directly applies to them. For exceedingly general nonconvex spaces, it is simply a very hard problem to derive good optimality loss bounds. However, our framework does have a broader application: for any non-convex $\beta$-fair region, if one can constructs a bespoke inner ellipsoidal approximation of Eqn 2's feasible region, $\mathcal{B}(1)\cap\mathcal{W}(\beta;\Delta)$, which is always possible, then to this ellipsoid, $\mathcal{E}'(\beta)$, they can apply our Table 1 row 4 bounds, getting an upper bound on $|OBJ(w_{{res}}) - OBJ(w_{u})|> U$, which, because $\mathcal{E}'(\beta)$ is a restriction, upper bounds the loss due to fairness constraints.

---

> > ### Author Rebuttal · Reviewer_tP79 · 2026-04-02
> >
> > I thank the authors for their response. I will adjust my score accordingly.

---

> > > ### Author Response · Authors · 2026-04-02
> > >
> > > We thank for the reviewer for taking the time to read our response and being willing to increase their score.

---

### Decision · Program_Chairs · 2026-04-30

**Decision:**

Accept (regular)

**Comment:**

The paper investigates fairness of incentivized effort in strategic classification problems. The model is quite general, which gives practical relevance to the results. The paper provides bounds on the objective function with imposing various constraints in the convex setting.

All the reviewers found the paper to be a solid theoretical contribution with practical relevance. Some of them note that the empirical evaluation could be more elaborated and I do encourage the authors to include the additional material shown during the rebuttal in their final version. Even though the experiments could be more developed, they do bring some insight and I think the current state of empirical evaluation is sufficient for acceptance given the nature of the contribution.